# Dissecting tumor microenvironment from spatially resolved transcriptomics data by heterogeneous graph learning

Chunman Zuo [1,2] ✉, Junjie Xia[1,3] & Luonan Chen [4,5,6] ✉

Spatially resolved transcriptomics (SRT) has enabled precise dissection of tumor-microenvironment (TME) by analyzing its intracellular molecular networks and intercellular cell-cell communication (CCC). However, lacking computational exploration of complicated relations between cells, genes, and histological regions, severely limits the ability to interpret the complex structure of TME. Here, we introduce stKeep, a heterogeneous graph (HG) learning method that integrates multimodality and gene-gene interactions, in unraveling TME from SRT data. stKeep leverages HG to learn both cell-modules and gene-modules by incorporating features of diverse nodes including genes, cells, and histological regions, allows for identifying finer cell-states within TME and cell-state-specific gene-gene relations, respectively. Furthermore, stKeep employs HG to infer CCC for each cell, while ensuring that learned CCC patterns are comparable across different cell-states through contrastive learning. In various cancer samples, stKeep outperforms other tools in dissecting TME such as detecting bi-potent basal populations, neoplastic myoepithelial cells, and metastatic cells distributed within the tumor or leading-edge regions. Notably, stKeep identifies key transcription factors, ligands, and receptors relevant to disease progression, which are further validated by the functional and survival analysis of independent clinical data, thereby highlighting its clinical prognostic and immunotherapy applications.

Cancer can be viewed as a tumor ecosystem where cancer cells cooperate with non-cancer cells (i.e., immune and stromal cells) within their tumor microenvironment (TME), struggling to survive under various harsh conditions (e.g., hypoxia and oxidative stress)[1–3]. Among them, the cancer cell-states are influenced not only by intracellular molecular networks such as gene regulatory networks (GRNs) but also by external signals transmitted through cell-cell communication

(CCC)[4,5]. Recent studies have emphasized the importance of TME in disease initiation, progression, metastasis, and anti-cancer treatment[6–10]. Therefore, it is an urgent task to comprehensively understand how cancer cells adapt to their TME through their intracellular molecular networks and intercellular CCC.

The current popular spatially resolved transcriptomics (SRT) technologies such as Visium, Stereo-seq[11–13], and NanoString CosMX™

[1]Institute of Artificial Intelligence, Shanghai Engineering Research Center of Industrial Big Data and Intelligent System, Donghua University, Shanghai 201620, China. [2]Key Laboratory of Symbolic Computation and Knowledge Engineering of Ministry of Education, Jilin University, Changchun 130022, China. [3]Department of Applied Mathematics, Donghua University, Shanghai 201620, China. [4]Key Laboratory of Systems Biology, Shanghai Institute of Biochemistry and Cell Biology, Center for Excellence in Molecular Cell Science, Chinese Academy of Sciences, Shanghai 200031, China. [5]Key Laboratory of Systems Health Science of Zhejiang Province, School of Life Science, Hangzhou Institute for Advanced Study, University of Chinese Academy of Sciences, Chinese Academy of Sciences, Hangzhou 310024, China. [6]West China Biomedical Big Data Center, Med-X center for informatics, West China Hospital, Sichuan University, Chengdu 610041, China. ✉e-mail: cmzuo@dhu.edu.cn; lnchen@sibs.ac.cn

Spatial Molecular Imager (SMI)[14], allow for profiling gene expression patterns while preserving spatial location in the tissue[15], resulting in multimodal data that contains histology, spatial location, and gene expression. This provides opportunities to accurately identify molecular networks and CCC associated with different cancer cell-states. However, SRT data analysis is hindered by several challenges: low throughput and sensitivity, and high levels of sparsity and noise[16,17]. Previously, we proposed a multi-view graph collaborative-learning model stMVC, which addresses the challenges of analyzing SRT data for clarifying tumor heterogeneity, by integrating two intercellular graphs through semi-supervision of histological regions (i.e., tumor position) using attention[18]. While stMVC leveraged histological regions as labels to enhance the accuracy of identifying tumor heterogeneity and addressed the problem of non-identical tumor cells within the same tumor region through semi-supervised learning, it unavoidably sacrifices some cell heterogeneity. Furthermore, stMVC primarily focused on capturing global intercellular relations and ignored the local hierarchical associations between cells, genes, and histological regions, thereby hindering its capability to analyze TME heterogeneity.

Recently, various computational methods have been developed to analyze SRT data for identifying spatial domains, CCC, and molecular networks[19,20]. Specifically, (i) deep learning-based models[21] such as SpaGCN[22], STAGATE[23], CCST[24], RESEPT[25], and Squidpy[26], as well as statistics-based models like Giotto[27], BayesSpace[17], SpatialPCA[28], and SOTP[29], have been designed for spatial clustering. Although these methods have yielded many interesting findings, they often do not leverage disease-related information within the histological images, e.g., histological regions, limiting their ability to capture cell heterogeneity accurately; (ii) cell population-based models like Giotto[27], and CellPhoneDB v3[30], as well as single-cell-centered models like SVCA[31], MISTy[32], SpaTalk[33], NCEM[34], and COMMOT[35], have been developed to infer CCC. While these methods have contributed to our understanding of CCC, they may not account for the differences between distinct cell-states, thereby limiting their capacity to construct accurate associations between CCC and disease progression; and (iii) a Bayesian-based statistical model called SpaceX was proposed to identify shared and cluster-specific co-expression networks[36]. SpaceX employed rigorous criteria to infer statistically significant gene-gene interactions, yet it does not leverage any prior knowledge such as GRN and protein-protein interaction (PPI), inevitably resulting in some false-positive relations. Thus, computational methods capable of identifying heterogeneous cell populations in TME and further revealing their internal molecular network and external cellular communication mechanisms are lacking.

Here, we present a unified computational model for SRT data to accurately understand how cancer cells adapt to various conditions by achieving the following three goals: (i) detecting diverse cancer cell-states or cell-modules from heterogeneous TME by integrating multimodal data; (ii) identifying gene-modules specific to each cancer cell-state with the key transcription factors (TFs) and biological functions; and (iii) inferring CCC patterns present in individual cells by globally considering their states, and subsequently predicting the ligand-receptor pairs (LRPs) associated with disease status.

In this work, we introduce stKeep, a graph embedding method that integrates multimodal data (i.e., histology, gene expression, spatial location, and histological regions) and gene-gene interactions (i.e., GRN, PPI, and LRP), to dissect TME heterogeneity by identifying cell-modules, gene-modules, and CCC. Different from previous methods, stKeep utilizes HG to capture intricate relations between cells/spots, genes, and histological regions, where nodes represent different entities and edges capture their relations. The HG is constructed and then an attention-based multi-relation graph embedding algorithm is employed to project diverse nodes into a low-dimensional space. This allows for detecting more cell-states within TME and cell-state-specific gene-gene relations from the learned

embeddings of cell-modules and gene-modules, respectively. Furthermore, stKeep leverages heterogeneous (i.e., multiple) graphs to aggregate ligand signals from neighboring cells for each individual cell, while ensuring that learned CCC accurately characterizes the cell-state differences within TME through contrastive learning. This enables identifying important ligands and receptors associated with disease development. stKeep shows its versatile applications in dissecting heterogeneous TME across diverse cancer types, including human breast (i.e., luminal B, Her2+, and triple-negative breast cancer), lung, colorectal, and liver metastasis, using single-cell or spot (measuring multiple cells) SRT data. It successfully identifies TME-related cancer cell-states such as bi-potent basal populations, neoplastic myoepithelial cells, and metastatic cells within the regions of tumor and leading-edge, along with their key TFs, ligands, and receptors, which are further validated by clinical data from other independent studies, demonstrating potential clinical prognostic and immunotherapy applications from SRT data. Importantly, such an HG learning model employed by stKeep offers a flexible framework to decode TME, not only by integrating SRT data from spatial multi-omics data but also by incorporating spatial epigenomics or proteomics data.

## Results
### Overview of stKeep model
stKeep integrates histological images, spatial location, gene expression, histological regions, and gene-gene interactions such as GRN, PPI, and LRP, to dissect tumor ecosystems, by constructing cell-modules, gene-modules, and CCC through HG learning (Fig. 1a–g). By leveraging HG to capture complex relations among cells/spots, genes, and histological regions, stKeep facilitates the learning of comparable embedding spaces for TME-related cells/spots and genes, leading to the identification of cell-modules and gene-modules, respectively. Furthermore, stKeep leverages HG and contrastive learning to aggregate signals from neighboring cells/spots to infer CCC, which effectively reflects the differences in cell-states within the TME.

stKeep first encodes heterogeneous nodes such as genes, cells/spots, and histological regions (or cell-states) into a unified graph, where nodes represent entities and edges indicate relations between entities. The cell module incorporates various relations including a cell/spot expressing genes and belonging to a histological region, as well as multiple semantic relations. Similarly, the gene module encompasses relations such as a gene being expressed by cells/spots and over-expressed by a specific cell-state, as well as known gene-gene interactions including GRN and PPI. Once the HG is constructed, for each cell/spot ($v_i$), cell module separately calculates local hierarchical representations ($\mathbf{R}_i^1$) from its linked genes, and region and global semantic representations ($\mathbf{R}_i^2$) from its semantically associated cells/spots. stKeep leverages a contrastive self-supervised learning mechanism to link $\mathbf{R}_i^1$ and $\mathbf{R}_i^2$, reinforcing each other and producing two distinct yet semantically related representations (Fig. 1d). The obtained representations $\mathbf{R}$ (i.e., the concatenation of $\mathbf{R}^1$ and $\mathbf{R}^2$) can be further utilized for spatial clustering, visualization, and data denoising (Fig. 1g). Then, for each gene ($v_k$), gene module automatically integrates information from associated cells/spots and cell-state via attention, as well as known gene-gene relations through contrastive learning (Fig. 1e). The resulted gene representations $\mathbf{G}$ enable the identification of cell-state-specific gene-modules by unsupervised clustering (Fig. 1g). Finally, stKeep infers LRP interactions ($\mathbf{L}_i$) for a given cell/spot ($v_i$) by combining ligand information from neighboring cells/spots via attention-based heterogeneous graphs, while assuring that inferred CCC patterns are comparable across different cell-states within TME through contrastive learning (Fig. 1f). The derived LRP interactions $\mathbf{L}$ can be applied to identify LRPs related to different cancer cell-states by differential analysis (Fig. 1g).

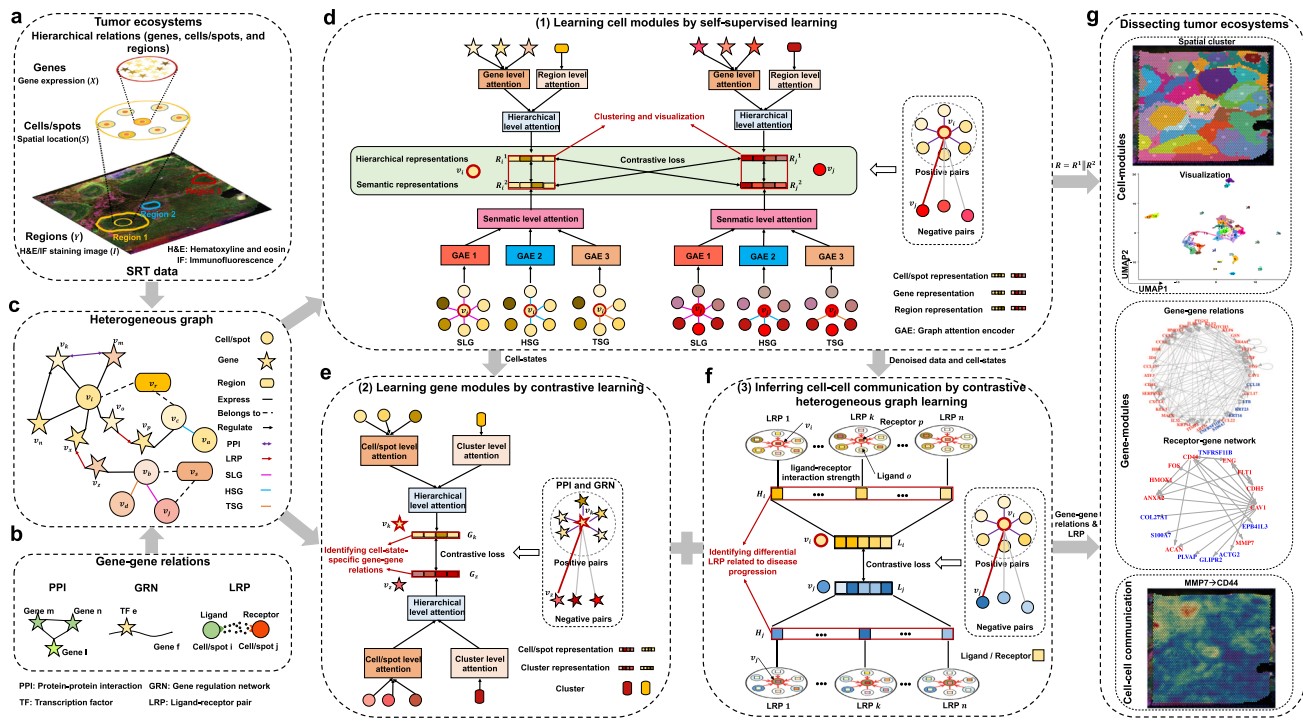

**Fig. 1 | Overview of the stKeep model. a–c** Given each SRT data with four-layer profiles: histological images (**I**), spatial locations (**S**), gene expression (**X**), histological regions (**Y**), and gene-gene interactions such as protein-protein interaction (PPI), gene regulation network (GRN), and ligand-receptor pair (LRP) as the input, stKeep integrates them to construct heterogeneous graph (HG) for dissecting tumor ecosystems. **d** Cell module adopts a cell/spot-centered HG to capture local hierarchical representations (**R**$_i^1$) through aggregating features from genes and regions by attention, while leveraging multiple semantic graphs including spatial location graph (SLG), histological similarity graph (HSG), and transcription similarity graph (TSG) to learn global semantic representations (**R**$_i^2$), and collaboratively integrates two representations by self-supervised learning. **e** Gene module utilizes a gene-centered HG to learn low-dimensional representations by combining features from cells/spots and clusters using attention, while ensuring co-relational gene pairs are embedded adjacent to each other using contrastive learning. **f** Cell-cell communication (CCC) module leverages attention-based heterogeneous graphs to infer ligand-receptor interaction strength (**H**$_i$) for each cell/spot by aggregating ligand information from the neighbors for a central cell/spot, while guaranteeing that CCC patterns can characterize diverse cell-states within TME. Note that each graph indicates one LRP. **g** The unified framework with three modules (**d**–**f**) can be used to dissect tumor ecosystems by detecting spatial clusters and visualizing them, identifying cell-state-specific gene-modules and receptor-gene interaction networks, and inferring cellular communication strength.

## stKeep improves the dissection of heterogeneous cell populations by cell module, gene module, and cell-cell communication

To comprehensively evaluate the performance of stKeep, we analyzed 12 human dorsolateral prefrontal cortex (DLPF) slices from three individual experiments (Supplementary Table 1), where each was manually annotated with layers and white matter (WM) based on morphological features and gene markers[37]. We compared stKeep with three recently developed methods (i.e., Squidpy, STAGATE, and stMVC) for analyzing the DLPFC dataset. We predicted clusters using the Louvain algorithm, assessed clustering accuracy using average silhouette width (ASW)[38], and then visualized low-dimensional representations using uniform manifold approximation and projection (UMAP) spaces. The results showed that the predictions of stKeep are more consistent with the annotations than other methods, but the intra-cluster representation of stKeep seemed to be slightly further away compared to stMVC, suggesting that stKeep helps dissect more heterogeneous cell populations (Fig. 2a, b and Supplementary Fig. 1a, b).

We next investigated whether stKeep can identify layer-specific gene-modules. For each slice, we calculated 50-dimensional features for 2000 highly variable genes using gene module, and then identified gene-modules from these features by the Louvain algorithm. We found that (i) gene pairs identified from gene-modules are more closely characterized than randomly selected gene pairs (Wilcoxon test, $p < 2.22e\text{-}16$, a notation representing very small numbers close to zero); (ii) the gene expression correlation of identified gene pairs is significantly higher than that of randomly selected pairs (Wilcoxon test, $p < 2.22e\text{-}16$); (iii) most gene-modules show over-expression in a specific cluster, and known layer-specific genes are distributed among different gene-modules. For instance, in slice 151507, Layer 1 specific genes such as *AQP4*, *RELN*, and *FABP7* are found in gene-module 2 (detailed layer-specific genes in Supplementary Table 2)[37]; (iv) each gene-module exhibits specific functions: the WM gene-module is linked to central nervous system myelination and oligodendrocyte differentiation; Layer 6 is associated with cell response to glucocorticoid stimulus; Layer 5 relates to the dopamine neurotransmitter release cycle; Layer 4 is involved in various functions like neurofilament bundle assembly, peripheral nervous system axon regeneration, and neuroendocrine cell differentiation; Layer 3 contributes to corticotrophin secretion and the glucocorticoid receptor signaling pathway; Layer 2 is associated with regulating synaptic transmission and glutamatergic; and Layer 1 participates in neurotransmitter uptake and metabolism in glial cells and the formation of the anterior neural plate; and (v) the gene-modules for slice 151507 display layer-specific patterns across at least four other independent slices, particularly in slices 151508–151510, confirming the reliability of the findings. Together, these results indicated that stKeep enables the confident identification of biologically meaningful gene-modules (Fig. 2c–f, Supplementary Figs. 2–4, and Supplementary Table 3).

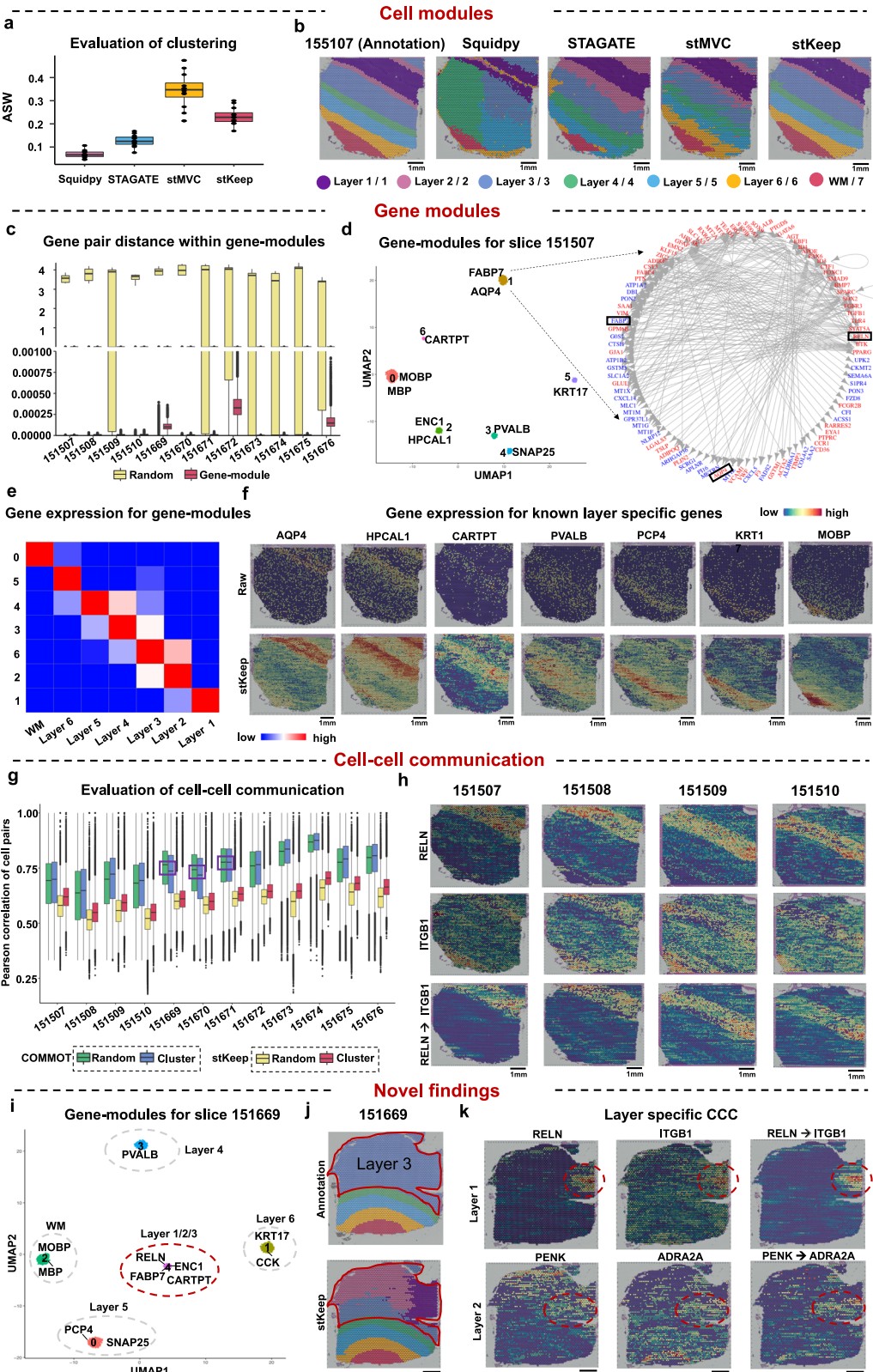

We compared stKeep with COMMOT to assess if the inferred CCC reflects variations among diverse cell populations. We observed that (i) while the correlation of CCC intensities is slightly higher for COMMOT than that of stKeep, significant variations exist in the correlations between spot pairs within clusters and the randomly selected pairs in stKeep versus COMMOT, highlighting that the CCC intensities inferred by stKeep better capture biological relevance and specificity; (ii)

surprisingly, in the CCC analysis by COMMOT on slices 151669–151671, spot correlations within clusters are lower than those of the randomly selected spot pairs. This might be due to COMMOT emphasizing the interplay among diverse ligands and receptors alongside spatial distances, potentially overlooking cell state heterogeneity during CCC inference; and (iii) the CCC model infers layer-specific and shared LRP interaction patterns, assessed through the Gini index-based metric

**Fig. 2 | stKeep identifies cell-modules, gene-modules, and CCC on human DLPFC dataset. a** Boxplot showing average of silhouette width (ASW) for $n = 12$ slices by Squidpy, STAGATE, stMVC, and stKeep. **b** Spatial clusters annotated by the previous study[37], and detected by Squidpy, STAGATE, stMVC, and stKeep, on slice 151507. **c** Boxplot displaying the Euclidean distance of low-dimensional features for gene-pairs from gene-modules and the same number of the randomly selected gene-pairs, for $n = 12$ slices. The number of gene-pairs is as follows: 151507 ($n = 12,023$), 151508 ($n = 14,578$), 151509 ($n = 11,179$), 151510 ($n = 9,645$), 151669 ($n = 11,864$), 151670 ($n = 12,834$), 151671 ($n = 15,996$), 151672 ($n = 15,988$), 151673 ($n = 14,379$), 151674 ($n = 14,917$), 151675 ($n = 13,309$), 151676 ($n = 12,651$). **d** UMAP visualization of gene representations for slice 151507 by stKeep (left panel). Each color indicates a gene-module. Right panel indicates the identified gene-module for Layer 1. Regulators and their target genes are colored in red and blue, respectively. For clear visualization, we displayed the over-expressed genes in Layer 1 versus other layers, with a $\log_2 FC$ greater than 0.8. **e** Heatmap showing mean expression

of the identified gene-modules for slice 151507. **f** Spatial expression of the known layer-specific genes, for slice 151507 data denoised by stKeep, where we also provide raw data as a comparison. **g** Boxplot showing the Pearson correlation of interaction strength between $n = 10,000$ spot pairs within a cluster, inferred for $n = 12$ slices by stKeep and COMMOT, where we also provide $n = 10,000$ randomly selected spot pairs for comparison. For each boxplot of (**a**, **c**, **g**) the center line, box limits and whiskers separately indicate the median, upper and lower quartiles and $1.5\times$ interquartile range. **h** Spatial expression of the highly expressed ligands and receptors, and their corresponding CCC interaction strengths, for $n = 4$ slices. **i** UMAP visualization of gene embeddings for slice 151669 by stKeep. Each gene-module is indicated by one color. **j** Spatial clusters annotated by the previous study[37] (top panel), and predicted by stKeep (bottom panel), on slice 151669. **k** spatial expression of ligands (*RELN* and *PENK*), and receptors (*ITGB1* and *ADRA2A*), and their corresponding CCC interaction strengths, on slice 151669. Source data are provided as a Source Data file.

(see Supplementary Note 1). For instance, notable interactions like $RELN \rightarrow ITGB1$ and $PENK \rightarrow ADRA2A$ are observed in Layer 1 and Layer 2, respectively, while $CALM1 \rightarrow PTPRA$ interactions dominate in Layers 2-6. These findings demonstrate that stKeep consistently estimates CCC intensities across different cell-states, highlighting its ability to capture relevant communication patterns (Fig. 2g, h and Supplementary Fig. 5a–e).

Interestingly, we found a gene-module in slice 151669 (previously annotated with Layers 3, 4, 5, 6, and WM) that exhibited over-expression of marker genes for Layer 1, 2, and 3, suggesting that Layer 3 might be a heterogeneous tissue containing Layer 2 and 1. To validate this finding, we performed further analysis of the identified cell-modules and CCC within annotated Layer 3. We observed that (i) there are three distinct clusters within Layer 3, two of which represent Layers 1 and 2, as verified by known marker genes such as *AQP4* and *FABP7* for Layer 1, and *ENC1* and *HPCAL1* for Layer 2; (ii) Layer 1 and 2 specific CCCs interact highly and specifically in Layer 3; and (iii) more importantly, Layer 1 and 2 specific genes and CCCs show over-activation in Layer 3 across three independent slices:151670, 151671, and 151672, further supporting our findings. These results demonstrated that stKeep improves the annotation accuracy using cell-module, gene-module, and CCC (Fig. 2i–k and Supplementary Fig. 5b–e and 6a, b).

## stKeep contributes to identifying biologically meaningful cell-modules and gene-modules

We tested whether stKeep can clarify different cancer cell-states and their internal gene programs on IDC (i.e., Luminal B) and BAS1 (i.e., Her2$^+$) breast cancer samples published by 10X Genomics (Fig. 3a, g, and Supplementary Tables 1 and 3). We noted that (1) stKeep outperforms other computational methods in detecting more cell-states in cancer-rich regions. Especially in the yellow-marked area of IDC, stKeep, and stMVC identify six clusters, while Squidpy and STAGATE separately identify three and five clusters. In particular, in the areas of invasive cancer outlined in red in both samples, stKeep detects clusters 21 and 28, which other methods fail to identify; and (2) the feature embeddings extracted by stKeep exhibit better separation between different cell-states compared to Squidpy, STAGATE, and stMVC. Furthermore, each cell-state has its specific gene signatures called spatially variable genes (SVGs) (Fig. 3b–d, g, h, and Supplementary Fig. 7a, b, 8a, b).

To validate the accuracy of identified cell-states missed by competing methods, especially cluster 21 in IDC sample, we further investigated their internal gene-modules, key TFs, and associated biological functions. We identified 31 gene-modules comprising 3534 known gene pairs using gene module (Supplementary Fig. 7c, d). The features of the identified gene pairs displayed significantly closer proximity than randomly selected gene pairs. Moreover, the gene expression correlation coefficients among 3534 gene pairs were significantly higher than those in random pairs. On average,

approximately 65% of gene pairs from a gene-module exhibited significant correlations, in contrast to about 14% of random gene pairs. These findings collectively validated the reliability of our identified gene-modules (Fig. 3e). Notably, we identified a gene-module regulated by TFs such as *SREBF1* and *PTK7*, in cluster 21 on IDC sample. Previous studies have shown that *SREBF1* is positively associated with tumor differentiation, tumor-node-metastasis stage, and lymph node metastasis[39], while *PTK7* is linked to cancer cell motility and metastasis[40]. Furthermore, we found a significant correlation between the average expression of 25 signature genes within gene-module and shorter overall survival, which was validated using an independent breast cancer dataset from the TCGA database (Fig. 3f). In addition, for cluster 28 cells on BAS1 sample, we identified a gene-module (with *ENO1* and *PGK1*) regulated by TFs such as *STAT1*[41], that are related to glycolysis/gluconeogenesis and PI3K-Akt signaling pathway (Supplementary Fig. 8c–f).

To assess its capability in understanding the ecosystem at the tumor leading-edge, we chose cluster 29 (annotated with connective tissue) in the BAS1 sample for in-depth analysis. In summary, we observed that (1) there are two gene-modules: one regulated by *TP63* and the other by *RUNX1*, *ETS2*, and *EBF1*. These genes are involved in various functions, including angiogenesis, chemokine and interleukin-mediated signaling pathways, collagen degradation, mesenchyme migration, innate immune response, cell-cell adhesion, neutrophil and monocyte chemotaxis; (2) the cells exhibit over-expression of markers for basal and myoepithelial cells (e.g., *KRT23*, *KRT6B*, *KRT5/14*, and *ACTA2*), cancer-related fibroblasts (CAFs) (*COL1A1* and *COL1A2*), endothelial cells (*ENG* and *VWF*), as well as interleukins and chemokines (*CXCL8*, *CX3CL1*, *CCL17*, and *CXCL3*)[42] (Fig. 3h–k and Supplementary Fig. 8g, h). Collectively, these results suggested that cluster 29 represents a heterogeneous tissue composed of basal tumor cells, myoepithelial cells, and a fibrovascular niche, indicating a potential route for cancer cell invasion through migration, which is consistent with previous findings[43]. This opens up the possibility of identifying further therapeutic targets for the treatment of aggressive cancers.

Taken together, stKeep is more conducive to detecting cell-states (or cell-modules) missed by other methods, and further clarifying how cancer cells regulate internal gene programs (or gene-modules) to adapt to diverse conditions, which has the potential for clinical targeting therapy and prognostic applications.

## stKeep enhances the detection of bi-potent basal populations by cell-cell communication and gene module

To demonstrate the ability of stKeep to elucidate how cancer cells cooperate with immune and stromal cells in different cancer ecosystems, we conducted further analysis on the IDC dataset. We inferred 2681 ligand-receptor interactions for each cell through CCC model, and found that the LRP interaction strength correlation between spots within one cell-state is significantly higher than that of randomly

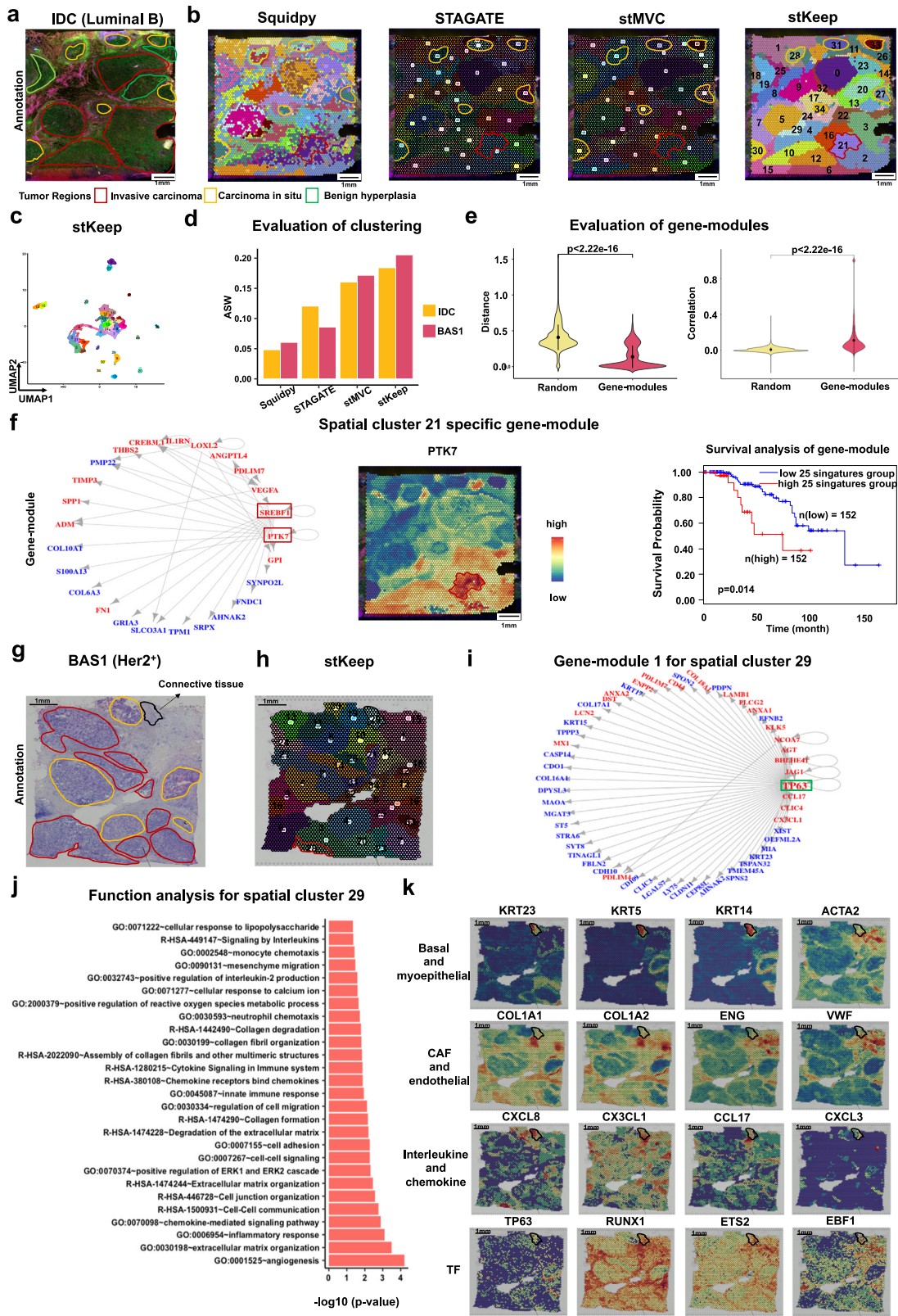

selected spot pairs (Wilcoxon test, $p < 2.22\text{e-}16$), highlighting the biological relevance and specificity of the inferred LRP strength for each cell-state.

The spatial distribution of classical marker genes across different cell types in cancer cell-states, indicated the heterogeneity of TME (Supplementary Fig. 9a). To investigate the involvement of different cell types in CCC within diverse cancer cell-states, we reanalyzed the

scRNA-seq data of 21,580 cells with 24 cell types from seven independent luminal B patients[44]. We separately identified 392 and 348 cell type-specific ligands and receptors from scRNA-seq data (Fig. 4a, b), and further utilized them as reference for analyzing inferred CCC. We noted that (1) different cell types show varying degrees of involvement in different cancer cell-states; (2) CCC is more activated in benign hyperplasia and invasive cancer regions compared to the carcinoma

**Fig. 3 | stKeep enables us to detect biologically meaningful cell-modules and gene-modules on IDC (Luminal B) and BAS1 (Her2⁺) samples.**
**a** Immunofluorescent staining of IDC tissue with 13 regions: invasive carcinoma (red), carcinoma in situ (orange), and benign hyperplasia (green). The intensity of DAPI, fiducial frame, and anti-CD3 is indicated by green, blue, and yellow. **b** Spatial clustering by Squidpy, STAGATE, stMVC, and stKeep. **c** UMAP visualization of latent features by stKeep. Each color indicates a cluster. **d** Bar plot of clustering score ASW on IDC and BAS1 samples. **e** Boxplot showing the Euclidean distance of low-dimensional features (left panel) and Pearson correlation of gene expression (right panel), for $n = 3,534$ gene pairs identified from gene-modules and $n = 3,534$ randomly selected gene pairs. Unadjusted two-sided unpaired Wilcoxon test. Note that, on average, in a gene-module there are roughly 12% of gene pairs with negative correlations, whereas only about 1.2% of them exhibit significant negative correlations. For each boxplot, the center line, box limits and whiskers separately indicate the median, upper and lower quartiles and $1.5\times$ interquartile range. **f** The

identified gene-module for cluster 21 by stKeep (left panel). Regulators and their target genes are indicated in red and blue. Spatial expression of key TF (*PTK7*) in the gene-module (center panel). Total survival rate of patients with the average expression of $n = 25$ signature genes for cluster 21 in luminal B breast cancer from TCGA by GEPIA2[97] (right panel). Unadjusted one-sided Log-rank test. **g** H&E plot with 13 tumor regions on the BAS1 sample. The black outline indicates connective tissue, while the orange and red colors are consistent with (**a**). **h** Spatial clustering by stKeep. **i** One gene-module for cluster 29 by stKeep. **j** Functional annotation of gene-modules for spatial cluster 29 by DAVID. Unadjusted one-sided Fisher's exact test. **k** Spatial expression of key markers for basal and myoepithelial cells (*KRT23*, *KRT5*, *KRT14*, and *ACTA2*), CAFs (*COL1A1* and *COL1A2*), endothelial cells (*ENG* and *VWF*), interleukins and chemokines (*CXCL8*, *CX3CL1*, *CCL17*, and *CXCL3*), and key TFs (*TP63*, *RUNX1*, *ETS2*, and *EBF1*) in cluster 29. Source data are provided as a Source Data file.

in situ region, except for cells in spatial cluster 28; and (3) luminal progenitors are more involved in spatial cluster 28, consistent with their higher cell populations as estimated by GraphST[45]. In addition, cells in spatial cluster 28, characterized by high expression levels of markers for both luminal progenitors (e.g., *KRT8*, and *KRT18*) and basal progenitors (e.g., *KRT14*, *KRT5*, and *KRT6A/B*), are identified as the bio-potent basal populations[46] (Fig. 4c, d and Supplementary Fig. 9b, c).

To confirm the potential role of spatial cluster 28 cells as cells-of-origin in breast cancer, we conducted additional analysis of their internal gene-modules and external CCC. We found that (1) two gene-modules regulated by some key TFs, play a crucial role in promoting breast cancer stem-like cells. For example, (i) *FOS* regulates pro-inflammatory cytokine *TNF*, activating the NF-kB and MAPK signaling pathways, thereby increasing the proportion of stem-like cells[47,48]; (ii) *CEBPD* links IL-6 and HIF-1 signaling to promote stem-cell associated properties[49], and also the PI3KAkt signaling pathway is known to be essential for stem-cells maintenance[50]; and (iii) basal and luminal progenitor markers are assigned to different gene-modules; (2) *CD44*, a well-established breast stem-cell marker[51], and is most significantly over-expressed receptor in spatial cluster 28 cells. *FOS* and *TNF* are identified as regulators of *CD44*; the ligands *MMP9/7* are also highly expressed in these cells; and significant CCCs are observed between *MMP9/7* and *CD44*, with analysis data indicating that *CD44* can regulate the expression of *MMP7*. Together, these findings demonstrated the presence of a positive loop for maintaining stem properties[52] (Fig. 4e–h and Supplementary Figs. 7e and 9c).

In conclusion, by leveraging CCC and gene model, as well as scRNA-seq data, stKeep presents a promising approach to identifying cancer cell regions with the activities of basal and luminal cells, thus offering an avenue for exploring lineage hierarchies in breast cancer research.

## stKeep enables discovering key TFs and ligand-receptors in neoplastic myoepithelial cells by integrating scRNA-seq data

To demonstrate the capability of stKeep in dissecting intratumor heterogeneity, we utilized a triple-negative breast cancer (i.e., TNBC) sample from a previous study[44] (Fig. 5a and Supplementary Tables 1 and 3). Compared to other methods, stKeep (i) helps detect more cancer cell-states within the tumor region, which aligns with prior observations; and (ii) is able to identify three distinct functional subclusters within cluster 9, which was achieved using hierarchical clustering analysis based on the cell proportions of different cell types (Fig. 5b, Supplementary Figs. 10a–e and 11a, and Supplementary Note 2). Additionally, we applied graph-based clustering analysis to scRNA-seq data of 1,627 epithelial cells from the same patient. This analysis revealed 10 distinct clusters characterized by differentially expressed genes (DEGs) (Fig. 5c).

By deconvoluting SRT with scRNA-seq data using GraphST[45], we found that (1) luminal progenitor 1, myoepithelial 1, and mature

luminal cells are primarily enriched in the normal ductal region, while cancer basal and cycling cells display enrichment in the carcinoma in situ region, consistent with previous annotations; (2) interestingly, luminal progenitor 2, previously classified as normal, is exclusively enriched in the invasive cancer region. Compared to luminal progenitor 1, their over-expressed genes are related to cell adhesion, B cell mediated immunity, and T cell activation. More importantly, this enrichment was observed in two independent datasets: bi-potent basal populations in cluster 28 on the IDC sample, and a niche comprising basal tumor cells and CAFs in cluster 29 on the BAS1 sample, providing further support for our findings. These results suggest that luminal progenitor 2 is associated with intratumor heterogeneity and pro-invasiveness[53]; and (3) similarly, myoepithelial 2 cells are exclusively enriched in the invasive cancer region. Compared with myoepithelial 1 cells, their up-regulated genes are involved in antigen processing and presentation, innate and adaptive immune responses, and cell migration. These findings collectively suggest that myoepithelial 2 cells may be the neoplastic cells, and contribute to increasing the invasiveness of tumor cells, in line with the previous research[54,55] (Fig. 5d–f and supplementary Fig. 11b, c).

To verify the role of myoepithelial 2 cells in promoting cancer invasiveness, we analyzed their intracellular genetic program and extracellular CCC. We noted that (1) *TP63*, crucial for maintaining basal epithelial identity and promoting cell invasion and stemness[56], regulates a gene-module specific to myoepithelial 2 cells; (2) myoepithelial cells are enriched in both normal ductal and invasive carcinoma regions, especially myoepithelial 2 cells, which are primarily situated at the margins of the carcinoma nest and in contact with the stroma, aligning with the prior findings[57,58], indicating that these cells may originate from neoplastic/cancer and undergo malignant transformation; (3) *COMP* is exclusively secreted by myoepithelial 2 cells compared to all other cell types using Gini index[18] (i.e., a statistical measure to estimate a degree of inequality in the distribution of genes among different cell populations); (4) in the invasive carcinoma region, the gene signature scores of myoepithelial 2 cells and CAFs are significantly correlated with *COMP* expression, with coefficients of 0.38 and 0.49, respectively. Notably, myoepithelial 2 cells express *COMP* much more than CAFs in the scRNA-seq data; (5) strong interactions between *COMP* and its receptors (*ITGAV*, *SDC1*, *ITGB1*, and *ITGA5*) are inferred by stKeep in the invasive cancer region. These inferred CCCs are significantly associated with the signature scores of myoepithelial 2 cells and CAFs, respectively. Furthermore, these receptors are involved in cancer progression, metastasis, and invasiveness[59–62]; and (6) *COMP* expression significantly correlates with shorter overall survival, as observed in an independent TNBC dataset from the TCGA database. A study has highlighted that *COMP* contributes to disease severity through metabolic switching (i.e., Warburg effect), and enhanced tumor cell viability[63]. Collectively, both myoepithelial 2 cells and CAFs can promote cancer invasiveness through the expression of *COMP*.

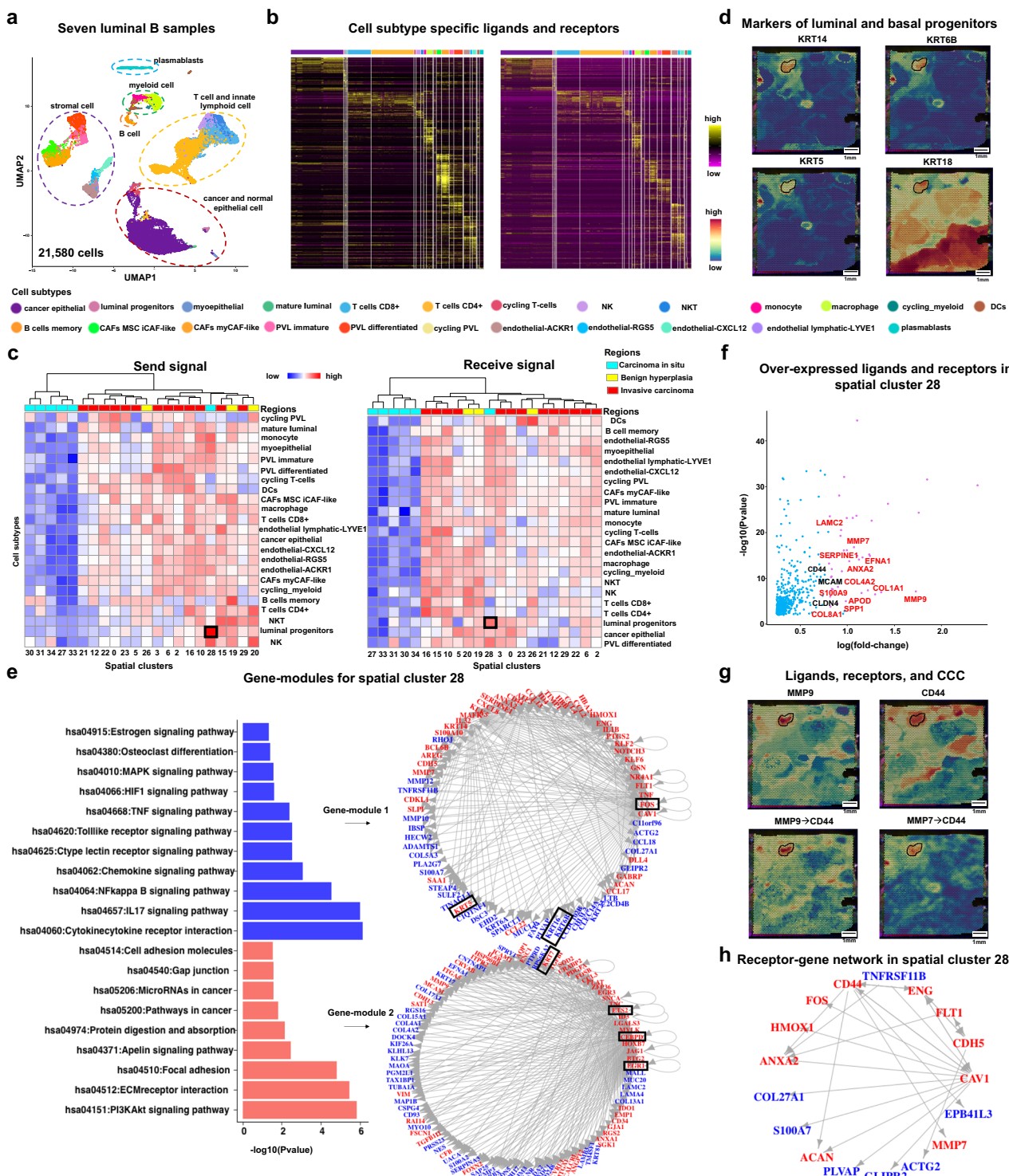

**Fig. 4 | stKeep can detect bio-potent basal populations by cell-cell communication on IDC sample. a** UMAP plot for 24 cell subtypes for 21,580 cells from seven Luminal B patients. Each color denotes a cell subtype. **b** Heatmap showing the expression levels of ligands and receptors for each cell subtype. **c** Heatmap displaying the number of ligands (left panel) and receptors (right panel) by each cell subtype from the inferred CCC by stKeep. Rows and columns denote different cell subtypes and spatial cancer clusters. **d** Spatial expression of the markers of basal progenitor (*KRT14*, *KRT6B*, and *KRT5*) and luminal (*KRT18*). **e** Functional annotation and visualization of two gene-modules for spatial cluster 28 by stKeep. Unadjusted one-sided Fisher's exact test. Note that regulator and target genes are colored in red and blue, respectively. **f** Scatter plot showing the genes over-expressed by spatial cluster 28, where red and black indicate ligand and receptor, respectively. Unadjusted two-sided unpaired Wilcoxon test. **g** Spatial expression of highly expressed ligand and receptor, and interaction strength of their corresponding CCC. **h** The identified receptor-gene interaction networks for spatial cluster 28 by stKeep. Source data are provided as a Source Data file.

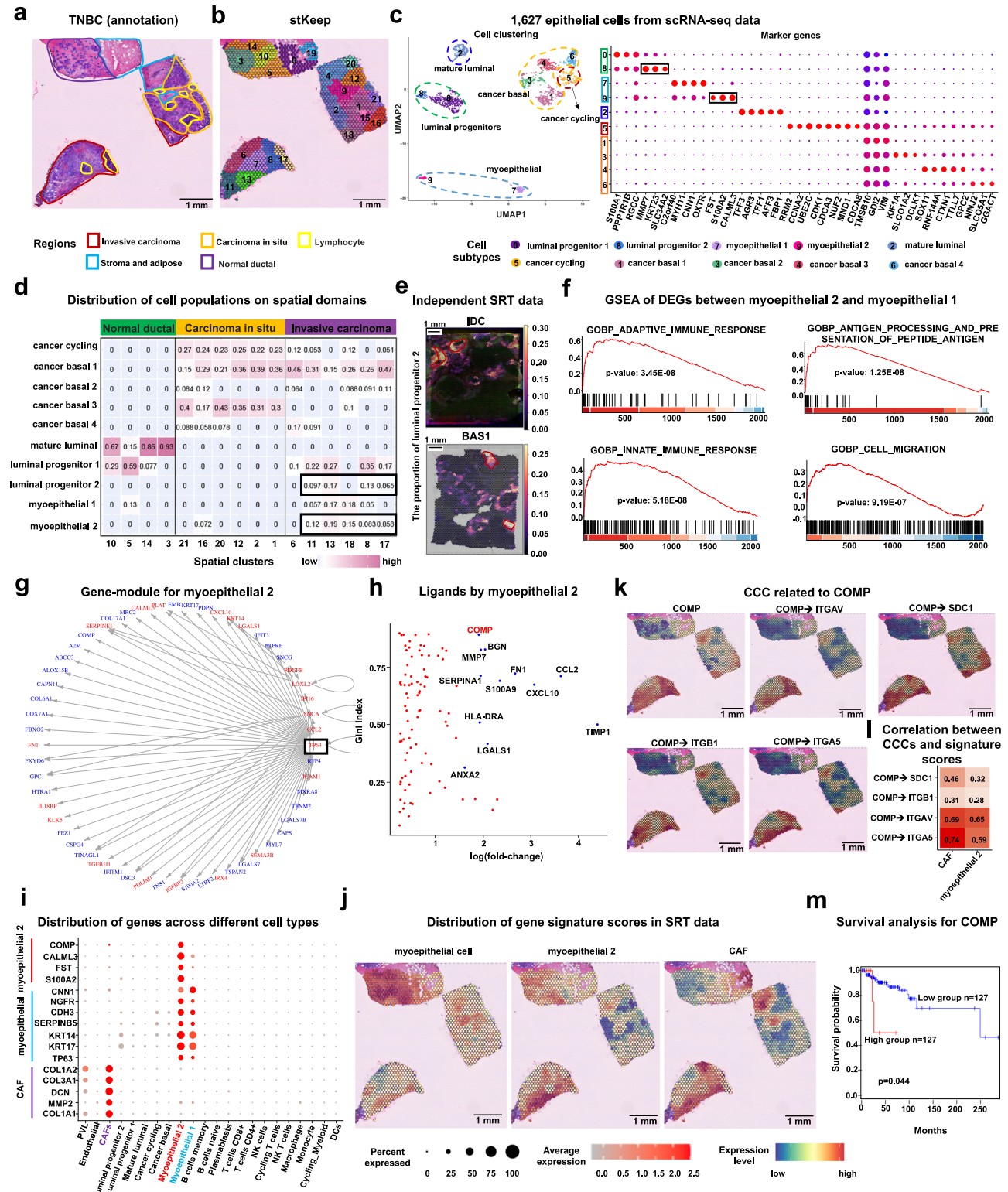

Additionally, our analysis revealed that *COMP* is regulated by *TP63*, suggesting that myoepithelial 2 cells potentially upregulate *TP63* and its target *COMP* to enhance cancer invasiveness (Fig. 5g–m and Supplementary Fig. 11d, e).

In summary, by employing cell module, gene module, and CCC, stKeep enables the identification of neoplastic myoepithelial cells, and discovers important genes such as TFs, ligands, and receptors, associated with disease progression. These findings may provide targets for immunotherapy in the treatments of TNBC.

**stKeep facilitates identifying metastasis cells by integrative analysis of primary colorectal cancer and liver metastasis**

To illustrate the versatile application of stKeep in deciphering heterogeneous TME across different cancer types, we analyzed primary colorectal cancer (i.e., P1) and paired liver metastasis (i.e., LM1) from a recent study[64] (Fig. 6a and h, and Supplementary Tables 1 and 3). stKeep consistently outperforms other methods in detecting more cancer cell-states in P1, particularly cluster 6 (purple, supplementary 12a). This finding was supported by three independent ways: (1)

**Fig. 5 | stKeep identifies neoplastic myoepithelial cells on TNBC sample. a** H&E plot with 16 histological regions: invasive carcinoma (red), carcinoma in situ (orange), lymphocyte (golden), stromal and adipose (blue), and normal ductal (purple). **b** Spatial clustering by stKeep. **c** UMAP plot showing 10 epithelial cell subtypes (left), with marker for each subtype (right). Markers for each subtype are as follows: luminal progenitor 1 (*S100A1*, *PPP1R1B*, and *RGCC*), luminal progenitor 2 (*MMP7*, *KRT23*, and *SLC34A2*), myoepithelial 1 (*C2orf40*, *MYH11*, *CNN1*, and *OXTR*), myoepithelial 2 (*FST*, *S100A2*, and *CALML3*), mature luminal (*TFF3*, *AGR3*, *TFF1*, *AFF3*, and *FBP1*), cancer cycling (*RRM2*, *CCNA2*, *UBE2C*, *CDK1*, *CDCA3*, *NUF2*, *MND1*, and *CDCA8*), cancer basal 1 (*TMSB10*, *GDI2*, and *VIM*), cancer basal 2 (*KIF1A*, *SLCO1A2*, and *DCLK1*), cancer basal 3 (*SOX11*, *RNF144A*, *CTXN1*, *TTLL7*, and *GPC2*), cancer basal 4 (*NINJ2*, *SLCO5A1*, and *GGACT*). Dot size and color indicate the percentage and mean expression level of each gene. **d** Heatmap showing the proportion of epithelial subtypes in different clusters. **e** Spatial distribution of luminal progenitor 2 in IDC and BAS1 samples. **f** Gene set enrichment analysis (GSEA) of DEGs in myoepithelial 2 versus myoepithelial 1. Unadjusted one-sided Kolmogorov–Smirnov test. **g** Myoepithelial 2 gene-module by stKeep. Here are the top 200 over-expressed genes in myoepithelial 2 versus other clusters. Regulators and their target genes are indicated in red and blue, respectively. **h** Scatter plot displaying the specificity and fold-change level of up-regulated ligands in myoepithelial 2 versus myoepithelial 1. **i** Dot plot displaying expression levels of representative genes for myoepithelial 2, myoepithelial, and CAF across various cell types. **j** Spatial feature plots of gene signature score for myoepithelial 2 (*CALML3*, *FST*, and *S100A2*), myoepithelial (*CNN1*, *NGFR*, *CDH3*, *SERPINB5*, *KRT14*, *KRT17*, and *TP63*), and CAF (*COL1A2*, *COL3A1*, *DCN*, *MMP2*, and *COL1A1*). **k** Spatial expression of *COMP* and its interaction strength with four receptors. **l** Spearman correlation between inferred CCCs for *COMP* and gene signature scores for CAFs and myoepithelial 2 cells. **m** Overall survival rate of TNBC patients based on *COMP* expression using TCGA by GEPIA2. Unadjusted one-sided Log-rank test. Source data are provided as a Source Data file.

cluster 6 cells exhibited high expression of genes (e.g., *SPP1*, *FN1*, *APOE*, and *IFIT1*) associated with shorter overall survival in colon adenocarcinoma (COAD) from the TCGA database; (2) the deconvolution of SRT data by scRNA-seq of 4,069 cells from one independent patient[65] confirmed diverse cell populations in different clusters, i.e., plasma cells are primarily enriched in normal and stomal regions, while CMS2 (i.e., cancer cell), stomal and immune cells exclusively display enrichment in tumor region; and (3) *SPP1* and *FN1* showed the highest potential for secretion by *SPP1*+ macrophages and stomal cells in cluster 6, respectively. These secreted factors interacted with cancer cells via receptors (e.g., *ITGAV*, *ITGB5*, *ITGA6*, and *ITGB6*), influencing cell migration, invasion, and initiation of metastasis[66] (Fig. 6b–g and supplementary Fig. 12b, c).

We conducted further analysis on liver metastasis (i.e., LM1) data, and observed that (1) stKeep and STAGATE detect clusters 5 (yellow) and 4 (black) at the tumor leading-edge, but stKeep provides a more structured region detection; (2) similar to P1, plasma cells display enrichment in normal region, while CMS2, stomal, and immune cells are exclusively enriched in tumor region. Interestingly, cluster 4 in normal liver shows CMS2 enrichment, indicating the presence of metastasis cancer cells; (3) *EREG* and *AREG* exhibit the highest and specific potential for secretion by immune or epithelial cells in cluster 5, and interact with the receptor *ERBB3* of cancer cells to promote tumor progression[67]. Previous studies have shown that *EREG*, *AREG*, and *ERBB3* are liver metastasis-associated genes[68,69]. In addition, we observed over-activation of these interactions at the tumor leading-edge in an independent liver metastasis sample (i.e., LM2). These findings suggested that cluster 5 cells employ the CCC mechanisms to promote cancer cell metastasis to the normal liver (Fig. 6h–m and supplementary Fig. 12d, e).

Taken together, stKeep is a valuable tool for dissecting the tumor ecosystem in diverse cancer types, allowing for the identification of key mechanisms associated with disease progression and distant metastases, which has the potential to uncover therapeutic targets for combating cancer metastasis.

## Discussion

This study introduces stKeep, a heterogeneous graph learning approach for analyzing SRT data to dissect tumor ecosystems by identifying cell-modules, gene-modules, and CCC, based on multimodal data (i.e., histological images, spatial location, gene expression, and histological regions) and gene-gene interactions (i.e., GRN, PPI, and LRP). stKeep encodes diverse nodes such as genes, cells, and histological regions (or cell-states) into a unified graph, where nodes indicate entities and edges represent relations between nodes. The learned cell-modules by incorporating information from related genes and histological regions, and semantically linked cells, facilitates detecting finer cell-states within TME. Similarly, the learned gene-modules by integrating information from associated cells and cell-state, and gene-gene relations, enables identifying gene-modules specific to cell-states. Furthermore, distinct from previous CCC methods, stKeep employs HG and contrastive learning to infer CCC reflecting the differences in cell-states within TME. The resulting CCC can help identify ligand-receptor pairs involved in disease development. Ablation studies demonstrated the significance and impact of various graphs within stKeep (Supplementary Note 3 and Supplementary Figs. 13a–c and 14a). Additionally, comprehensive comparisons revealed that stKeep is robust to incomplete or incorrect gene-gene interactions (GRN and PPI) and varying numbers of histological regions (Supplementary Notes 4 and 5 and Supplementary Figs. 14b–e, 15a, 16a, and 17a–d).

The evaluations on human breast, lung, colorectal, and liver metastasis cancer samples demonstrated the unique advantages of stKeep described above, which can detect more TME-related cell-states including bi-potent basal populations with characteristics of both basal and luminal, neoplastic myoepithelial cells, and metastatic cells distributed in tumor or leading-edge regions, providing valuable biological insights into the heterogeneity of tumor ecosystems. In particular, in the TNBC dataset, we demonstrated potential immunotherapy application from SRT data, by identifying neoplastic myoepithelial cells misclassified as normal cells in the previous study, along with their key TFs, ligands, and receptors. Furthermore, we revealed cancer cell invasion mechanisms by identifying adjacent connective tissue near ductal carcinoma in site, consisting of basal cells and a fibrovascular niche. In addition, we identified key cell populations and CCC mechanisms associated with colorectal cancer cells metastasis to normal liver. These biological findings were further validated by clinical data.

In this study, we primarily focused on the spot-based SRT data from fresh frozen samples. We also demonstrated the effectiveness of stKeep in elucidating complex structures in formalin-fixed, paraffin-embedded (FFPE) samples. By analyzing an FFPE (Her2+ breast cancer) sample from Visium, stKeep can detect more cancer cell-states, such as cluster 15, which is affected by *EGF*-regulated CCCs such as *MIF* and *CD44* (Supplementary Fig. 18a–g). Moreover, we have shown the versatility of stKeep in analyzing a non-small-cell lung cancer (NSCLC) FFPE sample (~100 K cells) by NanoString (see Supplementary Note 6). The results highlighted the efficiency of stKeep in detecting diverse cancer cell-states, especially within the interface region between tumors and stromal cells, and also elucidating the possible mechanisms involving *CAV1*-regulated CCCs like *COL3A1* and *DDR1* in mediating cell proliferation[70] (Supplementary Fig. 19a–g).

By leveraging the strengths of both scRNA-seq and SRT data, stKeep offers a comprehensive understanding of TMEs and their underlying biological mechanisms. scRNA-seq quantifies individual cell transcriptomes, enabling the characterization of distinct cell populations. Meanwhile, SRT preserves spatial location, identifying tissue structures and facilitating CCC inference. Compared with

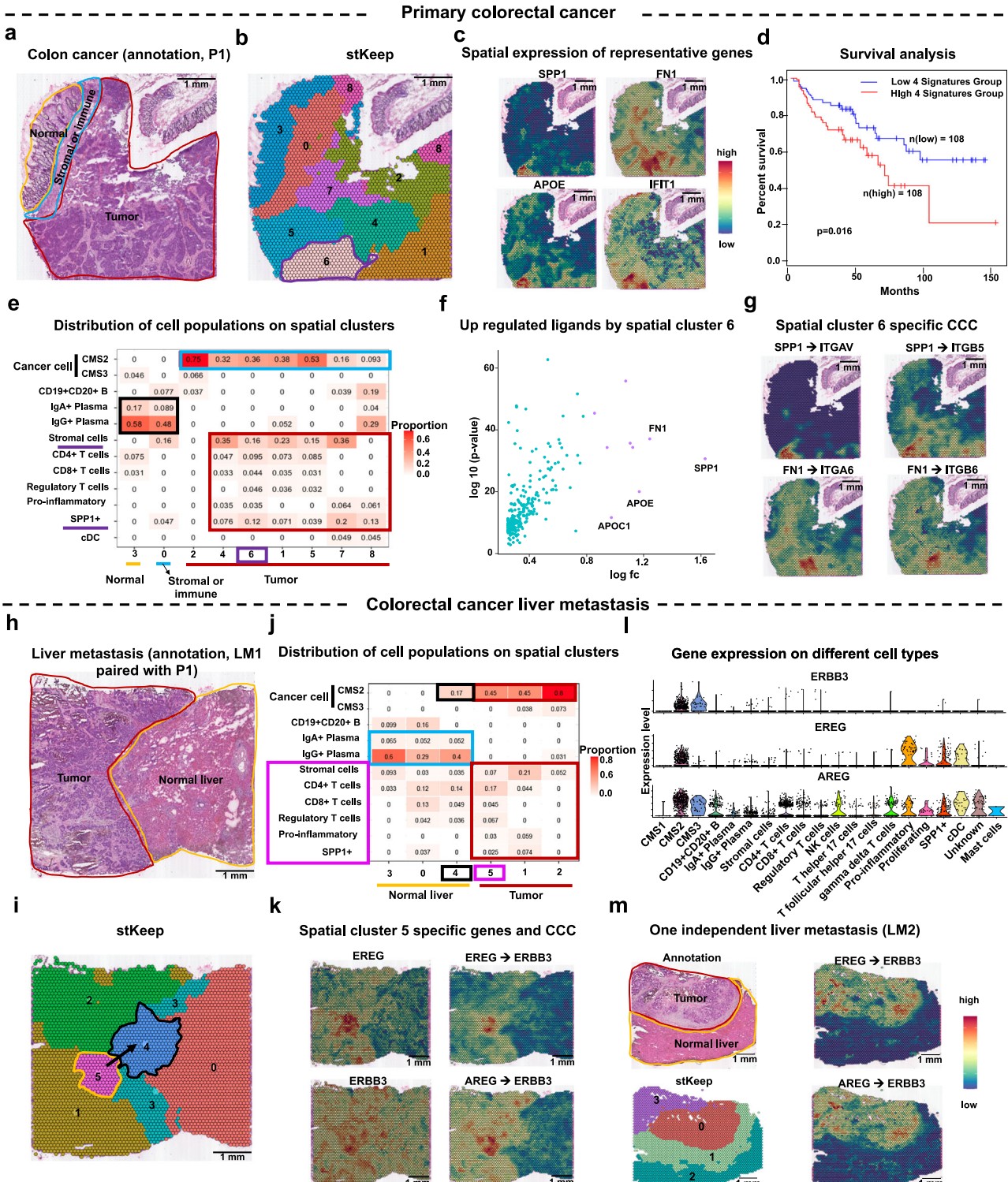

**Fig. 6 | stKeep is able to identify metastasis cells by analyzing primary color-ectal cancer (P1) and paired liver metastasis (LM1) samples. a** H&E plot of P1 tissue with manual annotation of three regions: cancer (red), stromal and immune (blue), and normal (yellow). **b** Spatial clustering by stKeep on P1 sample. **c** Spatial expression of representative genes: *SPP1*, *FN1*, *APOE*, and *IFIT1*. **d** Overall survival rate of COAD patients based on mean expression of four representative genes from TCGA using GEPIA2. Unadjusted one-sided Log-rank test. **e** Heatmap showing cell subtype proportions in different spatial clusters. **f** Scatter plot of over-expressed ligands in spatial cluster 6. Unadjusted two-sided unpaired Wilcoxon test. **g** Spatial interaction strength of LRPs: *SPP1 → ITGAV*, *SPP1 → ITGB5*, *FN1 → ITGA6*, and *FN1 →*

*ITGB6*. **h** H&E plot of LM1 tissue with manual annotation of two regions. The annotated colors are consistent with (**a**). **i** Spatial clustering by stKeep on LM1 sample. **j** Heatmap displaying cell subtype proportions in different spatial clusters. **k** Spatial expression of highly expressed ligand and receptor, and inter-action strength of their corresponding CCC. **l** Violin plot of gene expression of *ERBB3*, *EREG*, and *AREG* across cell subtypes. **m** H&E plot of LM2 tissue with manual annotation of two regions (top left panel). Spatial clustering by stKeep on LM2 sample (bottom left panel). Spatial interaction strength of LRPs including *EREG → ERBB3*, *AREG → ERBB3* (right panel). Source data are provided as a Source Data file.

previous methods[71], stKeep adopts a different approach to integrate these two types of data. Specifically, (i) detecting key players within TME via CCC: Utilizing cell type-specific ligands and receptors from scRNA-seq, stKeep quantifies the involvement of different cell populations in various cancer cell-states (or cell-modules), pinpointing key cell types; (ii) identifying cell-state specific gene-modules: Gene-modules might contain genes expressed across multiple cell types, however, the identification of gene-gene relations from the known PPI and GRN knowledge ensures its accuracy. stKeep utilizes scRNA-seq data to infer where co-related gene pairs might be co-expressed; and (iii) predicting crucial mechanisms in disease development: Mapping cell subtypes from scRNA-seq over different spatial cancer regions, stKeep combines cell subtype-specific over-expressed genes and gene-modules with the inferred CCC from SRT data to uncover regulatory mechanisms relevant to disease progression. In our data integration analysis, the two types of data might not originate from the same patient. Combining scRNA-seq and SRT data offers valuable insights into tumor ecosystem heterogeneity and holds promise for innovating cancer immunotherapy.

We benchmarked the running time and memory usage of stKeep on the simulated datasets by subsampling cells from NSCLC sample. We found that stKeep is fast, and it takes 24 min and 13GB of memory to process the SRT dataset with 17 K cells. In particular, the running time is approximately linearly proportional to the number of input cells (Supplementary Fig. 20), which is considered as an advantage of stKeep for processing a bigger dataset.

In addition to the multiple intercellular semantic relations encoded from spatial location, histology, and gene expression data, our knowledge-primed HG model can be extended to model more complicated intercellular associations, for example, CCC by ligands and receptors (i.e., cell $i \rightarrow$ ligand $m \rightarrow$ receptor $n \rightarrow$ cell $j$), which can be used to complete the CCC-based knowledge graph[72]. With the advance of spatial chromatin accessibility (ATAC-seq)[73], proteomics[74], and mass spectrometry imaging (MSI)[75,76], stKeep exhibits flexibility and can be easily adapted by replacing the feature matrix from gene expression data with that from ATAC-seq, proteomics, and MSI data. Furthermore, with the advance of spatial multi-omics technology[77], stKeep can be further adapted by either adding more graphs created by different omics data or substituting the feature matrix from single-omics data with that from multi-omics data fusion[78,79]. Furthermore, stKeep focuses on elucidating the TME of intra-tumoral heterogeneity rather than inter-tumoral heterogeneity. Biological variations (e.g., gender, age, medical treatment, and disease status) as well as technique challenges (e.g., batch effects in sample preparation or sequencing) create complexities when attempting to simultaneously analyze intra- and inter-tumor heterogeneity within a unified framework. To address this issue, we propose to combine stKeep with SRT integrative tools[80–82] to provide a more comprehensive understanding of both intra- and inter-tumor heterogeneity, leveraging their strengths in analyzing different aspects of tumor complexity across diverse samples.

There are still some limitations in stKeep. Specifically, we (1) identified known co-relational gene pairs within gene-modules to minimize false positives. However, many co-related genes for each cluster might remain unidentified. Exploring shared biological behaviors among gene-gene interactions shared by multiple clusters is crucial. Additionally, while we focused on establishing associations between genes, considering the directions of their relations is essential. In future studies, we plan to develop sophisticated algorithms to infer more gene-gene relations and their directions by carefully leveraging the publicly published ATAC-seq or Chip-seq data; and (2) identified the histological regions from histological images may be time-consuming. To address it, we intend to develop computational models based on the segment anything model (SAM)[83] to automatically segment the tumor regions from Hematoxylin and Eosin (H&E) or immunofluorescence (IF) staining images.

## Methods

### stKeep model
stKeep integrates four-layer profiles from SRT data: histological image ($\mathbf{I} = (i_1, \ldots, i_n) \in R^{\text{width} \times \text{height}}$), spatial location ($\mathbf{S} = (\mathbf{s}_1, \ldots, \mathbf{s}_n) \in R^{n \times 2}$, $\mathbf{s}_i = (s_{ix}, s_{iy})$), gene expression ($\mathbf{X} \in R^{m \times n}$) with $m$ and $n$ being the number of genes and cells/spots, and histological regions ($\mathbf{Y} = (y_1, \ldots, y_n)^T \in R^{n \times 1}, y_i \in \{1, \ldots, K\}$) with $K$ being the number of regions, as well as gene-gene interactions like PPI, GRN, and LRP, in dissecting tumor ecosystems by HG learning (Fig. 1a–g). stKeep employs a three-step approach to analyze SRT data. Specifically, it (1) learns cell-module representations ($\mathbf{R}_i \in R^{2d \times 1}$) by combining local hierarchical representations ($\mathbf{R}_i^1 \in R^{d \times 1}$) from related genes and regions via HG, with global semantic representations ($\mathbf{R}_i^2 \in R^{d \times 1}$) from associated cells/spots through multiple graphs, where two types of graphs are collaboratively trained each other in the representation space (Fig. 1d); (2) learns the gene-module representations ($\mathbf{G}_k \in R^{b \times 1}$) by aggregating information from cells/spots and cell-state via HG, and utilizing known gene-gene interactions through contrastive learning (Fig. 1e); and (3) infers ligand-receptor interaction strengths ($\mathbf{H}_i \in R^{M \times 1}$ with $M$ being the number of LRPs) for each cell/spot ($v_i$) by aggregating ligand signals from neighboring cells/spots via attention-based HGs, and simultaneously guarantees that learned CCC patterns are comparable between different cell-states within TME (Fig. 1f).

### Learning cell modules by self-supervised learning
Our cell module captures local hierarchical relations with genes and histological regions, while integrating global inter-cell/spot associations from spatial location, histology, and gene expression data, allowing us to efficiently capture diverse cell-states within a heterogeneous TME. During the encoding process, we separately learned hierarchical and semantic representations, and meanwhile collaboratively integrated them through contrastive learning to learn high-level representations. Once converged, we applied the concatenation of two representations for further identifying cell-modules (Fig. 1d).

**stKeep for learning local hierarchical representations.** We characterized three different types of entities and their hierarchical relations from SRT data via HG $\mathbf{g}_1 = (\mathbf{V}_1, \mathbf{E}_1, \mathbf{\Lambda}_1, \mathbf{\Gamma}_1, \mathbf{\Phi}_1, \mathbf{\Psi}_1)$, where $\mathbf{V}_1$ and $\mathbf{E}_1$ indicate sets of nodes (i.e., genes, cells/spots, and regions) and edges, and their associated node type mapping function $\mathbf{\Phi}_1 : \mathbf{V}_1 \rightarrow \mathbf{\Lambda}_1$ and edge type mapping function $\mathbf{\Psi}_1 : \mathbf{E}_1 \rightarrow \mathbf{\Gamma}_1$, and also $\mathbf{\Lambda}_1$ and $\mathbf{\Gamma}_1$ denote the sets of entity and relation types. Figure 1c shows an example of two types of relations in $\mathbf{g}_1$: "express" and "belongs to", i.e., a cell/spot expresses gene and belongs to a region. The specific model for graph construction and representation learning is described as follows.

(1) To establish the relations between regions and cells/spots, and encode regions, we (i) defined the histological regions as follows: cells/spots of the same histological type that are separated by immune or other mesenchymal cells may exhibit different cell-states, implying that they belong to different histological regions[18]. Specifically, we utilized antibody colors in the IF staining images to distinguish these regions, while leveraging the spatial distribution of classical marker genes of T (*CD3D*, *CD4*, and *CD8*), myoid (*CD68* and *CD163*), B (*CD19*), and stromal cells (*COL1A1*) to distinguish different histological regions in H&E staining images. These marker genes exhibit consistent expression patterns across a wide range of tissue and disease types[84–86], suggesting the potential application of histological region segmentation methods in analyzing more tumor types. Subsequently, we assigned each cell/spot to its corresponding region by calculating the ratio of the interaction area of the spot/cell with a given region and the spot area[18]; and (ii) employed the One-Hot Encoding method to indicate the specificity of each region.

(2) To construct the relations between genes and cells/spots, and encode genes, with the selected highly variable genes (HVGs)

using Seurat[87] as the reference, we (i) sorted genes for each cell/spot ($v_i$) based on the ratio of its expression value to mean expression, selected the top $m$ genes that best represent its cell-specific state, and then established a connection (i.e., express) between $v_i$ and $m$ genes; and (ii) applied One-Hot Encoding method to encode gene features that are used to characterize gene specificity.

(3)   To encode cells/spots, we employed an autoencoder-based framework[18] to lean low-dimensional features for each cell/spot, which was achieved by maximizing the marginal likelihood function of the observed gene expression data, and then applied these low-dimensional features to characterize cell specificity.

(4)   To learn local hierarchical features for cells/spots, we adopted the following methods:

(i)   since the features of regions, genes, and cells/spots are encoded in different spaces, we mapped them into a common feature space using the following equation:

$$\mathbf{d}_i = f_{\varnothing_i}(\mathbf{l}_i) = \mathrm{relu}\left(\mathbf{W}_{\varnothing_i}\mathbf{l}_i + \mathbf{b}_{\varnothing_i}\right) \tag{1}$$

where $\mathbf{d}_i$ indicates the features mapped by node $v_i$, $\mathbf{W}_{\varnothing_i}$ and $\mathbf{b}_{\varnothing_i}$ represent the parameter to be learned;

(ii)   for each cell/spot $v_i$, we applied two different attention strategies to automatically calculate the contributions of the genes it expresses and the region it belongs to. Specifically, we defined the gene level attention for leveraging gene information as follows:

$$\mathbf{d}_i^{\Omega_2} = \mathrm{relu}\left(\sum_{j\in\mathbf{N}_i^{\Omega_2}} \alpha_{ij}^{\Omega_2}\cdot\mathbf{d}_j\right)$$

$$\alpha_{ij}^{\Omega_2} = \frac{\exp\left(\mathrm{LeakyReLU}\left(\mathbf{a}_{\Omega_2}^T\cdot\left[\mathbf{d}_i||\mathbf{d}_j\right]\right)\right)}{\sum_{m\in\mathbf{N}_i^{\Omega_2}}\exp\left(\mathrm{LeakyReLU}\left(\mathbf{a}_{\Omega_2}^T\cdot\left[\mathbf{d}_i||\mathbf{d}_m\right]\right)\right)} \tag{2}$$

where $\mathbf{d}_j$ and $\mathbf{d}_i$ are the features of gene $v_j$ and cell/spot $v_i$, $\alpha_{ij}^{\Omega_2}$ denotes the weight of $v_j$ on $v_i$, $\mathbf{a}_{\Omega_2}$ is the gene level attention vector, $\mathbf{N}_i^{\Omega_2}$ indicates gene set for characterizing $v_i$, LeakyReLU is the activation function, and $||$ is the concatenation operation. Similarly, we can compute cell/spot features $\mathbf{d}_i^{\Omega_1}$ by aggregating information from regions using the same equations as defined above;

(iii)   after calculating $\mathbf{d}_i^{\Omega_1}$ and $\mathbf{d}_i^{\Omega_2}$, we integrated two types of information using hierarchical level attention to calculate the low-dimensional representations of cells/spots. Firstly, we determined the weights of each type of node using the following equations:

$$\beta_{\Omega_m} = \frac{\exp\left(\mathbf{w}_{\Omega_m}\right)}{\sum_{i=1}^{S}\exp\left(\mathbf{w}_{\Omega_i}\right)}$$

$$\mathbf{w}_{\Omega_m} = \frac{1}{|\mathbf{V}|}\sum_{i\in\mathbf{V}}\mathbf{a}_H^T\cdot\tanh\left(\mathbf{W}_H\mathbf{d}_i^{\Omega_m}+\mathbf{b}_H\right) \tag{3}$$

where $\beta_{\Omega_m}$ indicates the contribution of genes or regions to the low-dimensional features, where $m\in\{1,2\}$, $S=2$, $\mathbf{V}$ is the set of cells/spots, $\mathbf{W}_H$ and $\mathbf{b}_H$ are the parameters to be learned, and tanh is the activation function, and $\mathbf{a}_H$ represents the activation vector of the hierarchical view. Thus, the low-dimensional representations of a cell/spot can be defined as follows:

$$\mathbf{R}_i^1 = \sum_{m=1}^{S}\beta_{\Omega_m}\cdot\mathbf{d}_i^{\Omega_m} \tag{4}$$

**stKeep for learning global semantic representations.** We established three types of semantic graphs from histological images, spatial location, and gene expression data. Specifically, we (i) created the spatial location graph (SLG, $\mathbf{G}^1=(\mathbf{V},\mathbf{E}^1)$) by measuring the Euclidian

distance between cells/spots in the tissue slice; (ii) adopted stMVC to extract visual features from H&E and IF staining histological images and then constructed the histological similarity graph (HSG, $\mathbf{G}^2=(\mathbf{V},\mathbf{E}^2)$); and (iii) constructed transcription similarity graph (TSG, $\mathbf{G}^3=(\mathbf{V},\mathbf{E}^3)$) based on the Cosine similarity of the cell/spot's encodings. Each graph consists of a six-nearest neighbor graph for each cell/spot, and it is worth noting that three graphs have the same nodes but different edges.

Subsequently, we employed two steps to learn global semantic representations as follows:

(i)   for each graph, we adopted a graph attention encoder (GAE) to learn low-dimensional features with the following inputs: an adjacency matrix ($\mathbf{A}^m\in R^{n\times n}$) representing $m$th graph $\mathbf{G}^m$, where $m\in\{1,2,3\}$, and cell encodings. A GAE can be built by stacking multiple multi-head graph attention layers[88]. Specifically, each layer is defined as:

$$\mathbf{h}_i^{(l+1)} = \sigma\left(\frac{1}{Q}\sum_{q=1}^{Q}\sum_{j\in\mathbf{N}_i^L}\alpha_{ij}^q\mathbf{W}^q\mathbf{h}_j^l\right)$$

$$\alpha_{ij}^q = \frac{\exp\left(\mathrm{LeakyReLU}\left((\mathbf{a}^q)^T\left[\mathbf{W}^q\mathbf{h}_i^l||\mathbf{W}^q\mathbf{h}_j^l\right]\right)\right)}{\sum_{o\in\mathbf{N}_i^L}\exp\left(\mathrm{LeakyReLU}\left((\mathbf{a}^q)^T\left[\mathbf{W}^q\mathbf{h}_i^l||\mathbf{W}^q\mathbf{h}_o^l\right]\right)\right)} \tag{5}$$

where $Q$ indicates the number of head attention and the default value is two, $\alpha_{ij}^q$ is normalized attention coefficients computed by the $q$th attention mechanism ($a^q$), $\mathbf{W}^q$ is the corresponding input linear transformation's weight matrix, $\mathbf{N}_i^L$ is the neighborhood of cell ($v_i$) in the graph, $\mathbf{h}_j^l$ is the input feature of node $j$ of the $l$th layer. The embedding for each graph is represented by $\mathbf{P}^m$;

(ii)   inspired by attention-based models emphasizing capturing more critical information to the current task from abundant information[89,90], we adopted the attention mechanism to learn the weight of each graph for the final representations by the following equations:

$$\mathbf{R}_i^2 = \sum_{m=1}^{M}\gamma_i^m\mathbf{p}_i^m$$

$$\gamma_i^m = \frac{\exp\left(\mathbf{a}_m\cdot\mathbf{p}_i^C\right)}{\sum_{o=1}^{M}\exp\left(\mathbf{a}_o\cdot\mathbf{p}_i^C\right)} \tag{6}$$

where $M$ is the number of graphs, $\mathbf{p}_i^C\in R^{3d\times1}$ is the concatenation of all graph-specific representations of cell/spot $v_i$, and $\mathbf{a}_m\in R^{3d\times1}$ is the feature vector of the $m$th graph, describing what kinds of cells/spots will consider the $m$th graph as informative. If $\mathbf{p}_i^C$ and $\mathbf{a}_m$ have a large dot product, indicating cell/spot $v_i$ believes that the $m$th graph is an informative graph, and vice versa.

**stKeep for learning cell modules features by integrating local hierarchical and global semantic representations.** We applied contrastive learning to collaboratively integrate local hierarchical and global semantic representations by supervising each other in the representation space. The equation for hierarchical representations ($\mathbf{R}^1$) under the supervision of semantic representations ($\mathbf{R}^2$) is given by:

$$L_i^H = -\log\frac{\sum_{j\in\mathbf{P}_i}\exp\left(\mathrm{sim}\left(\mathbf{R}_i^2,\mathbf{R}_j^1\right)/\tau\right)}{\sum_{k\in\{\mathbf{P}_i\cup\mathbf{N}_i\}}\exp\left(\mathrm{sim}\left(\mathbf{R}_i^2,\mathbf{R}_k^1\right)/\tau\right)} \tag{7}$$

Similarly, under the supervision of hierarchical representations, the formula for semantic representations is defined as

follows:

$$L_i^S = -\log \frac{\sum_{j \in \mathbf{P}_i} \exp\left(\text{sim}\left(\mathbf{R}_i^1, \mathbf{R}_j^2\right)/\tau\right)}{\sum_{k \in \{\mathbf{P}_i \cup \mathbf{N}_i\}} \exp\left(\text{sim}\left(\mathbf{R}_i^1, \mathbf{R}_k^2\right)/\tau\right)} \quad (8)$$

The overall loss function is:

$$L_{total} = \frac{1}{|\mathbf{V}|} \sum_{i \in \mathbf{V}} \left[\lambda \cdot L_i^H + (1-\lambda) \cdot L_i^S\right] \quad (9)$$

where $\lambda$ is used to balance the importance of the two types of representations, $\text{sim}(\cdot, \cdot)$ represents the cosine similarity between two vectors, $\tau$ denotes the temperature parameter, and $\mathbf{P}_i$ and $\mathbf{N}_i$ indicate the set of positive and negative samples of cell/spot $v_i$. Positive samples for each cell/spot are defined as its spatial nearest neighbors belonging to the same region, while all other cells/spots are considered as negative samples. This strategy encourages cells/spots from the same region to be clustered closely in the latent space, thereby facilitating sub-clustering within the regions while preserving overall cell heterogeneity. The model optimization process stops when $L_{total}$ no longer decreases. Finally, we applied $\mathbf{R} = \mathbf{R}^1 || \mathbf{R}^2$ for further spatial clustering, visualization, and denoising (Fig. 1g). Compared to the simple concatenation of $\mathbf{R}^1$ and $\mathbf{R}^2$ learned separately, even for optimal dimensions, our collaborative learning method is able to generate two representations that efficiently characterize cellular heterogeneity (Supplementary Figs. 15a, b and 16a, b).

### Learning gene modules by contrastive learning

We applied a similar method as $\mathbf{g}_1$ to construct gene-centered HG $\mathbf{g}_2 = (\mathbf{V}_2, \mathbf{E}_2, \mathbf{\Lambda}_2, \mathbf{\Gamma}_2, \mathbf{\Phi}_2, \mathbf{\Psi}_2)$, where $\mathbf{V}_2$ and $\mathbf{E}_2$ indicate sets of nodes (i.e., genes, cells/spots, and cell-states/clusters) and edges, and with their associated node type mapping function $\mathbf{\Phi}_2 : \mathbf{V}_2 \rightarrow \mathbf{\Lambda}_2$ and edge type mapping function $\mathbf{\Psi}_2 : \mathbf{E}_2 \rightarrow \mathbf{\Gamma}_2$, and also $\mathbf{\Lambda}_2$ and $\mathbf{\Gamma}_2$ denote the set of entity and relation types. There are two types of relations in $\mathbf{g}_2$: "expressed by" and "over-expressed by", i.e., a gene is expressed by cells/spots and over-expressed by a cell-state.

To identify gene-modules specific to cell-states (Fig. 1e), we followed these steps:

(1) adopted the cell module to identify different cell-states and their over-expression genes (i.e., SVGs), and treated these SVGs as candidate genes, generating the relations between cell-states and genes;

(2) utilized the same strategies as $\mathbf{g}_1$ to create relations between genes and cells/spots, encode genes, cells/spots, and cell-states (similar to regions);

(3) similar to cell module, mapped encodings of three types of nodes into the common embedding space, leveraged two different types of attention to aggregate information from cells/spots and cell-states, and then combined representations from cells/spots and cell-states via attention to calculate low-dimensional gene representations;

(4) for each gene, regarded its associated genes in PPI and GRN that are co-expressed in the same cells/spots as positive samples, while all other genes as negative samples, and leveraged the contrastive learning model to guarantee that co-relational gene pairs are embedded adjacently in the representation spaces; and

(5) once converged, applied the representations ($\mathbf{G}_k$) for further clustering and visualization to identify gene-modules specific to cell-states (Fig. 1g).

### Learning cell-cell communication by contrastive heterogeneous graph learning

We constructed cell-cell communication networks for each cell by leveraging heterogeneous graphs, where each graph represents a ligand-receptor interaction between a cell (or spot) and its spatial $K$ nearest neighbors (Fig. 1f). For the $kth$ LRP (with ligand $o$ and receptor $p$), the interaction degrees of ligand from its neighbors to the receptors for the central cell/spot ($v_i$) are not available. To address this, we proposed an attention-based method to fuse the ligand information from its neighbors to obtain ligand strength information $LR_{i,k}$, and then calculated the interaction strength for $LRP_{i,k}$ as the product of $LR_{i,k}$ and the expression value of the receptor $p$. The corresponding equations are defined as follows:

$$LRP_{i,k} = LR_{i,k} \cdot x_{i,p}$$
$$LR_{i,k} = \sum_{z \in \mathbf{M}_i} \alpha_{i,z} \cdot x_{z,o}$$
$$\alpha_{i,z} = \frac{\exp\left(\text{LeakyReLU}\left(\mathbf{a}_k^T \cdot \left[x_{i,p} || x_{z,o}\right]\right)\right)}{\sum_{l \in \mathbf{M}_i} \exp\left(\text{LeakyReLU}\left(\mathbf{a}_k^T \cdot \left[x_{i,p} || x_{l,o}\right]\right)\right)} \quad (10)$$

where $\mathbf{M}_i$ represents the neighborhood set of cells/spots around $v_i$, $\alpha_{i,z}$ is the attention coefficient of various cells/spots in the neighborhood, denoting the weight, $\mathbf{a}_k^T$ is the attention vector of the cell/spot for $kth$ LRP, and $x_{i,p}$ and $x_{z,o}$ represent the expression level of receptor $p$ for $v_i$ and ligand $o$ for $v_z$, respectively. This allows for inferring the interaction strength for each LRP where the receptor is expressed by a central cell/spot and the ligand by surrounding cells/spots.

To ensure that learned CCC patterns between different cell-states are comparable, we (i) horizontally concatenated the interaction strengths of all LRPs into a vector $\mathbf{H}_i$ for $v_i$, named as CCC interaction strength; (ii) further extracted the latent features of $\mathbf{H}_i$ using the function $\mathbf{K}_i = \mathbf{W}\mathbf{H}_i$, which represents the cell-state of $v_i$ influenced by its neighboring cells/spots through CCC; (iii) leveraged contrastive learning to make sure that cells/spots in the same cell-states are embedded nearby in the latent feature space. The specific loss function is summarized as follows:

$$L_{LRP-i} = -\log \frac{\sum_{j \in \mathbf{P}_i} \exp\left(\text{sim}\left(\mathbf{k}_i, \mathbf{k}_j\right)/\tau\right)}{\sum_{m \in \{\mathbf{P}_i \cup \mathbf{N}_i\}} \exp\left(\text{sim}(\mathbf{k}_i, \mathbf{k}_m)/\tau\right)} \quad (11)$$

where $\mathbf{P}_i$ and $\mathbf{N}_i$ indicate the set of positive and negative samples for cell/spot $v_i$, which are defined in the same way as in Eq. (9); and (iv) once converged, applied the CCC interaction strength ($\mathbf{H}_i$) to elucidate diverse cancer cell-states within TME (Fig. 1g).

### Datasets and preprocessing

**SRT data and preprocessing.** In our study, we analyzed SRT data for human DLPFC, breast, lung, colorectal, and liver metastasis samples, including gene expression, histology (H&E and IF staining images), and spatial location (see Supplementary Table 1 for details). Specifically, (i) the DLPFC dataset contains 12 slices with varying number of spots ranging from 3460 to 4789[37]; (ii) IDC (Luminal B breast cancer) sample has 4727 spots with gene expression and IF staining image using an anti-human CD3 antibody and DAPI; (iii) BAS1 (Her2⁺ breast cancer) sample has 3798 spots; (iv) TNBC (triple-negative breast cancer) sample contains 1162 spots from a previous research[44]; (v) primary colorectal sample (P1) includes 2,917 spots from a previous study[64]; (vi) two liver metastasis samples (LM1 and LM2) contain 3826 and 3721 spots, respectively[64]. Note that P1 and LM1 samples are from the same patient; (vii) FFPE (Her2⁺ breast cancer) sample contain 2239 spots; and (viii) NSCLC (lung cancer) FFPE sample with 98,002 cells assembled from 30 fields of view (FOV).

We applied the 'vst' method of Seurat[87] to identify the top 3000 HVGs in each gene expression dataset, which were used to comprehensively compare each computational method. In addition, for each dataset, we used the 50-dimensional latent features learned from an autoencoder-based framework as input for stKeep.

**scRNA-seq data**. To clearly understand CCC in different cancer cell-states on the IDC sample, we re-analyzed scRNA-seq data from a previous study[44]. The dataset includes 21,580 cells from seven luminal B patients, which consists of epithelial cells, T cells, and innate lymphoid cells, myeloid cells, stromal cells, B cells, and plasmablasts. For the SRT data of TNBC sample (CID44971), we re-analyzed paired scRNA-seq data of 1627 epithelial cells, including mature luminal, luminal progenitors, myoepithelial cells, cancer basal cells, and cancer cycling cells[44]. To decipher intratumor heterogeneity of primary colorectal and liver metastasis, we analyzed scRNA-seq data of 4069 cells from SMC07 patient[65], including 20 different cell subtypes. Detailed annotation of the different cell types or subtypes of the analyzed datasets is provided in Supplementary Table 4.

### Collection of PPI, GRN, and LRP from public databases

We directly download 3,621,987 PPIs and 3,592,299 GRNs from data source of NicheNet[91] with the link: https://doi.org/10.5281/zenodo.3260758. Moreover, we collected 4,257 LRPs from CellChatDB[92], connectomeDB[93], CellphoneDB[94], and NicheNet[91].

### Clustering and visualization

For each SRT data, we applied stKeep to separately learn representations from cell module and gene module, and further adopted 'FindNeighbors' and 'FindClusters' function with default parameters from the Seurat package[87] to determine k-nearest neighbors (KNNs) for each cell/spot and gene, construct the shared nearest neighbor graph, and identify the cell-modules and gene-modules by the Louvain algorithm. We also utilized the UMAP algorithm to map the low-dimensional features to two-dimensions, visualized the distance of embeddings between different populations by 'Dimplot' function, and visualized the clustering and gene expression patterns at the spatial level by 'SpatialDimPlot' and 'SpatialFeaturePlot' function, respectively.

### Evaluation of the clustering

We applied silhouette width-based measure to evaluate the spatial clustering[38]. The metric measures how similar a cell/spot is to its predicted cluster compared to other clusters, and a higher value denotes that the cell/spot well-assigned to its cluster, which is calculated as follows:

$$\mathrm{SW}(i) = \frac{c(i) - d(i)}{\max\{c(i), d(i)\}} \tag{12}$$

where $d(i)$ and $c(i)$ separately denote the average Euclidean distance of the low-dimensional representations between a cell/spot ($v_i$) and other cells/spots in the same cluster, and $v_i$ to all cells/spots in the nearest cluster to which $v_i$ does not belongs. The average of silhouette width (ASW) as the final metrics to evaluate clustering performance.

### Data denoising and identification of SVGs

We adopted the previous KNN-smoothing algorithm to aggregate information from three nearest cells/spots for each cell/spot to denoise the gene expression data[18]. We identified SVGs from 3000 HVGs among different clusters from the stKeep by 'FindAllMarkers' from Seurat package.

### Statistics and reproducibility

No statistical methods were employed to predefine the sample size. Neither biological nor technical replications were performed on the biological samples outlined in Figs. 3a, g, 5a, 6a, h, and 6m. All data were sourced from the public domain, and no exclusions were made from the analysis. The experiments were not conducted randomly, and the researchers were not blinded to allocation during the experiment and assessment of results. Further details can be found in the Reporting summary file.

### Reporting summary

Further information on research design is available in the Nature Portfolio Reporting Summary linked to this article.

## Data availability

The raw count matrix, histology, and spatial location data for IDC, BAS1, and FFPE (Her2+ breast cancer) samples are publicly available at the 10X Genomics Website (https://support.10xgenomics.com/spatial-gene-expression/datasets), while TNBC sample are available from the Zenodo data repository (https://doi.org/10.5281/zenodo.4739739). The human DLPFC dataset are available from the R package spatialLIBD (http://spatial.libd.org/spatialLIBD/)[37]. Primary colorectal cancer and two liver metastasis samples are available from the website: (http://www.cancerdiversity.asia/scCRLM/). The NSCLC sample are downloaded from website (https://nanostring.com/products/cosmx-spatial-molecular-imager/ffpe-dataset/nsclc-ffpe-dataset/). The scRNA-seq datasets of breast and colorectal cancers are publicly available from Gene Expression Omnibus database with GSE176078 and GSE132465, respectively. The C5: ontology gene sets are available from MSigDB database (https://www.gsea-msigdb.org/gsea/msigdb/human/genesets.jsp?collection=C5). The ligand-receptor interaction databases are located at CellChatDB (http://www.cellchat.org/cellchatdb/), ConnectomeDB (https://db.humanconnectome.org/), CellphoneDB (https://www.cellphonedb.org/), and NicheNet (https://zenodo.org/records/7074291). Source data provided for this paper are available as a source data file and at figshare[95]. Source data are provided with this paper.

## Code availability

stKeep is implemented based on python 3.8.18 and R 4.0.0. Other tools and packages used in the data analysis include: numpy 1.22.4, pandas 2.0.3, scipy 1.8.1, scikit-learn 1.3.2, torch 1.13.0, tqdm 4.51.0, scanpy 1.9.6, Pillow 9.5.0, seaborn 0.11.1, matplotlib 3.7.3, glob2 0.7, anndata 0.9.2, argparse 1.1, json 2.0.9, cv2 (opencv-python) 4.8.1.78, torchvision 0.14.0, STAGATE 1.0.1, Squidpy 1.2.2, stMVC 0.0.2, Seurat v4, ggplot2 3.3.6, igraph 1.3.2, and ggrepel 0.9.1. The codes are available at Zenodo https://zenodo.org/records/10869542[96]. The stKeep tool will be maintained and updated at https://github.com/cmzuo11/stKeep.

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

## Acknowledgements

This work was supported by the National Natural Science Foundation of China (Nos. 12131020, 31930022, T2350003, T2341007 and 1202660 to L.C., Nos. 32300523 and 62132015 to C.Z.), Strategic Priority Research Program of the Chinese Academy of Sciences (No. XDB38040400 to L.C.), Shanghai Sailing Program (No. 22YF1401700 to C.Z.), Fundamental Research Funds for the Central Universities (No. 2232022D-30) and JLU to C.Z., Science and Technology Commission of Shanghai Municipality (No. 23JS1401300 to L.C.), Shanghai Science and Technology Program (No. 20DZ2251400 to C.Z.), open project of BGI-Shenzhen (No. BGIRSZ20210010 to L.C.), R&D project of Pazhou Lab (Huangpu) (No. 2023K0602 to L.C.), and JST Moonshot R&D (No. JPMJMS2021 to L.C.). We thank Drs. Hao Dai, Yijian Zhang, and Jing Zhang for helpful discussions.

## Author contributions

C.Z. conceived and designed the study, implemented the model, performed all the experiments, and wrote the manuscript with feedback from all authors. J.X. analyzed the TNBC sample by scRNA-seq dataset. C.Z. and L.C. supervised the study, and L.C. revised the manuscript. The authors read and approved the final manuscript.

## Competing interests

The authors declare no competing interests.
