## [Peer Review File · Nature Communications]

REVIEWER COMMENTS

Reviewer #1 (Remarks to the Author): Expert in spatial transcriptomics, computational genomics, statistics, and cell-cell communication analysis

This work uses a graph neural network on spatial transcriptomics data to learn representations for cells, genes, and cell-cell communication activities at the same time which are regularized by predetermined histological regions. These representations together with various downstream analysis tools were used for detailed analysis of tumor micro-environments for identifying local tumor domains and the associated gene programs and cell-cell communication activities. The analyses results shown are appealing. However, I have some major concerns about how the histological regions affect the analysis results and the motivation of regularizing representations with contrastive learning with the regions.

1. How strongly are the analysis results dependent on the histological regions? The histological regions are used as elements in both the heterogeneous graph and for the joint learning of local and global features of cells. Due to Eqn. (7), cells in the same histological region will have similar features and are more likely to be clustered into the cluster. Therefore, several clarifications are needed: 1) How different are the segmentation results from simply using the histological regions as segmentation? Does stKeep mostly lead to results as subclustering of histological regions or are there significant number of cell pairs that are placed in one cluster by stKeep but in different histological regions? 2) How robust the results are given different histological regions? 3) What is the performance of the method without using the histological regions?

2. Related to the previous point, the histological regions for the tumor sample was determined using both the histology image and prior knowledge of marker genes. Was there any known marker genes for cortex layers used in the DLPFC example? How well does the method work when there is little prior knowledge when studying a novel tumor type? These can be clarified by performing the same analyses with the histological regions determined only from the imaging.

3. What is the motivation for placing R^1 and R^2 in the same space or two comparable spaces? What is the benefit of this regularization step compared to simply concatenate the separately learned representations? The ideal dimensions for the local hierarchical representation and the global semantic representation are likely different but this step forces the two representations to have the same dimension.

4. In “Learning cell-cell communication by contrastive heterogeneous graph learning”, what is the interpretation of the attention vector “ a ”? For example, let cell 1 be the center cell and cells 2 and 3 be its two neighbors. Does it mean that when cells 1 and 2 express many compatible LR pairs and cell 1 and 3 only has one significant LR pair expressed, cell 2 will have a much higher attention than cell 3? Then since this attention is constant across different LR pairs, will the significant pair between cell 1 and 3 be missed?

5. Also in cell-cell communication, the histological regions were used again for contrastive learning of embeddings describing cell-cell communication. This again forces the cells within the same region to have similar CCC representation and cells in different regions to have different representations which naturally lead to identification of many cluster-level DE CCC programs. However, this could be an artifact and it may be the case that several regions or even the entire sample have some shared CCC programs. In such cases, will stKeep fail to capture such CCC? It should be clarified what type of CCC can be identified using this approach and what biologically significant CCC could be de-emphasized by the computational method.

6. All results presented used Visium data and the H&E images. Can the method be applied to Visium (FFPE) or even other ST technologies. If the method is restricted to Visium and/or H&E images, this should be made clear in the title or abstract.

7. The Github repo could be improved. I was not able to test the codes as the required files by the scripts are missing in the test_data folder. I also had to install opencv-python and torchvision which were missing from the requirements.txt. I suggest the authors to make sure the example on Github is runnable and wrap the tool as a Python package available through PyPI.

Reviewer #2 (Remarks to the Author): Expert in spatial transcriptomics, statistics, bioinformatics, and tissue heterogeneity; co-reviewed with Reviewer #3

In this study, the authors present stKeep, a GNN-based approach to dissect and analyse the tumour microenvironment. The problem is well motivated, since TMEs are complex, constantly changing, and differ between tumour types. The method relies on construction of a heterogeneous graph, using information from histological images and regions, spatial location of cells, gene expressions, and gene-gene relations to identify gene modules, cell modules, and cell-cell communication modules. These modules are further used to detect and differentiate between cell states in TME as well as to identify cell state-specific gene-gene interactions. The study uses SRT, scRNA-seq, PPI, GRN, and LRP data, but the method is specifically applied to various cancer datasets (and one human DLPFC dataset) generated from spot-based SRT technologies, with no more than 5000 spots in any of them. This calls to attention the model's lack of scalability to more than a few thousand cells, which is a major problem.

The method the authors present builds upon their previous work stMVC, which is a multi-view graph collaborative learning approach used to identify TMEs based on only cell module-like information; stKeep goes a step further by utilising gene and CCC modules. Overall, the manuscript is written

well, the proposed solution is innovative, and the method is very relevant to the field of TMEs. Below, I've listed additional questions and comments that I think need to be addressed.

- The datasets assessed with stKeep were all spot-based, with <5000 spots per dataset. How well would stKeep work with image-based SRT datasets, which can have data from up to hundreds of thousands of cells?

- The python code for the stKeep model is available on github and is well-documented, including installation guidelines. The documentation shows an example run on one slice from the DLPFC dataset that highlights runtime for each step of the model. However, there is no mention of the overall memory and runtime requirements for stKeep, especially as the number of cells/spots increases, in the main text. Self-supervised learning algorithms are usually computationally intensive, especially for large datasets. What is the feasibility of its application to larger datasets with hundreds of thousands of cells? As above, this speaks to the scalability of the model.

- The spot data have been referred to as cell data throughout the manuscript (e.g. Fig 1a), which is misleading. There needs to be clarity when describing spot-level data as opposed to single-cell spatial data. How does stKeep address the issue that spot-level SRT data is often a mixture of different cells?

- It's unclear to what extent each graph component helps in the performance of stKeep. The authors could perform an in-silico study to examine the impact of various information sources (e.g. histopathology, PPI, etc.). Was stKeep tested on simulated data for proof of concept?

- Related to the above, how robust is stKeep to the presence of incomplete or even incorrect prior knowledge embedding in the heterogeneous graphs? E.g., the PPI, GRN, LRP, or TF external data that is described in Fig 1c.

- It is unclear to what extent stKeep is applicable to datasets containing multiple SRT samples. If stKeep can be run with SRT from multiple samples, to what extent is it robust to technical (batch) artefacts between SRT samples?

- Some parts of the Results section are difficult to navigate, since the authors have used points within points, which is very confusing and obstructs readability. E.g., under Results section "stKeep contributes to identifying biologically meaningful cell-modules and gene-modules", the reported outputs on Pg 8 Lns 15-31 are in the format: "...as an example, we conducted further analysis and

observed that (1) two gene-modules are identified. Specifically, i) TP63-regulated genes are related to keratinocyte...”. These sections need to be restructured to improve content legibility.

- The authors have used stKeep on different datasets, but the outputs are scattered across the manuscript, making a comparison of how well it works on different datasets difficult to discern. A supplementary table summarising the clustering outputs of stKeep from different datasets would be helpful – the table contents should include columns for number of clusters identified in annotated data, stKeep, and other methods; gene-modules columns showing range of gene-modules and significantly correlated gene pairs identified by stKeep; and CCC-modules column indicating range of CCC-modules found in stKeep clusters.

- The authors describe various cell subtypes and cell-specific markers throughout the manuscript, which is quite confusing. Suppl. Table 1 can be made more comprehensive by including cell type, cell subtype, and cell/cell-state-specific markers information for all datasets analysed in this study. This would be very helpful while navigating the outputs.

- Fig. 2b and Suppl Fig 1a: There is some discrepancy between cluster colours for sample 151507 (please correct the label in Fig. 2b) in these two figures. In Fig. 2b, the stKeep clusters from 151507 match the annotated data, but in Suppl. Fig. 1a, the WM and Layer4 clusters have switched places. Which of these two figures is correct? Or is it that some additional methods were applied to get the clustering shown in Fig. 2b? If so, then please specify that in the figure legend.

- Suppl Fig. 1: Slices 151669–72 contain Layers 3–6 and WM as per the annotated data, but the stKeep and other methods show Layer1, 2 clusters and no WM cluster. Is this the actual output or an issue with cluster colours?

- Pg 6 Ln 23 and Suppl Fig. 3: In point (iv), the authors mention that “gene-modules for slice 151507 display layer-specific patterns in 11 other independent slices.” However, in the data shown in Suppl. Fig. 3 heatmaps, the layer-specificity of the gene-modules does not always match between slices. E.g., the Layer6-specific gene-modules from 151507 slice show a higher expression in WM layers of 7 independent slices (151508–10, 151673–76). Does this mean that the gene-modules that appear as Layer6-specific in slice 151507 are WM-specific in slice 151509? Based on this, the layer-specificity of gene-modules is not reliable between slices.

- Suppl Fig. 5: The Layer1-specific gene AQP4 also shows high expression in the region annotated as WM. Also, the Layer3-specific gene CCK shows higher expression in the region annotated as Layer6 than the red-outlined Layer3. How can this be explained? Perhaps these regions should be tested further for heterogeneity.

- Also, the figure legend looks incomplete.

- Fig. 3e: The right panel shows correlation between gene pairs. Are the gene pairs taken from gene-modules (shown in the red violin plot) significantly correlated to each other, specifically those that are highly correlated? On average, what proportion of the gene pairs from a gene module are significantly correlated?

- Moreover, the correlation y-axis only has positive values. Does this mean that no negative correlation was observed between the gene pairs? Do the gene modules capture negatively correlated genes?

- Fig. 3h and k: The black squares shown in these figures highlight cluster 29 but also contain pieces of other clusters at the edges. In Fig. 3k, the genes associated with CAF and endothelial cells have high expression around the edges of the black square, and the cluster 29 may not be as heterogeneous as it appears based on the gene expression seen in the black square outline. Would it be possible to add an exact outline of cluster 29 to these figures in place of the black box to make it clearer. A similar approach for the other figures showing gene expression in specific clusters might also be useful.

- Fig. 5a: Compared to the annotation, the stKeep cluster 9 has stroma and adipose as well as lymphocyte regions. Has cluster 9 been examined further to ensure its homogeneity as indicated by stKeep output?

- Pg 16 Ln 5–6: The authors mention “then assigned each cell to its corresponding region using established software.” Please specify the software used.

Minor comments:

- The histology and spot images showing gene expression should have a scale bar, since each spot has multiple cells in it.

- Several datasets were evaluated in this study, and it would be good to have them listed in a table – the contents should include dataset name, sample type, number of replicates (if any), SRT technology used, number of spots, number of genes, and publication describing annotated data.

Some of this information is provided in the Methods section, but it would be better presented in a table.

- Please mention the full form of SVG in the main text.

- Fig. 2b and Suppl Figs. 1, 4 – Could the thick black, circular border be replaced with a thin border for the legends representing layers in these figures.

- Pg 7 Ln 3 – Point (iv) should be point (iii).

- Fig. 5c: In the right panel, the marker gene labels are not readable. The label sizes need to be increased, or the marker genes can be mentioned in the figure legend in the order they are shown in the figure.

Reviewer #3 (Remarks to the Author): Expert in spatial transcriptomics, statistics, bioinformatics, and tissue heterogeneity; co-reviewed with Reviewer #2

Reviewer #4 (Remarks to the Author): Expert in cancer genomics, breast cancer, tumour microenvironment, and spatial transcriptomics

The stKeep method by Zuo & Chen describes a new approach for multi-modal analyses of spatial transcriptomics data using heterogeneous graph learning. stKeep first embeds nodes and edges of a unified graph using data modalities from gene expression, spatial location and histological regions. The method first identifies cell-modules by learning hierarchical and global semantic representations in this graph, which are integrated by contrastive learning into a space for spatial

clustering and visualization. stKeep then leverages known gene databases to learn gene-modules which describe gene-regulatory networks and protein-protein interactions. The gene-module analysis extends to identify ligand-receptor pairs in neighboring cells/spots to predict cell signaling interactions. The authors apply stKeep to publicly available datasets generated on the Visium platform, which yield interesting results in model tissue types such as the brain, and in complex tissue types such as breast and other cancer tissues. The code for stKeep appears to be well packaged, documented and readily installable for public use in a user-friendly manner.

Whilst stKeep appears to strongly outperform other methods it is benchmarked against, one main limitation is its primary application to Visium data in the study, which are not single-cell resolution, and 'spots' are not 'cells' as described throughout the manuscript. Visium spots are comprised of multiple cells depending on tissue and pathological region, which makes it somewhat difficult to interpret GRNs, PPIs and LRPs in 'cells' which are rather composed of many cell types. This factor should be discussed in the manuscript. Furthermore, have the authors applied stKeep to other publicly available datasets from higher resolution spatial transcriptomics technologies to address this?

The authors should also clarify other details of their study.

- It is unclear how much of each data modality input contributes to the overall interpretations of the heterogeneous graph and the resulting cell- and gene- modules. The authors should comment on how each of the input data modalities contribute to the resulting cell-modules e.g. how manually-defined histological regions compare to spatial locations, which are both very similar inputs of spot locations.

- On a similar note, the authors should clarify how manually annotated histological regions impact the subsequent downstream learning of cell-modules. Is it unclear how this step is performed via the expression of marker genes for cancer epithelial, stromal and immune cells, as stated in the methods. Considering the authors utilize 10X Visium data, which are not close to single-cell and also suffer from gene drop-out, the expression of these marker genes may not be robust to clearly define accurate histological regions compared to specialized pathology annotations, particularly for heterogeneous breast tumors.

- One concern is that the spatial clusters identified by stKeep (e.g. Fig 2b) map nearly perfect (NMI ≈ 1) to the pre-defined annotations used as input into stKeep. How much of a bias do these manually defined histological regions have on the contrastive learning step when integrating local hierarchical and global semantic representations of the graph? Considering "positive samples are defined as its spatial nearest neighbors belonging to the same region, and all other cells are

considered negative samples”, does the optimization process favor cell-modules which reflect the manually annotated histological regions?

- Considering that cell-states/-modules are used as input for learning gene-modules, a follow-on question is how these manually defined histological regions further impact the learning of gene-modules and subsequent GRNs, PPIs and LRPs?

- Similarly, provided that cell-states/-modules are used in the learning of subsequent gene-modules, the over-expression of gene modules in specific spatial clusters feels highly circular in nature (Fig 2e; p6 line 20-21). Thus, it is not unexpected that gene-pairs will be more closely characterized and correlate better than random gene pairs (Fig 2c; Fig 3e). Can the authors apply additional metrics such as pathway enrichment to show that gene-module signatures are enriched for independent layer specific genes beyond a small number of markers? This feels like the minimal data required to suggest that stKeep identifies ‘biologically meaningful gene-modules’.

- Can the authors comment on why COMMOT (Fig 2g), which is a highly optimized method for computing cell-cell communication using Optimal Transport, produces almost random noise within a cluster compared to stKeep, with Pearson correlation values in cell-pairs ~ 0 ?

- Since the authors compute gene modules, it is unclear why they only used one gene for survival analysis (e.g. TP63 or PTK7; Fig 3f & i). How were these genes from each module selected - was this a ranked approach based on some gene weightings? It appears more appropriate that survival analysis is performed on a signature built from these modules.

- Considering that the authors are studying breast cancers (e.g. in spatial cluster 29 of Fig 3g-k), the term keratinocytes and tumor-specific keratinocytes (TSK) are not appropriate for studying breast tissue and should be reserved for skin & melanoma studies. Furthermore, spatial cluster 29 in the BAS1 Her2+ breast cancer (Fig 3g-k) appears in regions annotated as ‘carcinoma in-situ’ and many of the genes described are markers of myoepithelial cells (KRT5/14). It is possible this region represents contaminating normal breast epithelium? Can the authors should provide pathology annotations and a high-resolution insert of this region to confirm this?

- Breast myoepithelial cells make up the outer layer of the breast epithelium and are thought of the physical barrier which break down as breast cancers transition from in-situ to invasive stages. In Fig 5d, it is not expected to observe no enrichment of myoepithelial-1 or -2 signals in normal ductal or carcinoma in-situ regions, where they are known to be by histology. Their enrichment only in invasive carcinoma regions seems like an artefact and are likely due to their similar gene expression profiles to basal-like cancer cells. Interactions such as COMP -> ITGAV, CD47 etc. seem to be

mostly enriched in invasive cancer regions (bottom left tissue; Fig 5i) and are likely all interactions of basal-like cancer cells in the TNBC sample used. The authors should revise the manuscript to reflect this or show that their CCC findings are indeed restricted to myoepithelial cells rather than cancer cells.

- Can the authors clarify why all significance values in the paper result in $2.22e-16$?

Reviewer #1

Comment #1: This work uses a graph neural network on spatial transcriptomics data to learn representations for cells, genes, and cell-cell communication activities at the same time which are regularized by predetermined histological regions. These representations together with various downstream analysis tools were used for detailed analysis of tumor micro-environments for identifying local tumor domains and the associated gene programs and cell-cell communication activities. The analyses results shown are appealing. However, I have some major concerns about how the histological regions affect the analysis results and the motivation of regularizing representations with contrastive learning with the regions.

Response: Yes, we thank the reviewer for the comment and encouragement.

Comment #2: How strongly are the analysis results dependent on the histological regions? The histological regions are used as elements in both the heterogeneous graph and for the joint learning of local and global features of cells. Due to Eqn. (7), cells in the same histological region will have similar features and are more likely to be clustered into the cluster. Therefore, several clarifications are needed: 1) How different are the segmentation results from simply using the histological regions as segmentation? Does stKeep mostly lead to results as subclustering of histological regions or are there significant number of cell pairs that are placed in one cluster by stKeep but in different histological regions? 2) How robust the results are given different histological regions? 3) What is the performance of the method without using the histological regions?

Response: We thank the reviewer for the comment. In light of the comment, we have provided additional clarification regarding the definition of histological regions ¹. We defined histological regions as follows: cells/spots of the same histological type that are separated by immune or other mesenchymal cells may exhibit different cell-states, implying that they belong to different histological regions. Specifically, we used the colors of the antibodies in the immunofluorescence (IF) staining images to distinguish these regions, while leveraging the spatial distribution of classical marker genes of T (*CD3D*, *CD4*, and *CD8A*), myoid (*CD68* and *CD163*), B (*CD19*), and stromal (*COL1A1*) cells to distinguish different histological regions from Hematoxylin and Eosin (H&E) staining images. In summary, we classified the pathology annotations into distinct histological regions based on spatial distribution of immune or stromal cells, rather than defining histological regions solely based on spatial distribution of different cell types. We have included the text as suggested on page 17.

Moreover, for each issue, we have the following replies. Specifically,

1) Compared to traditional pathological annotations, our segmentation method often yields a more detailed breakdown of pathological regions. For instance, in an IDC sample annotated with three different types of tumors, we identified 14 histological regions (see Supplementary Fig. 15a). In our contrastive learning strategy, we consider each cell/spot's spatially nearest neighbors within the same histological regions as positive samples, while treating all other cells/spots as negative samples. This strategy encourages cells/spots from the same region to be clustered closely in the latent space, thereby facilitating sub-clustering within these regions while preserving overall cell heterogeneity. In light of the comment, we have added the related text on page 20.

2) We applied stKeep to the analysis of the two cancer datasets (IDC and BAS1) trained with varying numbers of histological regions. Our results revealed that the majority of clustering results exhibit high consistency, indicating the robustness of stKeep to different histological region inputs.

Clearly, leveraging our defined 14 histological regions in both samples as input, stKeep demonstrates an enhanced ability to detect finer cancer cell-states, e.g., cluster 34 in IDC and cluster 20 in BAS1, as shown in Supplementary Figs.15a and 16a. We have added the text on page 13.

3) In the model training conducted on IDC and BAS1 samples without prior utilization of histological regions, we leveraged spatial nearest neighbors as positive samples, while treating others as negative samples. The analysis results demonstrated that stKeep exhibits comparable performance when compared to models using histological regions as input. However, the inclusion of histological regions enhances the clarity of clustering boundaries, e.g., cluster 10 in IDC and cluster 16 in BAS1. The results are shown in Supplementary Figs.15a and 16a. In light of the comment, we have added the text on page 13.

Supplementary Figure 15. Evaluation of spatial clustering with different numbers of histological regions and dimensional sizes on the IDC sample. a Spatial clustering by stKeep using varying numbers of histological regions. The top panel indicates the varied histological regions, while the

bottom panel displays the corresponding spatial clustering results. In the model training without histological regions, positive samples are determined based on spatial nearest neighbors, while others are considered as negative samples. It is noteworthy that stKeep uses 14 histological regions as input for this analysis. **b** Spatial clustering predictions using the separately learned local hierarchical representations (R^1) and global semantic representations (R^2) with different dimensions (30, 40, and 50), as well as their simple concatenation (Concat (R^1, R^2)). Additionally, we present spatial clustering results by concatenating the optimal dimensions for R^1 and R^2 . Source data are provided as a Source Data file.

Supplementary Figure 16. Evaluation of spatial clustering with varying numbers of histological regions and dimensional sizes on the BAS1 sample. a Spatial clustering by stKeep using different numbers of histological regions. The top panel indicates the varied histological regions, while the bottom panel displays the corresponding spatial clustering results. In the model training without histological regions, positive samples are determined based on spatial nearest neighbors, while others

are considered as negative samples. It is noteworthy that stKeep uses 14 histological regions as input for this analysis. **b** Spatial clustering predictions using the separately learned local hierarchical representations (R^1) and global semantic representations (R^2) with different dimensions (30, 40, and 50), as well as their simple concatenation (Concat (R^1, R^2)). Additionally, we present spatial clustering results by concatenating the optimal dimensions for R^1 and R^2 . Source data are provided as a Source Data file.

- 1 Zuo, C. *et al.* Elucidating tumor heterogeneity from spatially resolved transcriptomics data by multi-view graph collaborative learning. *Nature Communications* **13**, 5962 (2022).

Comment #3: Related to the previous point, the histological regions for the tumor sample was determined using both the histology image and prior knowledge of marker genes. 1) Was there any known marker genes for cortex layers used in the DLPFC example? 2) How well does the method work when there is little prior knowledge when studying a novel tumor type? 3) These can be clarified by performing the same analyses with the histological regions determined only from the imaging.

Response: We thank the reviewer for the comment. We have done the following analyses to reply each issue. Specifically,

1) Yes, we have added some commonly used layer-specific marker genes from a previous study² for the DLPFC sample on page 6. The marker genes are shown in Supplementary Table 2.

2) We have noted a consistent expression pattern of classical marker genes associated with tumor microenvironment (TME) cells, including T, myoid, and stromal cells, across various tissue and disease types³⁻⁵. This observation suggests the potential application of histological region segmentation methods in the analysis of new tumor types. To provide clearer insights into the utilization of marker genes, we have made revisions to the text on page 17.

3) Yes, we further analyzed the IDC and BAS1 samples by utilizing histological annotations from the histological image, and found that stKeep produces similar spatial clustering results to those by using our defined 14 histological regions, but the utilization of our defined histological regions can help us to detect more cancer cell heterogeneity such as cluster 34 in IDC, and cluster 20 in BAS1. The results are shown in Supplementary Figs.15a and 16a (ref to Comment #2). We have added the text on page 13.

Layers	Genes
Layer 1	FABP7, AQP4, RELN
Layer 2	HPCAL1
Layer 3	CARTPT, FREM3, ADCYAP1
Layer 4	PVALB, RORB
Layer 5	TRABD2A, PCP4
Layer 6	KRT17, NTNG2
WM	MOBP, MBP

Supplementary Table 2. Known layer-specific genes for the human DLPFC dataset from a previous study².

- 2 Maynard, K. R. *et al.* Transcriptome-scale spatial gene expression in the human dorsolateral prefrontal cortex. *Nature neuroscience* **24**, 425-436 (2021).
- 3 Cheng, S. *et al.* A pan-cancer single-cell transcriptional atlas of tumor infiltrating myeloid cells. *Cell* **184**, 792-809. e723 (2021).
- 4 Luo, H. *et al.* Pan-cancer single-cell analysis reveals the heterogeneity and plasticity of cancer-associated fibroblasts in the tumor microenvironment. *Nature Communications* **13**, 6619 (2022).
- 5 Zheng, L. *et al.* Pan-cancer single-cell landscape of tumor-infiltrating T cells. *Science* **374**, abe6474 (2021).

Comment #4: 1) What is the motivation for placing R^1 and R^2 in the same space or two comparable spaces? 2) What is the benefit of this regularization step compared to simply concatenate the separately learned representations? 3) The ideal dimensions for the local hierarchical representation and the global semantic representation are likely different but this step forces the two representations to have the same dimension.

Response: We thank the reviewer for the comment. We have done the following analyses for each issue, specifically,

1) We leveraged a contrastive self-supervised learning mechanism to link local hierarchical representations and global semantic representations, reinforcing each other and producing high-level embeddings. In light of the comment, we have added the related text on pages 5 and 17.

2) Compared to the separately learned representations, our method produces two distinct yet semantically related representations. The concatenation of those representations efficiently characterizes cell heterogeneity, as supported by our results in Supplementary Figs.15a-b and 16a-b (ref to Comment #2). In light of the comment, we have added the text on page 21.

3) The different dimensionalities between the two representations may hinder efficient knowledge transfer. Maintaining consistency in dimensions for both representations aligns with the biological assumption that these representations show similarity for the same cell/spot in a low-dimensional space. In addition, spatial clustering predicted by simply concatenating the optimal dimensions for R^1 and R^2 is better than using identical dimensions for both representations, but slightly lags behind our mutually regularized representations. The results are shown in Supplementary Figs.15a-b and 16a-b (ref to Comment #2). In light of the comment, we have included the text on page 21.

Comment #5: In “Learning cell-cell communication by contrastive heterogeneous graph learning”, what is the interpretation of the attention vector “a”? For example, let cell 1 be the center cell and cells 2 and 3 be its two neighbors. Does it mean that when cells 1 and 2 express many compatible LR pairs and cell 1 and 3 only has one significant LR pair expressed, cell 2 will have a much higher attention than cell 3? Then since this attention is constant across different LR pairs, will the significant pair between cell 1 and 3 be missed?

Response: We apologize for the confusion arising from the incorrect formula description. In our CCC model, the attention vector ‘a’ is computed for each LR pair, instead of using a constant vector for all LR pairs. This allows the CCC model to infer the interaction strength for each LR pair where the receptor is expressed by a central cell (or spot) and the ligand by surrounding cells (or spots). In light of the comment, we have revised the text on page 22.

Comment #6: Also in cell-cell communication, the histological regions were used again for contrastive learning of embeddings describing cell-cell communication. This again forces the cells within the same region to have similar CCC representation and cells in different regions to have different representations which naturally lead to identification of many cluster-level DE CCC programs. However, this could be an artifact and it may be the case that several regions or even the entire sample have some shared CCC programs. In such cases, will stKeep fail to capture such CCC? It should be clarified what type of CCC can be identified using this approach and what biologically significant CCC could be de-emphasized by the computational method.

Response: That's a good point. We have introduced a Gini index (GI)-based metric to assess the degree of inequality among different clusters in the distribution of CCC inferred by stKeep. Specifically, for each ligand receptor pair (LRP), we computed the average interaction strength in each cluster, and then calculated GI for each LRP. This allowed us to evaluate the inequality level of CCC across different clusters. To provide a comprehensive comparison, we applied the statistical measure to assess the distribution of various entities, including the corresponding used ligands and receptors, housekeeping genes, and known layer-specific genes, and all expressed ligands and receptors, on 12 slices of the human DLPFC dataset. In summary, we found that (1) the GI score of CCC is marginally lower than the higher score for the relevant ligand and receptor within each LRP, and yet slightly surpasses the scores of the individual ligand and receptor. This trend agrees with the biological principle that CCC activation depends on the simultaneous expression of ligand and receptor for each LRP; (2) the GI distribution of the used ligands and receptors is the same as that of all expressed ligands and receptors; and (3) the GI distribution of CCC falls between that of housekeeping genes and that of layer-specific genes, suggesting that CCC exhibits both cluster-specificity and commonality. On average, approximately 31% of the LRPs display a GI score exceeding the mean value of layer-specific genes, while 15% exhibit a GI score below the mean value of housekeeping genes. Overall, these results indicated that stKeep is able to identify cluster-specific and shared CCC patterns. The results for the comment are shown in Supplementary Fig.5a-e. In light of the comment, we have added the text on page 7.

Supplementary Figure 5. stKeep is able to identify the layer specific and shared CCC. **a** Boxplot displaying the GI score of interaction strengths inferred by stKeep on 12 slices of human DLPFC dataset. The GI score is presented for expression levels of the used ligands and receptors, maximum value of ligand and receptor for each LRP, all expressed ligands and receptors, housekeeping genes⁶, and known layer-specific genes (see Supplementary Table 2) as a comparison. For each boxplot, the center line, box limits and whiskers separately indicate the median, upper and lower quartiles and $1.5 \times$ interquartile range. **b** Manual annotation of six layers and WM on 12 slices of the human DLPFC dataset². **c** Spatial expression of ligand *RELN* and receptor *ITGB1*, and their interaction strength on 12 slices

of the human DLPFC dataset. **d** Spatial expression of ligand *PENK* and receptor *ADRA2A*, and their interaction strength on 12 slices of the human DLPFC dataset. **e** Spatial expression of ligand *CALMI* and receptor *PTPRA*, and their interaction strength on 12 slices of the human DLPFC dataset. Source data are provided as a Source Data file.

- 2 Maynard, K. R. *et al.* Transcriptome-scale spatial gene expression in the human dorsolateral prefrontal cortex. *Nature neuroscience* **24**, 425-436 (2021).
- 6 Eisenberg, E. & Levanon, E. Y. Human housekeeping genes, revisited. *Trends Genet* **29**, 569-574 (2013).

Comment #7: All results presented used Visium data and the H&E images. Can the method be applied to Visium (FFPE) or even other ST technologies. If the method is restricted to Visium and/or H&E images, this should be made clear in the title or abstract.

Response: We thank the reviewer for the comment. Our application of stKeep to Visium (FFPE) sample (Her2+ breast cancer) yielded conclusions similar to those from Visium fresh frozen samples, demonstrating the enhanced detection of cell-states, associated gene-modules and CCCs. Moreover, we have shown the versatility of stKeep in analyzing a non-small cell lung cancer (NSCLC) FFPE sample (~100K cells) by NanoString (a single-cell resolution technology). The results showed that stKeep enables to detect more cancer cell-states, particularly in the interface region between tumors and stromal cells, and elucidate internal gene regulatory and CCC mechanisms. The results for Visium (FFPE) and NanoString (FFPE) are presented separately in Supplementary Figs.18a-g and 19a-g. In light of the comment, we have introduced the NanoString technology, and incorporated the corresponding analysis result on pages 2, 4, 14, 23, 24, and 25. In addition, we have revised the text as suggested on pages 4, 19, and 23, emphasizing stKeep's capability to process spot- and single-cell-level SRT data, along with H&E and HF staining images.

Supplementary Figure 18. Method comparisons on the FFPE (Her2⁺ breast cancer) sample. **a** H&E tissue plot displaying manual annotation of 19 tumor regions from the 10X Genomics website (<https://www.10xgenomics.com/resources/datasets/human-breast-cancer-ductal-carcinoma-in-situ-invasive-carcinoma-ffpe-1-standard-1-3-0>), excluding 279 necrotic cells from our analysis. **b** Spatial clustering predicted by Squidpy, STAGATE, stMVC, and stKeep. **c** The identified gene-modules in seven spatial cancer clusters. UMAP visualization presented on the left panel, while the right panel exhibits mean gene expression of gene-modules across different clusters. **d** The identified gene-module for spatial cluster 15 by stKeep, with regulator genes in red and target genes in blue. **e** Spatial expression of genes over-expressed in spatial cluster 15. **f** Scatter plot displaying the ligands (black) and receptors (red) over-expressed in spatial cluster 15. **g** Spatial expression of ligand *MIF* and receptor *CD44*, along with their corresponding CCC interaction strengths. Source data are provided as a Source Data file.

Supplementary Figure 19. Method comparisons on the NSCLC (FFPE) sample by NanoString technology. **a** IF staining image showing manual annotation of 17 tumor regions (red outline), and niches from the NanoString website (<https://nanosttring.com/products/cosmx-spatial-molecular-imager/ffpe-dataset/nsclc-ffpe-dataset/>). The intensity of DAPI (cell nuclei), PanCK (tumor cells), CD45 (leucocytes), and CD3 (T cells) is indicated by blue, green, yellow, and red, respectively. **b** Spatial clustering predicted by Seurat and stKeep. **c** Spatial expression of *KRT19* for epithelial cells (left panel). Spatial clusters within tumor regions (right panel). **d** Violin plot displaying expression levels of *SLC2A1*, *COL3A1*, *SPINK1*, *HSPA1B*, *LIF*, *CD55*, and *MSMB* across seven clusters. Each color indicates one cluster. **e** The identified gene-module for spatial cluster 7. The regulator and its target genes are colored in red and blue, respectively. Here, we displayed the over-expressed genes in spatial cluster 7 compared to other clusters, with a \log_2fc greater than 3. **f** Scatter plot displaying the over-expressed genes in spatial cluster 7. **g** Spatial expression of ligand *COL3A1* and receptor *DDR1*, along with their corresponding CCC interaction strengths. The black outline in **a**, **b**, **c**, and **g** indicates tumor regions, consistent with the red color in **a**. Source data are provided as a Source Data file.

Comment #8: The Github repo could be improved. I was not able to test the codes as the required files by the scripts are missing in the test_data folder. I also had to install opencv-python and torchvision which were missing from the requirements.txt. I suggest the authors to make sure the example on Github is runnable and wrap the tool as a Python package available through PyPI.

Response: We are sorry for the oversight. In light of the comment, we have uploaded the missed files into 'test_data' folder, updated code on GitHub accordingly, and wrapped the tool as a Python package.

Now users can easily install it using the command: ‘pip install stKeep’. Our team members have tested the sample case available on GitHub to ensure the code runs smoothly.

Reviewer #2

Comment #1: In this study, the authors present stKeep, a GNN-based approach to dissect and analyse the tumour microenvironment. The problem is well motivated, since TMEs are complex, constantly changing, and differ between tumour types. The method relies on construction of a heterogeneous graph, using information from histological images and regions, spatial location of cells, gene expressions, and gene-gene relations to identify gene modules, cell modules, and cell-cell communication modules. These modules are further used to detect and differentiate between cell states in TME as well as to identify cell state-specific gene-gene interactions. The study uses SRT, scRNA-seq, PPI, GRN, and LRP data, but the method is specifically applied to various cancer datasets (and one human DLPFC dataset) generated from spot-based SRT technologies, with no more than 5000 spots in any of them. This calls to attention the model’s lack of scalability to more than a few thousand cells, which is a major problem.

Response: Yes, we thank the reviewer for the comment. In light of the scalability, we have performed additional experiments to support the capability of stKeep in processing a large-scale dataset, a non-small-cell lung cancer (NSCLC) sample with ~100K cells generated by NanoString technology. The results showed that stKeep enables the detection of more cancer cell-states, particularly in the interface region between tumors and stromal cells, and elucidates internal gene regulatory and CCC mechanisms. The results are shown in Supplementary Fig.19a-g. In light of the comment, we have added the related text on pages 2, 4, 14, 23, 24, and 25.

Supplementary Figure 19. Method comparisons on the NSCLC (FFPE) sample by NanoString technology. **a** IF staining image showing manual annotation of 17 tumor regions (red outline), and niches from the NanoString website (<https://nanosttring.com/products/cosmx-spatial-molecular-imager/ffpe-dataset/nsclc-ffpe-dataset/>). The intensity of DAPI (cell nuclei), PanCK (tumor cells), CD45 (leucocytes), and CD3 (T cells) is indicated by blue, green, yellow, and red, respectively. **b** Spatial clustering predicted by Seurat and stKeep. **c** Spatial expression of *KRT19* for epithelial cells (left panel). Spatial clusters within tumor regions (right panel). **d** Violin plot displaying expression levels of *SLC2A1*, *COL3A1*, *SPINK1*, *HSPA1B*, *LIF*, *CD55*, and *MSMB* across seven clusters. Each color indicates one cluster. **e** The identified gene-module for spatial cluster 7. The regulator and its target genes are colored in red and blue, respectively. Here, we displayed the over-expressed genes in spatial cluster 7 compared to other clusters, with a $\log_2 fc$ greater than 3. **f** Scatter plot displaying the over-expressed genes in spatial cluster 7. **g** Spatial expression of ligand *COL3A1* and receptor *DDR1*, along with their corresponding CCC interaction strengths. The black outline in **a**, **b**, **c**, and **g** indicates tumor regions, consistent with the red color in **a**. Source data are provided as a Source Data file.

Comment #2: The method the authors present builds upon their previous work stMVC, which is a multi-view graph collaborative learning approach used to identify TMEs based on only cell module-like information; stKeep goes a step further by utilising gene and CCC modules. Overall, the manuscript is written well, the proposed solution is innovative, and the method is very relevant to the field of TMEs. Below, I've listed additional questions and comments that I think need to be addressed.

Response: Yes, we thank the reviewer for the comment and encouragement.

Comment #3: The datasets assessed with stKeep were all spot-based, with <5000 spots per dataset. How well would stKeep work with image-based SRT datasets, which can have data from up to hundreds of thousands of cells?

Response: We have applied stKeep in analyzing an NSCLC sample with ~100K cells generated by NanoString (an image-based single-cell resolution technology). Based on analysis results, we found that stKeep can elucidate finer cell-states within TME, along with their regulatory and CCC mechanisms. The results are shown in Supplementary Fig.19a-g (ref to Comment #1). In light of the comment, we have added the related text on pages 2, 4, 14, 23, 24, and 25.

Comment #4: The python code for the stKeep model is available on github and is well-documented, including installation guidelines. The documentation shows an example run on one slice from the DLPFC dataset that highlights runtime for each step of the model. However, there is no mention of the overall memory and runtime requirements for stKeep, especially as the number of cells/spots increases, in the main text. Self-supervised learning algorithms are usually computationally intensive, especially for large datasets. What is the feasibility of its application to larger datasets with hundreds of thousands of cells? As above, this speaks to the scalability of the model.

Response: We have conducted additional experiments to check the scalability of stKeep. Our experiments indicated that stKeep is fast, and it takes 24 min and 13GB of memory to process the SRT dataset with 17K cells. In particular, the running time is approximately linearly proportional to the number of input cells (see Supplementary Fig.20), which is considered as an advantage of stKeep for processing a bigger dataset. Moreover, we extended our evaluation by processing an NSCLC sample assembled from 30 fields of view (FOV) with ~100K cells generated by NanoString technology. Specifically, we (1) trained the stKeep model by sequentially inputting SRT data from each FOV, and then utilized a batch size of 13K to subsample cells from 30 FOV for training the model; and (2) adopted the strategy detailed in (1) iteratively to train the model until convergence. The whole training process in the NSCLC sample takes about 100 min and 6GB of memory. The results are shown in Supplementary Fig. 19a-g (ref to Comment #1). In light of the comment, we have included the text on pages 14, and 15.

Supplementary Figure 20. Comparison of running time and memory usage for the training of stKeep model on the different numbers of cells by subsampling from NSCLC sample. The experiments were tested on a GPU server with two NVIDIA Tesla V100 GPUs addressing 64GB. Source data are provided as a Source Data file.

Comment #5: The spot data have been referred to as cell data throughout the manuscript (e.g., Fig 1a), which is misleading. There needs to be clarity when describing spot-level data as opposed to single-cell spatial data. How does stKeep address the issue that spot-level SRT data is often a mixture of different cells?

Response: Yes, we have revised the related figure and description regarding cell and spot as suggested throughout the manuscript to make it more clear, as suggested. Moreover, we adopted the following strategies to interpret results of cell-modules, gene-modules, and CCC modules from spot-level SRT data using single-cell RNA-seq (scRNA-seq) data. Specifically, (i) detecting key players within TME via CCC: Utilizing cell type-specific ligands and receptors from scRNA-seq, stKeep quantifies the involvement of different cell populations in various cancer cell-states (or cell-modules), pinpointing key cell types; (ii) identifying cell-state specific gene-modules: Gene-modules might contain genes expressed across multiple cell types, however, the identification of gene-gene relations from the known PPI and GRN database knowledge ensures its accuracy. stKeep utilizes scRNA-seq data to infer where co-related gene pairs might be co-expressed; and (iii) predicting crucial mechanisms in disease development: Mapping cell subtypes from scRNA-seq over different spatial cancer regions, stKeep combines cell subtype-specific over-expressed genes and gene-modules with the inferred CCC from SRT data to uncover regulatory mechanisms relevant to disease progression. In light of the comment, we have added the related text on pages 14 and 15.

Comment #6: It's unclear to what extent each graph component helps in the performance of stKeep. The authors could perform an in-silico study to examine the impact of various information sources (e.g., histopathology, PPI, etc.). Was stKeep tested on simulated data for proof of concept?

Response: That's a good point. We have conducted extensive experiments to assess the impact of each graph component on the performance of cell-modules and gene-modules using 12 slices from human DLPFC dataset. Specifically, (1) for each graph or component within the cell module, we evaluated its contribution to the identification of spatial clustering by removing the graph, and measured its impact using the average silhouette width (ASW). The lower the ASW, the higher the contribution of the component. The comparison results revealed the following order of contribution, from most to least: spatial location graph (SLG), cell-region graph (CRG), histological similarity graph (HSG), transcriptomics similarity graph (TSG), and cell-gene graph (CGG). Notably, while the removal of CGG results in tighter clustering, it fails to accurately detect boundaries in finer structures (e.g., cluster 1 in slice 151669). In line with the findings from the DLPFC dataset, CGG demonstrates an important role in accurately detecting boundaries within cancer samples, notably highlighting cluster 33 in IDC and cluster 17 in BAS1, which is visually marked by black outlines. These results confirmed that our cell-module's structure is well designed and it enables to detect cell heterogeneity from different histological regions; and (2) for each graph or component within the gene module, we assessed its contribution to the learning of gene embeddings by removing the graph, and measured this influence by calculating the distance between gene pairs identified from gene-modules using stKeep. The shorter the distance, the higher the contribution of the component. The comparison results showed the order of

contribution, from most to least: gene-cell state graph (GSG), gene-cell graph (GCG), PPI, and GRN, where PPI and GRN have a comparable performance. The results are shown in Supplementary Figs. 13a-c, and 14a. In light of the comment, we have included the text on page 13.

Supplementary Figure 13. Ablation analysis of graph component within the cell module. **a** Violin plot showing clustering score ASW used to evaluate the contribution of each graph within the cell module on 12 human DLPFC slices. Specifically, after removing each graph, we predicted the spatial clustering based on the learned low-dimensional features. The lower the ASW, the higher the contribution of the graph. **b** Bar plot showing clustering score ASW used to evaluate the contribution of each graph within the cell module on two cancer samples (IDC and BAS1). **c** Spatial clustering predicted by the low-dimensional features by removing each graph from stKeep on slice 151669, IDC, and BAS1 samples, where we also provide stKeep as a comparison. Note that spatial location graph, cell-region graph, histological similarity graph, transcriptomics similarity graph, and cell-gene graph

are abbreviated as SLG, CRG, HSG, TSG, and CGG, respectively. Source data are provided as a Source Data file.

Supplementary Figure 14. Evaluation of the gene and CCC modules in stKeep on 12 human DLPFC slices. **a** Violin plot showing distance of gene-pairs (identified by stKeep) to evaluate the contribution of each graph within the gene module. Specifically, after removing each graph, we calculated the distance of the gene-pairs based on the learned gene embedding. The shorter distance, the higher contribution of the graph. Note that gene-cell state graph and gene-cell graph are abbreviated as GSG and GCG, respectively. **b** Violin plot showing the false positive rate (FPR) of the learned gene embedding trained at different proportions of incomplete gene-gene interactions (PPI and GRN). **c** Violin plot showing the recover edge rate (RER) for the learned gene embedding trained at different scales of incomplete gene-gene interactions (PPI and GRN). **d** Line plot showing FPR of the learned gene embedding trained at different proportions of incorrect gene-gene interactions (PPI and GRN). Each color indicates a slice. **e** Violin plot showing Pearson correlation of the inferred CCC between spots within a cluster, inferred for 12 slices with different proportion LRPs removed. Source data are provided as a Source Data file.

Comment #7: Related to the above, how robust is stKeep to the presence of incomplete or even incorrect prior knowledge embedding in the heterogeneous graphs? E.g., the PPI, GRN, LRP, or TF external data that is described in Fig 1c.

Response: We thank the reviewer for the comment. In light of the comment, we evaluated the robustness of stKeep by computing the false positive rate (FPR) of the predicted gene-gene relations (PPI and GRN) for 12 slices in the human DLPFC dataset. We assessed the impact of incomplete or incorrect gene-gene interactions by randomly removing or adding gene-gene interactions (ranging from 10% to 90%) from the initial gene-gene relations as inputs for the gene module. We considered the identified gene-gene relations from the initial complete set as the ground truth, and calculated FPR. Notably, stKeep demonstrates robustness, showcasing an FPR of zero for incomplete graphs and around 0.2% for incorrect prior interactions. Moreover, we introduced the recover edge rate (RER) to measure the proportion of the removed gene-gene interactions that can be recovered from gene-modules trained using incomplete prior graphs at different scales. On average, stKeep is able to recover 23% of the gene-gene relations. In addition, we calculated the Pearson correlation of LRP between spots within each cluster by CCC modules trained on varying proportions of incomplete LRPs. The results consistently demonstrated the ability of stKeep to produce similar outcomes across different proportions of prior LRPs. As a suggestion, users are advised to input confident LRP to infer their interaction strength. The results are shown in Supplementary Fig.14b-e (ref to Comment #6). In light of the comment, we have included the text on page 13.

Comment #8: It is unclear to what extent stKeep is applicable to datasets containing multiple SRT samples. If stKeep can be run with SRT from multiple samples, to what extent is it robust to technical (batch) artefacts between SRT samples?

Response: We thank the reviewer for the comment. In this work, we focused on elucidating the TME of intra-tumoral heterogeneity rather than inter-tumoral heterogeneity. Biological variations (e.g., gender, age, medical treatment, and disease status) as well as technique challenges (e.g., batch effects in sample preparation or sequencing) create complexities when attempting to simultaneously analyze intra- and inter-tumor heterogeneity within a unified framework. To address this issue, we propose to combine stKeep with SRT integrative tools⁷⁻⁹ to provide a more comprehensive understanding of both intra- and inter-tumor heterogeneity, leveraging their strengths in analyzing different aspects of tumor complexity across diverse samples. In light of the comment, we have added the text in discussion on page 15.

7 Zhou, X., Dong, K. & Zhang, S. Integrating spatial transcriptomics data across different conditions, technologies and developmental stages. *Nature Computational Science*, 1-13 (2023).

8 Xia, C.-R., Cao, Z.-J., Tu, X.-M. & Gao, G. Spatial-linked alignment tool (SLAT) for aligning heterogeneous slices. *Nat Commun* **14**, 7236 (2023).

9 Wang, G. *et al.* Construction of a 3D whole organism spatial atlas by joint modelling of multiple slices with deep neural networks. *Nature Machine Intelligence*, 1-14 (2023).

Comment #9: Some parts of the Results section are difficult to navigate, since the authors have used points within points, which is very confusing and obstructs readability. E.g., under Results section “stKeep contributes to identifying biologically meaningful cell-modules and gene-modules”, the

reported outputs on Pg 8 Lns 15-31 are in the format: "...as an example, we conducted further analysis and observed that (1) two gene-modules are identified. Specifically, i) *TP63*-regulated genes are related to keratinocyte...". These sections need to be restructured to improve content legibility.

Response: Yes, we have revised the text as suggested to make it more clear on pages 8 and 9 (in the revised version).

Comment #10: The authors have used stKeep on different datasets, but the outputs are scattered across the manuscript, making a comparison of how well it works on different datasets difficult to discern. A supplementary table summarising the clustering outputs of stKeep from different datasets would be helpful – the table contents should include columns for number of clusters identified in annotated data, stKeep, and other methods; gene-modules columns showing range of gene-modules and significantly correlated gene pairs identified by stKeep; and CCC-modules column indicating range of CCC-modules found in stKeep clusters.

Response: Yes, we have added one table as suggested to present our detailed results generated by three different modules in Supplementary Table 3. In light of the comment, we have cited the table throughout the manuscript.

Technologies	Sample	Cell modules						Gene modules			CCC modules
		Annotation	Squidpy	STAGTE	stMVC	Seurat	stKeep	#modules / clusters	#gene pairs	#significant gene pairs	#LRPs
Visium (fresh frozen)	151507	7	7	7	7	NA	7	7 / 7	12023	4934	2580
	151508	7	7	7	7	NA	7	7 / 7	14578	6474	2512
	151509	7	7	7	7	NA	7	7 / 7	11179	4152	2605
	151510	7	7	7	7	NA	7	7 / 7	9645	4287	2569
	151669	5	5	5	5	NA	5	5 / 5	11864	5911	2522
	151670	5	5	5	5	NA	5	5 / 5	12834	6046	2463
	151671	5	5	5	5	NA	5	5 / 5	15996	7256	2652
	151672	5	5	5	5	NA	5	5 / 5	15988	8108	2580
	151673	7	7	7	7	NA	7	7 / 7	14379	3331	2656
	151674	7	7	7	7	NA	7	7 / 7	14917	2150	2877
	151675	7	7	7	7	NA	7	7 / 7	13309	3463	2614
	151676	7	7	7	7	NA	7	7 / 7	12651	2953	2585
	IDC	14	35	35	35	NA	35	32 / 22	3454	2450	2681
	BAS1	14	35	35	35	NA	35	29 / 23	2694	1965	2994
	TNBC	16	22	22	22	NA	22	NA	NA	NA	2643
	P1	3	9	9	9	NA	9	NA	NA	NA	2702
LM1	2	6	6	6	NA	6	NA	NA	NA	3190	
LM2	2	NA	NA	NA	NA	4	NA	NA	NA	2874	
Visium (FFPE)	FFPE (Her2+ BRCA)	19	16	16	16	NA	16	7 / 7	3630	3397	2910
NanoString (FFPE)	NSCLC (FFPE)	18	NA	NA	NA	18	18	8 / 7	5833	3391	828
scRNA-seq	CID44971	5	NA	NA	NA	10	NA	16 / 10	53149	50088	NA

Supplementary Table 3. Summary of all analyzed datasets in this study. NA indicates that there is no corresponding analysis result.

Comment #11: The authors describe various cell subtypes and cell-specific markers throughout the manuscript, which is quite confusing. Suppl. Table 1 can be made more comprehensive by including cell type, cell subtype, and cell/cell-state-specific markers information for all datasets analysed in this study. This would be very helpful while navigating the outputs.

Response: Yes, we agree. We have added the comprehensive information, including cell types or subtypes and their associated marker genes for all analyzed datasets in Supplementary Table 4 (in the revised version). In light of the comment, we have cited the table on page 23.

Tumor type	Sample	Cell types	Cell subtypes/states	#Cells	Marker genes	Reference
Breast cancer	Luminal B (scRNA-seq)	Epithelial cells	Cancer epithelial	6,223	EPCAM, KRT19	10
			Luminal progenitors	125	ALDH1A3	
			Myoepithelial	109	KRT14	
			Mature luminal	84	FOXA1	
		Lymphocytes	T cells CD8+	2,736	CD8	
			T cells CD4+	5,017	CD4	
			Cycling T-cells	177	CD3D, MKI67	
			NK	525	AREG	
			NKT	284	FCGR3A	
		Myeloid cells	Monocyte	268	FCGR3A, SI100A9	
			Macrophage	698	EGR1, CXCL10	
			Cycling_myeloid	51	MKI67, CD68	
			DCs	226	LAMP3	
		B cells	B cells memory	267	MS4A1	
		Stromal cells	CAFs MSC iCAF-like	550	KLF4, LEPR	
			CAFs myCAF-like	869	FAP, COL1A1	
			PVL immature	485	ALDH1A1	
			PVL differentiated	1,028	MYH11	
			Cycling PVL	20	ACTA2, MKI67	
			Endothelial-ACKR1	703	ACKR1	
Endothelial-RGS5	229		RGS5			
Endothelial-CXCL12	425		CXCL12			
Endothelial lymphatic-LYVE1	30	LYVE1				

Colorectal cancer		Plasma cells	Plasmablasts	451	JCHAIN	11	
		Total	24	21,580	32		
	TNBC (CID44971, scRNA-seq)	Epithelial cells	Mature luminal		169		FOXA1
			Luminal progenitors		442		ALDH1A3
			Myoepithelial cells		124		KRT14
			Cancer basal cells		646		VIM
			Cancer cycling cells		246		MKI67
	Total	5	1,627	5			
	TNBC (CID44971, scRNA-seq)	Epithelial cells	Cancer epithelial		892		EPCAM, KRT19
			Normal epithelial		735		LTF, KRT8
		Stromal cells	Endothelial		217		ENG, VWF
			CAFs		582		DCN, COL1A2
			PVL		91		ALDH1A1
		B cells	B cells		369		MS4A1
		T cells	T cells		4,366		CD3D
		Myeloid	Myeloid		684		CD68
		Plasma cells	Plasmablasts		48		JCHAIN
	Total	9	7,984	13			
	Colorectal cancer	SMC07 (scRNA-seq)	Tumor cells	CMS1	8		CASP7, CCND1
CMS2				493	ASCL2		
CMS3				40	TFF3, SLC26A3		
Stomal cells			Stomal cells	209	APOE		
Plasma cells			IgA+ Plasma	106	IGHA		
			IgG+ Plasma	415	IGHG		
B cells			CD19+CD20+ B	282	CD19, MS4A1, IGHM		
Lymphocytes			CD4+ T cells	934	CD4, IL7R		
			CD8+ T cells	486	CD8		
			Regulatory T cells	264	IL2RA, FOXP3, IL4R		
			NK cells	89	FCGR3A, KLRD1		
	T helper 17 cells	146	IL7A, IL17F, IL22				

			Gamma delta T cells	160	TRGC1, TRGC2, TRDC		
			T follicular helper cells	80	MAF, CXCL13, CXCR5, PDCD1		
		Myeloid cells	SPP1+	140	SPP1		
			Pro-inflammatory	72	IL1B, IL6, S100A8, S100A9		
			Proliferation	32	MKI67, STMN1		
			cDC	32	CD1C, FCER1A		
			Unknown	79	LILRA4, PTCRA		
			Mast cells	Mast cells	2	KIT	
		Total	20	4,069	41		
Breast cancer	IDC (SRT)	Cancer tissue	Cluster 21	119	SREBF1, PTK7	NA	
			Cluster 28	57	KRT8, KRT18, CD44		
	BAS1 (SRT)	Connective tissue (Cluster 29)	Cancer tissue	Cluster 28	51		STAT1, ENO1, PGK1
			basal and myoepithelial cells	CAFs	50		KRT23, KRT6B, KRT5/14, and ACTA2
				Endothelial cells			COL1A1 and COL1A2
				interleukins and chemokines			ENG and VWF
	TNBC (CID44971, scRNA-seq)	Epithelial cells	Luminal progenitor 1	373	S100A1, PPP1R1B, and RGCC		
			Luminal progenitor 2	65	MMP7, KRT23, and SLC34A2		
			myoepithelial 1	86	C2orf40, MYH11, CNN1, and OXTR		
			myoepithelial 2	37	CDH13, COMP, THY1, FST, BGN, IGFBP6, and S100A2		
mature luminal			174	TFF3, AGR3, TFF1, AFF3, and FBP1			
cancer cycling			135	RRM2, CCNA2, UBE2C, CDK1, CDCA3, NUF2, MND1, and CDCA8			

			cancer basal 1	316	TMSB10, GDI2, and VIM
			cancer basal 2	162	KIF1A, SLCO1A2, and DCLK1
			cancer basal 3	151	SOX11, RNF144A, CTXN1, TLL7, and GPC2
			cancer basal 4	128	NINJ2, SLCO5A1, and GGACT
	FFPE (Her2+ breast cancer, SRT)	Cancer tissue	Cluster 15	19	CL20, S100A8, and SAA1
Colorectal cancer and liver metastasis	P1 (SRT)	Cancer tissue	Cluster 6	260	SPP1, FNI, APOE, and IFIT1
	LM1 (SRT)	Cancer tissue	Cluster 5	146	AREG and EREG
Lung cancer	NSCLC (FFPE, SRT)	Cancer tissue	Cluster 3	9436	SLC2A1
			Cluster 7	5671	COL3A1
			Cluster 8	1692	SPINK1
			Cluster 12	509	HSPA1B
			Cluster 13	383	LIF
			Cluster 16	138	CD55
			Cluster 17	112	MSMB

Supplementary Table 4. The detailed annotation of different cell types, subtypes or cell-states on the analyzed datasets. NA represents the marker gene discovered in our work.

- 10 Wu, S. Z. *et al.* A single-cell and spatially resolved atlas of human breast cancers. *Nature genetics* **53**, 1334-1347 (2021).
- 11 Lee, H.-O. *et al.* Lineage-dependent gene expression programs influence the immune landscape of colorectal cancer. *Nature genetics* **52**, 594-603 (2020).

Comment #12: Fig. 2b and Suppl Fig 1a: There is some discrepancy between cluster colours for sample 151507 (please correct the label in Fig. 2b) in these two figures. In Fig. 2b, the stKeep clusters from 151507 match the annotated data, but in Suppl. Fig. 1a, the WM and Layer4 clusters have switched places. Which of these two figures is correct? Or is it that some additional methods were applied to get the clustering shown in Fig. 2b? If so, then please specify that in the figure legend.

Response: Yes, we have revised the cluster colors in Fig.2b and Suppl Fig. 1a to make them consistently with each other.

Comment #13: Suppl Fig. 1: Slices 151669–72 contain Layers 3–6 and WM as per the annotated data, but the stKeep and other methods show Layer1, 2 clusters and no WM cluster. Is this the actual output or an issue with cluster colours?

Response: We are sorry for the oversight in cluster colors annotation on slices 151669–72. In light of the comment, we have revised the cluster colors of these figures to make them accurate.

Comment #14: Pg 6 Ln 23 and Suppl Fig. 3: In point (iv), the authors mention that “gene-modules for slice 151507 display layer-specific patterns in 11 other independent slices.” However, in the data shown in Suppl. Fig. 3 heatmaps, the layer-specificity of the gene-modules does not always match between slices. E.g., the Layer6-specific gene-modules from 151507 slice show a higher expression in WM layers of 7 independent slices (151508–10, 151673–76). Does this mean that the gene-modules that appear as Layer6-specific in slice 151507 are WM-specific in slice 151509? Based on this, the layer-specificity of gene-modules is not reliable between slices.

Response: We thank the reviewer for the comment. In light of this, we have improved our heatmaps for each gene-module in each layer displaying gene expression data normalized at both cell and gene levels. Upon analysis, we observed that the gene-modules for slice 151507 display layer-specific patterns across at least four other independent slices, particularly in slices 151508–151510. The results are shown in Supplementary Fig.3. Additionally, we have revised the relevant text and figure on page 6.

Supplementary Figure 3. The heatmap showing the mean gene expression of the seven gene-modules identified from slice 151507 in 12 different slices of the human DLPFC dataset. Note that gene expression data are normalized at both cell and gene levels. For each slice, rows and columns represent gene-modules and layers, respectively. Source data are provided as a Source Data file.

Comment #15: Suppl Fig. 5: The Layer1-specific gene *AQP4* also shows high expression in the region annotated as WM. Also, the Layer3-specific gene *CCK* shows higher expression in the region annotated as Layer6 than the red-outlined Layer3. How can this be explained? Perhaps these regions should be tested further for heterogeneity. Also, the figure legend looks incomplete.

Response: We thank the reviewer for the comment, and also apologize for the mis-annotation of gene *CCK*, which is annotated to Layer 2, Layer 3, and Layer 6, whereas being slightly over-expressed in Layer 6 compared to Layer 3². In light of the comment, we have conducted a comprehensive analysis, expanding our selection of the known layer-specific genes to validate that representative Layer 1-specific genes (e.g., *AQP4*) are indeed over-expressed in Layer 1, and representative Layer 3-specific genes (e.g., *FREM3*) show over-expression in Layer 3 across eight independent slices: slice 151507-10, and 151673-76. In addition, the adult cortex has a relatively simple structure, consisting of six nearly continuous layers (Layer 6 → Layer 5 → Layer 4 → Layer 3 → Layer 2 → Layer 1), where Layer 1 is situated closest to the outer surface, while Layer 6 is positioned deepest, and above the WM¹². Hence, the likelihood of mixing of Layer 1 and WM is minimal, reinforcing the homogeneity of these regions. The results related to the comment are shown in Supplementary Fig.6a-b (in the revised version). Additionally, we have revised the figure legend to clearly and fully describe each sub-figure. In light of the comment, we have incorporated the relevant text and a table for layer-specific genes (Supplementary Table 2) on pages 6, and 7.

Supplementary Figure 6. a Spatial expression of known layer-specific genes on slices 151669, 151670, 151671, and 151672 from one experiment. The red outline indicates the previously annotated Layer 3². It is important to note that *CCK* is annotated to Layer 2, Layer 3, and Layer 6, whereas being relatively over-expressed in Layer 6. Similarly, *ENCL1* is annotated to Layer 2 and Layer 3 while exhibiting overexpression in Layer 2². **b** Violin plot illustrates gene expression of layer-specific genes for Layers 1, 2, and 3 across eight different slices: 151507-151510, and 151673-151676. Source data are provided as a Source Data file.

- 2 Maynard, K. R. *et al.* Transcriptome-scale spatial gene expression in the human dorsolateral prefrontal cortex. *Nature neuroscience* **24**, 425-436 (2021).
- 12 Gilmore, E. C. & Herrup, K. Cortical development: layers of complexity. *Current Biology* **7**, R231-R234 (1997).

Comment #16: Fig. 3e: The right panel shows correlation between gene pairs. 1) Are the gene pairs taken from gene-modules (shown in the red violin plot) significantly correlated to each other, specifically those that are highly correlated? 2) On average, what proportion of the gene pairs from a gene module are significantly correlated? 3) Moreover, the correlation y-axis only has positive values. Does this mean that no negative correlation was observed between the gene pairs? Do the gene modules capture negatively correlated genes?

Response: We thank the reviewer for the comment, and provide the following replies for each issue. Specifically,

1) The right panel shows the correlation coefficients between gene pairs from gene-modules, not exclusively highlighting the highly correlated pairs. We have revised the text on page 8 to clarify this point.

2) On average, approximately 65% of gene pairs from a gene module exhibit significant correlations, in contrast to about 14% of random gene pairs. We have included additional text on page 8 to enhance the reliability of gene-module identification.

3) On average, in a gene-module there are roughly 12% of gene pairs with negative correlations, whereas only about 1.2% of them exhibit significant negative correlations. Our study primarily emphasizes establishing associations between genes without exploring the directionality of these relationships. In our future work, we intend to develop more advanced computational models to predict directionality of these relationships. In light of the comment, we have revised the Fig. 3e to incorporate negative correlations and added the relevant text and on pages 16, 34, and 35.

Fig.3. stKeep enables us to detect biologically meaningful cell-modules and gene-modules on IDC sample (Luminal B) and BAS1 (Her2⁺) samples. **a** Immunofluorescent staining of IDC tissue with manual annotation of 13 tumor regions: invasive carcinoma (red), carcinoma in situ (orange), and

benign hyperplasia (green). The intensity of DAPI, fiducial frame, and anti-CD3 is indicated by green, blue, and yellow. **b** Spatial clustering by Squidpy, STAGATE, stMVC, and stKeep. **c** UMAP visualization of latent features by stKeep. Each color indicates a cluster. **d** Bar plot of clustering accuracy in terms of ASW for assessing the closeness of low-dimensional representations of the same cluster compared to the other clusters. Each color indicates a sample. **e** Boxplot showing the Euclidean distance of low-dimensional features (left panel) and Pearson correlation of gene expression (right panel), for 3,534 gene pairs identified from gene-modules and 3,534 randomly selected gene pairs. Note that, on average, in a gene-module there are roughly 12% of gene pairs with negative correlations, whereas only about 1.2% of them exhibit significant negative correlations. For each boxplot, the center line, box limits and whiskers separately indicate the median, upper and lower quartiles and 1.5× interquartile range. **f** The identified gene-module for spatial cluster 21 by stKeep (left panel). Note that a regulator and its target genes are colored in red and blue, respectively. Spatial expression of key TF (*PTK7*) in the gene-module (center panel). Total survival rate of patients with the average expression of 25 signature genes for cluster 21 in luminal B breast cancer from TCGA by GEPIA2¹³ (right panel). **g** H&E plot of the tissue and manual annotation with 13 tumor regions on the BAS1 sample. The black outline indicates connective tissue, while the orange and red colors are consistent with **a**. **h** Spatial clustering of cells by stKeep. **i** One gene-module for cluster 29 by stKeep. **j** Functional annotation of gene-modules for spatial cluster 29. **k** Spatial expression of key marker genes for basal and myoepithelial cells (*KRT23*, *KRT5*, *KRT14*, and *ACTA2*), CAF (*COL1A1* and *COL1A2*), endothelial cells (*ENG* and *VWF*), interleukins and chemokines (*CXCL8*, *CX3CL1*, *CCL17*, and *CXCL3*), and key TFs (*TP63*, *RUNX1*, *ETS2*, and *EBF1*) in cluster 29. Source data are provided as a Source Data file.

13 Tang, Z., Kang, B., Li, C., Chen, T. & Zhang, Z. GEPIA2: an enhanced web server for large-scale expression profiling and interactive analysis. *Nucleic Acids Res* **47**, W556-W560 (2019).

Comment #17: Fig. 3h and k: The black squares shown in these figures highlight cluster 29 but also contain pieces of other clusters at the edges. In Fig. 3k, the genes associated with CAF and endothelial cells have high expression around the edges of the black square, and the cluster 29 may not be as heterogeneous as it appears based on the gene expression seen in the black square outline. Would it be possible to add an exact outline of cluster 29 to these figures in place of the black box to make it clearer. A similar approach for the other figures showing gene expression in specific clusters might also be useful.

Response: We thank the reviewer for the comment. Our analysis revealed that cluster 29 in Fig.3h and k (ref to Comment #16) comprises a heterogeneous tissue, consisting of basal tumor cells, myoepithelial cells, cancer-related fibroblasts (CAFs), and endothelial cells. In light of the comment, we have revised all SRT-related figures throughout the manuscript to ensure that each outline accurately indicates our interested clusters.

Comment #18: Fig. 5a: Compared to the annotation, the stKeep cluster 9 has stroma and adipose as well as lymphocyte regions. Has cluster 9 been examined further to ensure its homogeneity as indicated by stKeep output?

Response: That's a good point. In light of the comment, we have thoroughly examined the homogeneity of cluster 9 by estimating the proportions of different cell types based on paired scRNA-seq data using GraphST¹⁴. By such analysis, we found that (1) the lymphocyte region annotated in cluster 9 comprises

not only T cells but also a large number of stromal cells. This mixed composition explains why stKeep identifies these regions as a cluster; and (2) other SRT methods are unable to identify distinct sub-regions within the lymphocyte region. To analyze heterogeneous cell populations within the stroma, adipose and lymphocyte regions, we conducted a detailed clustering analysis based on cell proportions of T and stromal cells, identifying three sub-clusters. To further validate the accuracy, we conducted gene enrichment analysis of the up-regulated genes of each sub-cluster using DAVID (<https://david.ncifcrf.gov/tools.jsp>). Three sub-clusters show different functions: sub-cluster 1 cells are related to extracellular matrix organization and collagen degradation, possibly influenced by infiltrating stromal cells; sub-cluster 2 cells participate in MAP kinase activation and overexpress *VEGFA* and *FGFR1*, potentially regulating endothelial cells response to stress; and sub-cluster 3 cells exhibit diverse functions such as T cell receptor signaling pathway, T cell activation, and Th1 and Th2 cell differentiation. The results for the comment are shown in Supplementary Fig.10a-e. In light of the comment, we have added the related text on page 10.

Supplementary Figure 10. Sub-clustering analysis on cluster 9 in the TNBC sample. a UMAP plot of nine different cell types of 7,984 cells from a TNBC sample. Each color denotes a cell type. **b** Spatial

proportions of nine cell types predicted by GraphST ¹⁴. Each spot is a pie chart of the probability of its corresponding cell type. Each color indicates one cell type. **c** Clustering analysis of cell proportions of T and stromal cells within stroma, adipose and lymphocyte regions. **d** Violin plot showing marker gene expression across three subclusters: *COL1A1*, *COL1A2*, and *COL3A1* for subcluster 1; *VEGFA*, *FGFR1*, and *EZH2* for subcluster 2; and *CXCL9*, *CCL19*, and *CD3D* for subcluster 3. **e** Functional annotation of over-expressed genes in each subcluster using DAVID (<https://david.ncicrf.gov/tools.jsp>). The color and size indicate $-\log_{10}(p - \text{value})$ and count, respectively. Source data are provided as a Source Data file.

14 Long, Y. *et al.* Spatially informed clustering, integration, and deconvolution of spatial transcriptomics with GraphST. *Nat Commun* **14**, 1155 (2023).

Comment #19: Pg 16 Ln 5–6: The authors mention “then assigned each cell to its corresponding region using established software.” Please specify the software used.

Response: Yes, we have made the following changes: we assigned each cell/spot to its corresponding region using our previous established software in stMVC ¹, and also have revised the text on page 17 (in the revised version).

1 Zuo, C. *et al.* Elucidating tumor heterogeneity from spatially resolved transcriptomics data by multi-view graph collaborative learning. *Nature Communications* **13**, 5962 (2022).

Comment #20: The histology and spot images showing gene expression should have a scale bar, since each spot has multiple cells in it.

Response: Yes, we have added scale bars into each plot for both spatial histological and gene expression data.

Comment #21: Several datasets were evaluated in this study, and it would be good to have them listed in a table – the contents should include dataset name, sample type, number of replicates (if any), SRT technology used, number of spots, number of genes, and publication describing annotated data. Some of this information is provided in the Methods section, but it would be better presented in a table.

Response: Yes, we agree. In light of this, we have included a table as suggested to present a detailed dataset information in Supplementary Table 1, and we have also cited the table throughout the manuscript.

SRT technology	Dataset name	Sample type	#Replicates	Slice	#Spots / cells	#Genes	Reference
	DLPFC	Human brain dorsolateral prefrontal cortex	4	151507	4,226	>20,000	²
				151508	4,384	>20,000	
				151509	4,789	>20,000	
				151510	4,634	>20,000	
			4	151669	3,661	>20,000	
				151670	3,498	>20,000	
				151671	4,110	>20,000	
				151672	4,015	>20,000	
			4	151673	3,639	>20,000	
				151674	3,673	>20,000	
				151675	3,592	>20,000	

Visium (fresh frozen)				151676	3,460	>20,000	
	IDC	Luminal B breast cancer	1	IDC	4,727	>20,000	15
	BAS1	Her2+ breast cancer	1	BAS1	3,798	>20,000	16,17
	TNBC	Triple-negative breast cancer	1	CID44971	1,162	>20,000	10
	P1 & LM1 (same patient)	Primary colorectal cancer	1	P1	2,917	>20,000	18
		Matched liver metastasis	1	LM1	3,826	>20,000	
LM2	Liver metastasis	1	LM2	3,721	>20,000		
Visium (FFPE)	FFPE (Her2+ breast cancer)	Her2+ breast cancer	1	FFPE (Her2+ breast cancer)	2,239	>20,000	19
NanoString (FFPE)	NSCLC (FFPE)	Non-small cell lung cancer	1	NSCLC (FFPE)	98,002	980	20

Supplementary Table 1. Description of all used SRT datasets in this study.

- 2 Maynard, K. R. *et al.* Transcriptome-scale spatial gene expression in the human dorsolateral prefrontal cortex. *Nature neuroscience* **24**, 425-436 (2021).
- 10 Wu, S. Z. *et al.* A single-cell and spatially resolved atlas of human breast cancers. *Nature genetics* **53**, 1334-1347 (2021).
- 15 Zhao, E. *et al.* Spatial transcriptomics at subspot resolution with BayesSpace. *Nature Biotechnology* **39**, 1375-1384 (2021).
- 16 Xun, Z. *et al.* Reconstruction of the tumor spatial microenvironment along the malignant-boundary-nonmalignant axis. *Nat Commun* **14**, 933 (2023).
- 17 Xu, C. *et al.* DeepST: identifying spatial domains in spatial transcriptomics by deep learning. *Nucleic Acids Res* **50**, e131-e131 (2022).
- 18 Wu, Y. *et al.* Spatiotemporal immune landscape of colorectal cancer liver metastasis at single-cell level. *Cancer discovery* **12**, 134-153 (2022).
- 19 10X Genomics. *Human Breast Cancer: Ductal Carcinoma In Situ, Invasive Carcinoma (FFPE)*, <<https://www.10xgenomics.com/resources/datasets/human-breast-cancer-ductal-carcinoma-in-situ-invasive-carcinoma-ffpe-1-standard-1-3-0>> (2023).
- 20 Nanostring-Biostats. *CosMx SMI NSCLC FFPE Dataset*, <<https://nanostring.com/products/cosmx-spatial-molecular-imager/ffpe-dataset/nsclc-ffpe-dataset/>> (2023).

Comment #22: Please mention the full form of SVG in the main text.

Response: Yes, we have added the spatially variable gene (SVG) as suggested on page 8.

Comment #23: Fig. 2b and Suppl Figs. 1, 4 – Could the thick black, circular border be replaced with a thin border for the legends representing layers in these figures.

Response: Yes, we have revised the annotation of cluster colors on Fig. 2b and Supplementary Figs. 1 and 5 (in the revised version) for clear visualization.

Comment #24: Pg 7 Ln 3 – Point (iv) should be point (iii).

Response: We have revised the text as suggested on page 7.

Comment #25: Fig. 5c: In the right panel, the marker gene labels are not readable. The label sizes need to be increased, or the marker genes can be mentioned in the figure legend in the order they are shown in the figure.

Response: Yes, we have increased the label size in Fig.5c and also added the marker genes in the figure legend on pages 38, and 39.

Reviewer #3

Comment #1: I co-reviewed this manuscript with one of the reviewers who provided the listed reports. This is part of the Nature Communications initiative to facilitate training in peer review and to provide appropriate recognition for Early Career Researchers who co-review manuscripts.

Response: Yes, we thank the reviewer for the comment.

Reviewer #4

Comment #1: The stKeep method by Zuo & Chen describes a new approach for multi-modal analyses of spatial transcriptomics data using heterogeneous graph learning. stKeep first embeds nodes and edges of a unified graph using data modalities from gene expression, spatial location and histological regions. The method first identifies cell-modules by learning hierarchical and global semantic representations in this graph, which are integrated by contrastive learning into a space for spatial clustering and visualization. stKeep then leverages known gene databases to learn gene-modules which describe gene-regulatory networks and protein-protein interactions. The gene-module analysis extends to identify ligand-receptor pairs in neighboring cells/spots to predict cell signaling interactions. The authors apply stKeep to publicly available datasets generated on the Visium platform, which yield interesting results in model tissue types such as the brain, and in complex tissue types such as breast and other cancer tissues. The code for stKeep appears to be well packaged, documented and readily installable for public use in a user-friendly manner.

Response: Yes, we thank the reviewer for the comment and encouragement.

Comment #2: 1) Whilst stKeep appears to strongly outperform other methods it is benchmarked against, one main limitation is its primary application to Visium data in the study, which are not single-cell resolution, and 'spots' are not 'cells' as described throughout the manuscript. 2) Visium spots are comprised of multiple cells depending on tissue and pathological region, which makes it somewhat difficult to interpret GRNs, PPIs and LRPs in 'cells' which are rather composed of many cell types. This factor should be discussed in the manuscript. 3) Furthermore, have the authors applied stKeep to other publicly available datasets from higher resolution spatial transcriptomics technologies to address this?

Response: We thank the reviewer for the comment. We have done the following analyses for each issue, specifically,

1) We have revised the corresponding text and figure regarding cell and spot as suggested throughout the manuscript to make it clearer.

2) We have added the following description in the discussion on pages 14 and 15 on the interpretation of cell-modules, gene-modules, and CCC modules from spot-level SRT data using scRNA-seq data. Specifically, (i) detecting key players within TME via CCC: Utilizing cell type-specific ligands and receptors from scRNA-seq, stKeep quantifies the involvement of different cell populations in various cancer cell-states (or cell-modules), pinpointing key cell types; (ii) identifying cell-state specific gene-modules: Gene-modules might contain genes expressed across multiple cell types, however, the identification of gene-gene relations from the known PPI and GRN database knowledge ensures its accuracy. stKeep utilizes scRNA-seq data to infer where co-related gene pairs might be co-expressed; and (iii) predicting crucial mechanisms in disease development: Mapping cell subtypes from scRNA-seq over different spatial cancer regions, stKeep combines cell subtype-specific over-expressed genes and gene-modules with the inferred CCC from SRT data to uncover regulatory mechanisms relevant to disease progression.

3) We have demonstrated the application of stKeep in processing single-cell resolution SRT data by NanoString, and found that stKeep enables to detect more cancer cell-states, particularly in the interface region between tumors and stromal cells, and elucidating internal gene regulatory and CCC mechanisms. The results are shown in Supplementary Fig.19a-g. In light of the comment, we have added the related text on pages 2, 4, 14, 23, 24, and 25.

Supplementary Figure 19. Method comparisons on the NSCLC (FFPE) sample by NanoString technology. **a** IF staining image showing manual annotation of 17 tumor regions (red outline), and niches from the NanoString website (<https://nanosttring.com/products/cosmx-spatial-molecular-imager/ffpe-dataset/nsclc-ffpe-dataset/>). The intensity of DAPI (cell nuclei), PanCK (tumor cells),

CD45 (leucocytes), and CD3 (T cells) is indicated by blue, green, yellow, and red, respectively. **b** Spatial clustering predicted by Seurat and stKeep. **c** Spatial expression of *KRT19* for epithelial cells (left panel). Spatial clusters within tumor regions (right panel). **d** Violin plot displaying expression levels of *SLC2A1*, *COL3A1*, *SPINK1*, *HSPA1B*, *LIF*, *CD55*, and *MSMB* across seven clusters. Each color indicates one cluster. **e** The identified gene-module for spatial cluster 7. The regulator and its target genes are colored in red and blue, respectively. Here, we displayed the over-expressed genes in spatial cluster 7 compared to other clusters, with a \log_2fc greater than 3. **f** Scatter plot displaying the over-expressed genes in spatial cluster 7. **g** Spatial expression of ligand *COL3A1* and receptor *DDR1*, along with their corresponding CCC interaction strengths. The black outline in **a**, **b**, **c**, and **g** indicates tumor regions, consistent with the red color in **a**. Source data are provided as a Source Data file.

Comment #3: It is unclear how much of each data modality input contributes to the overall interpretations of the heterogeneous graph and the resulting cell- and gene- modules. The authors should comment on how each of the input data modalities contribute to the resulting cell-modules e.g. how manually-defined histological regions compare to spatial locations, which are both very similar inputs of spot locations.

Response: Yes, we have further conducted substantial experiments to assess the impact of each graph component on the performance of cell-modules and gene-modules using 12 slices from human DLPFC dataset. Specifically, (1) for each graph or component within the cell module, we evaluated its contribution to the identification of spatial clustering by removing the graph, and measured its impact using the average silhouette width (ASW). The lower the ASW, the higher the contribution of the graph. The comparison results revealed the following order of contribution, from most to least: spatial location graph (SLG), cell-region graph (CRG), histological similarity graph (HSG), transcriptomics similarity graph (TSG), and cell-gene graph (CGG). Notably, while the removal of CGG results in tighter clustering, it fails to accurately detect boundaries in finer structures (e.g., cluster 1 in slice 151669). In line with the findings from the DLPFC dataset, CGG demonstrates an important role in accurately detecting boundaries within cancer samples, notably highlighting cluster 33 in IDC and cluster 17 in BAS1, which is visually marked by black outlines. These results confirmed that our cell-module's structure is well designed and it enables to detect cell heterogeneity from different histological regions; and (2) for each graph or component within the gene module, we assessed its contribution to the learning of gene embeddings by removing the graph, and measured this influence by calculating the distance between gene pairs identified from gene-modules using stKeep. The shorter the distance, the higher the contribution of the graph. The comparison results showed the order of contribution, from most to least: gene-cell state graph (GSG), gene-cell graph (GCG), PPI, and GRN, where PPI and GRN have a comparable performance. The results are shown in Supplementary Figs.13a-c, and 14a. In light of the comment, we have included the text on page 13.

Supplementary Figure 13. Ablation analysis of graph component within the cell module. **a** Violin plot showing clustering score ASW used to evaluate the contribution of each graph within the cell module on 12 human DLPFC slices. Specifically, after removing each graph, we predicted the spatial clustering based on the learned low-dimensional features. The lower the ASW, the higher the contribution of the graph. **b** Bar plot showing clustering score ASW used to evaluate the contribution of each graph within the cell module on two cancer samples (IDC and BAS1). **c** Spatial clustering predicted by the low-dimensional features by removing each graph from stKeep on slice 151669, IDC, and BAS1 samples, where we also provide stKeep as a comparison. Note that spatial location graph, cell-region graph, histological similarity graph, transcriptomics similarity graph, and cell-gene graph are abbreviated as SLG, CRG, HSG, TSG, and CGG, respectively. Source data are provided as a Source Data file.

Supplementary Figure 14. Evaluation of the gene and CCC modules in stKeep on 12 human DLPFC slices. **a** Violin plot showing distance of gene-pairs (identified by stKeep) to evaluate the contribution of each graph within the gene module. Specifically, after removing each graph, we calculated the distance of the gene-pairs based on the learned gene embedding. The shorter distance, the higher contribution of the graph. Note that gene-cell state graph and gene-cell graph are abbreviated as GSG and GCG, respectively. **b** Violin plot showing the false positive rate (FPR) of the learned gene embedding trained at different proportions of incomplete gene-gene interactions (PPI and GRN). **c** Violin plot showing the recover edge rate (RER) for the learned gene embedding trained at different scales of incomplete gene-gene interactions (PPI and GRN). **d** Line plot showing FPR of the learned gene embedding trained at different proportions of incorrect gene-gene interactions (PPI and GRN). Each color indicates a slice. **e** Violin plot showing Pearson correlation of the inferred CCC between spots within a cluster, inferred for 12 slices with different proportion LRPs removed. Source data are provided as a Source Data file.

Comment #4: On a similar note, the authors should clarify how manually annotated histological regions impact the subsequent downstream learning of cell-modules. Is it unclear how this step is performed via the expression of marker genes for cancer epithelial, stromal and immune cells, as stated in the methods. Considering the authors utilize 10X Visium data, which are not close to single-cell and also

suffer from gene drop-out, the expression of these marker genes may not be robust to clearly define accurate histological regions compared to specialized pathology annotations, particularly for heterogeneous breast tumors.

Response: We are sorry for the unclear description. In light of the comment, we have provided additional clarification regarding the definition of histological regions¹. We defined histological regions as follows: cells/spots of the same histological type that are separated by immune or other mesenchymal cells may exhibit different cell-states, implying that they belong to different histological regions. Specifically, we used the colors of the antibodies in the IF staining images to distinguish these regions, while leveraging the spatial distribution of classical marker genes of T (*CD3D*, *CD4*, and *CD8A*), myoid (*CD68* and *CD163*), B (*CD19*), and stromal (*COL1A1*) cells to distinguish different histological regions from H&E staining images. In summary, we classified the pathology annotations into distinct histological regions based on spatial distribution of immune or stromal cells, rather than defining histological regions solely based on spatial distribution of different cell types. We have included the text as suggested on page 17. In addition, regarding the impact of histological regions on the learning of cell-modules, please refer to Comment #5.

1 Zuo, C. *et al.* Elucidating tumor heterogeneity from spatially resolved transcriptomics data by multi-view graph collaborative learning. *Nature Communications* **13**, 5962 (2022).

Comment #5: One concern is that the spatial clusters identified by stKeep (e.g., Fig 2b) map nearly perfect (NMI ≈ 1) to the pre-defined annotations used as input into stKeep. How much of a bias do these manually defined histological regions have on the contrastive learning step when integrating local hierarchical and global semantic representations of the graph? Considering “positive samples are defined as its spatial nearest neighbors belonging to the same region, and all other cells are considered negative samples”, does the optimization process favor cell-modules which reflect the manually annotated histological regions?

Response: We appreciate the reviewer for the comment. In this study, our primary goal is to dissect the cell heterogeneity of spatial structure from SRT data, particularly within the tumor microenvironment from different histological annotations/regions. Regarding the usage of prior annotation in DLPC dataset as input, we acknowledged that the evaluation of NMI by annotation may be not fair. Consequently, we have decided to remove NMI from Fig 2b, retaining ASW as the metric for assessing the proximity of low-dimensional features within each cluster. To evaluate the ability of stKeep to unravel the heterogeneous tissue from histological regions, we used three different types of regions as the input of stKeep: five regions (WM, Layer 6, Layer 5, Layer 4, and a region containing Layers 3, 2, and 1), six regions (WM, Layer 6, Layer 5, Layer 4, Layer 3, and a region containing Layers 2, and 1), and six regions (Layer 5, Layer 4, Layer 3, Layer 2, Layer 1, and a region containing WM and Layer 6). Our comprehensive comparison revealed that stKeep enables dissecting heterogeneous cell populations from the provided histological regions, consistent with the findings from the cancer samples. In summary, we assert that the selection of positive and negative samples encourages cells/spots from the same region to be clustered closely in the latent space, thereby facilitating sub-clustering within these regions while maintaining overall cell heterogeneity. The results are shown in Supplementary Figs.15a, 16a, and 17a. In light of the comment, we have included the text on pages 13, and 20.

Supplementary Figure 15. Evaluation of spatial clustering with different numbers of histological regions and dimensional sizes on the IDC sample. **a** Spatial clustering by stKeep using varying numbers of histological regions. The top panel indicates the varied histological regions, while the bottom panel displays the corresponding spatial clustering results. In the model training without histological regions, positive samples are determined based on spatial nearest neighbors, while others are considered as negative samples. It is noteworthy that stKeep uses 14 histological regions as input for this analysis. **b** Spatial clustering predictions using the separately learned local hierarchical representations (R^1) and global semantic representations (R^2) with different dimensions (30, 40, and 50), as well as their simple concatenation (Concat (R^1 , R^2)). Additionally, we present spatial clustering results by concatenating the optimal dimensions for R^1 and R^2 . Source data are provided as a Source Data file.

Supplementary Figure 16. Evaluation of spatial clustering with varying numbers of histological regions and dimensional sizes on the BAS1 sample. a Spatial clustering by stKeep using different numbers of histological regions. The top panel indicates the varied histological regions, while the bottom panel displays the corresponding spatial clustering results. In the model training without histological regions, positive samples are determined based on spatial nearest neighbors, while others are considered as negative samples. It is noteworthy that stKeep uses 14 histological regions as input for this analysis. **b** Spatial clustering predictions using the separately learned local hierarchical representations (R^1) and global semantic representations (R^2) with different dimensions (30, 40, and 50), as well as their simple concatenation (Concat (R^1 , R^2)). Additionally, we present spatial clustering results by concatenating the optimal dimensions for R^1 and R^2 . Source data are provided as a Source Data file.

Supplementary Figure 17. The evaluation of histological regions on cell modules, gene modules, and CCC modules in four human DLPFC slices. **a** Spatial clustering by stKeep using varying numbers of histological regions, i.e., five regions (WM, Layer 6, Layer 5, Layer 4, and a region containing Layers 3, 2, and 1), six regions (WM, Layer 6, Layer 5, Layer 4, Layer 3, and a region containing Layers 2, and 1), and six regions (Layer 5, Layer 4, Layer 3, Layer 2, Layer 1, and a region containing WM and Layer 6). The left and right panels indicate spatial distribution and UMAP visualization of the prediction. **b** Identification of gene-modules through the predicted spatial clustering results trained under different numbers of histological regions. For each slice, the left, middle, and right panels represent the UMAP visualization, mean gene expression of the identified gene-modules in different predicted clusters, mean expression of the identified gene-modules in different annotated layers, respectively. **c** Boxplot showing Pearson correlation of interaction strength between spots within a cluster, inferred by stKeep under different histological regions for four slices, where we also provided the same number of the randomly selected spot pairs for comparison. For each boxplot, the center line, box limits and whiskers separately indicate the median, upper and lower quartiles and $1.5 \times$ interquartile range. **d** Heatmap showing the ratio of predicted the cluster-specific CCCs to layer-specific CCCs inferred by stKeep. Source data are provided as a Source Data file.

Comment #6: Considering that cell-states/-modules are used as input for learning gene-modules, a follow-on question is how these manually defined histological regions further impact the learning of gene-modules and subsequent GRNs, PPIs and LRPs?

Response: We thank the reviewer for the comment. We have performed additional evaluations of the resulting gene-modules based on the predicted cell-states/-modules under three different types of region inputs (ref to Comment #5) for slices 151673-151676. We found that (1) the identified gene-modules exhibit cell-state specific patterns, with most of them showing region-specific patterns based on gene expression levels between the input regions; and (2) more than 70% of gene-modules in the three types of heterogeneous region exhibit layer-specific patterns. In addition, we exploited positive pairs derived from these defined conditions to infer CCC. The results showed that (1) the Pearson correlation of CCC between spots within a cluster is comparable to that inferred from positive pairs based on the annotated seven layers, and is significantly higher than that from the randomly selected spot pairs; and (2) over 75% of cluster-specific LRPs are consistent with the layer-specific LRPs. Together, these findings suggest that, while coarse annotation of histological regions may sacrifice some within-region heterogeneity, they do not affect the identification of gene-modules and LRPs across different regions. The results are shown in Supplementary Fig.17b-d (ref to Comment #5). In light of the comment, we have added the text on page 13.

Comment #7: Similarly, provided that cell-states/-modules are used in the learning of subsequent gene-modules, the over-expression of gene modules in specific spatial clusters feels highly circular in nature (Fig 2e; p6 line 20-21). Thus, it is not unexpected that gene-pairs will be more closely characterized and correlate better than random gene pairs (Fig 2c; Fig 3e). Can the authors apply additional metrics such as pathway enrichment to show that gene-module signatures are enriched for independent layer specific genes beyond a small number of markers? This feels like the minimal data required to suggest that stKeep identifies ‘biologically meaningful gene-modules’.

Response: Yes, we agree with the reviewer. We have performed gene enrichment analysis for genes within a gene-module, and found that each gene-module exhibits specific functions: the WM gene-

module is linked to central nervous system myelination and oligodendrocyte differentiation; Layer 6 is associated with cell response to glucocorticoid stimulus; Layer 5 relates to the dopamine neurotransmitter release cycle; Layer 4 is involved in various functions like neurofilament bundle assembly, peripheral nervous system axon regeneration, and neuroendocrine cell differentiation; Layer 3 contributes to corticotrophin secretion and the glucocorticoid receptor signaling pathway; Layer 2 is associated with regulating synaptic transmission and glutamatergic; and Layer 1 participates in neurotransmitter uptake and metabolism in glial cells and the formation of the anterior neural plate. The results are shown in Supplementary Fig. 4. In light of the comment, we have included the related text on page 6.

Supplementary Figure 4. Functional annotation of seven gene-modules identified from slice 151507 using DAVID online website (<https://david.ncifcrf.gov/tools.jsp>). For each annotation on each layer, the color and size indicate the $-\log_{10}(p - \text{value})$ and count, respectively. Source data are provided as a Source Data file.

Comment #8: Can the authors comment on why COMMOT (Fig 2g), which is a highly optimized method for computing cell-cell communication using Optimal Transport, produces almost random noise within a cluster compared to stKeep, with Pearson correlation values in cell-pairs ~ 0 ?

Response: We have re-evaluated the interaction strengths inferred by COMMOT and calculated Pearson correlation between spots within a cluster. We found that (i) while the correlation of CCC intensities is slightly higher for COMMOT than that of stKeep, significant variations exist in the correlations between spot pairs within clusters and the randomly selected pairs in stKeep versus COMMOT, highlighting that the CCC intensities inferred by stKeep better capture biological relevance and specificity; (ii) surprisingly, in the CCC analysis by COMMOT on slices 151669-151671, spot correlations within clusters are lower than those of the randomly selected spot pairs. This might be due to COMMOT emphasizing the interplay among diverse ligands and receptors alongside spatial

distances, potentially overlooking cell state heterogeneity during CCC inference. The results are shown in Fig.2g. In light of the comment, we have revised the figure and the relevant text on pages 7, 32, and 33.

Fig.2. stKeep is able to identify cell-modules, gene-modules, and CCC on the human DLPFC dataset. **a** Boxplot of clustering score ASW for 12 slices by Squidpy, STAGATE, stMVC, and stKeep. **b** Spatial clusters annotated by the previous study², and detected by Squidpy, STAGATE, stMVC, and stKeep, on slice 151507. **c** Boxplot displaying the Euclidean distance of low-dimensional features for gene-pairs from gene-modules and the same number of the randomly selected gene pairs, for 12 slices. **d** UMAP visualization of gene-module representations for slice 151507 by stKeep (left panel). Each color indicates a gene-module. Right panel indicates the identified gene-module for Layer 1. The regulator and its target genes are colored in red and blue, respectively. For clear visualization, we displayed the over-expressed genes in Layer 1 compared to other layers, with a \log_2FC greater than 0.8. **e** Heatmap showing mean expression of the identified gene-modules for slice 151507. **f** Spatial expression of the known layer-specific genes, for slice 151507 data denoised by stKeep, where we also provide raw data as a comparison. **g** Boxplot showing the Pearson correlation of interaction strength between 100k spot pairs within a cluster, inferred for 12 slices by stKeep and COMMOT, where we also provide the same number of the randomly selected spot pairs for comparison. For each boxplot of **a**, **c** and **g**, the center line, box limits and whiskers separately indicate the median, upper and lower quartiles and $1.5 \times$ interquartile range. **h** Spatial expression of the highly expressed ligands and receptors, and their corresponding CCC interaction strengths, for four slices. **i** UMAP visualization of gene latent features for slice 151669 by stKeep. Each gene-module is indicated by one color. **j** Spatial clusters annotated by the previous study² (top panel), and predicted by stKeep (bottom panel), on slice 151669. **k** spatial expression of ligands (*RELN* and *PENK*), and receptors (*ITGB1* and *ADRA2A*), and their corresponding CCC interaction strengths, on slice 151669. Source data are provided as a Source Data file.

2 Maynard, K. R. *et al.* Transcriptome-scale spatial gene expression in the human dorsolateral prefrontal cortex. *Nature neuroscience* **24**, 425-436 (2021).

Comment #9: 1) Since the authors compute gene modules, it is unclear why they only used one gene for survival analysis (e.g., *TP63* or *PTK7*; Fig 3f & i). How were these genes from each module selected - was this a ranked approach based on some gene weightings? 2) It appears more appropriate that survival analysis is performed on a signature built from these modules.

Response: We thank the reviewer for the comment. For each issue, we have done the following. Specifically,

1) We applied the most highly linked transcription factors such as *PTK7* or *TP63* within our identified gene-module into the survival analysis.

2) We agree with the reviewer's opinion that gene signatures from gene-modules are more suitable for survival analysis. In light of the comment, we have performed a survival analysis on the gene signatures from the identified gene-module. Notably, we observed a significant correlation between the average levels of 25 signature genes regulated by *PTK7* and survival using an independent TCGA dataset. In addition, given that the region (i.e., connective tissue) represents a heterogeneous tissue comprising basal cells, normal myoepithelial cells, cancer-related fibroblasts (CAFs), and endothelial cells, we believed that it would be inappropriate to explore the relationship between gene-modules regulated by *TP63* and survival in the context of cancer samples from TCGA database. Therefore, we have chosen to remove the analysis from our study. The results for the comment are shown in Fig.3f, and g-k. In light of the comment, we have included the related text on pages 8, 34, and 35.

Fig.3. stKeep enables us to detect biologically meaningful cell-modules and gene-modules on IDC sample (Luminal B) and BAS1 (Her2+) samples. **a** Immunofluorescent staining of IDC tissue with manual annotation of 13 tumor regions: invasive carcinoma (red), carcinoma in situ (orange), and benign hyperplasia (green). The intensity of DAPI, fiducial frame, and anti-CD3 is indicated by green, blue, and yellow. **b** Spatial clustering by Squidpy, STAGATE, stMVC, and stKeep. **c** UMAP visualization of latent features by stKeep. Each color indicates a cluster. **d** Bar plot of clustering accuracy in terms of ASW for assessing the closeness of low-dimensional representations of the same cluster compared to the other clusters. Each color indicates a sample. **e** Boxplot showing the Euclidean distance of low-dimensional features (left panel) and Pearson correlation of gene expression (right panel), for 3,534 gene pairs identified from gene-modules and 3,534 randomly selected gene pairs. Note that, on average, in a gene-module there are roughly 12% of gene pairs with negative correlations, whereas only about 1.2% of them exhibit significant negative correlations. For each boxplot, the center line, box limits and whiskers separately indicate the median, upper and lower quartiles and 1.5× interquartile range. **f** The identified gene-module for spatial cluster 21 by stKeep (left panel). Note that a regulator and its target genes are colored in red and blue, respectively. Spatial expression of key TF (*PTK7*) in the gene-module (center panel). Total survival rate of patients with the average expression of 25 signature genes for cluster 21 in luminal B breast cancer from TCGA by GEPIA2¹³ (right panel). **g** H&E plot of the tissue and manual annotation with 13 tumor regions on the BAS1 sample. The black outline indicates connective tissue, while the orange and red colors are consistent with a. **h** Spatial clustering of cells by stKeep. **i** One gene-module for cluster 29 by stKeep. **j** Functional annotation of gene-modules for spatial cluster 29. **k** Spatial expression of key marker genes for basal and myoepithelial cells (*KRT23*, *KRT5*, *KRT14*, and *ACTA2*), CAF (*COL1A1* and *COL1A2*), endothelial cells (*ENG* and *VWF*), interleukins and chemokines (*CXCL8*, *CX3CL1*, *CCL17*, and *CXCL3*), and key TFs (*TP63*, *RUNX1*, *ETS2*, and *EBF1*) in cluster 29. Source data are provided as a Source Data file.

13 Tang, Z., Kang, B., Li, C., Chen, T. & Zhang, Z. GEPIA2: an enhanced web server for large-scale expression profiling and interactive analysis. *Nucleic Acids Res* **47**, W556-W560 (2019).

Comment #10: 1) Considering that the authors are studying breast cancers (e.g., in spatial cluster 29 of Fig 3g-k), the term keratinocytes and tumor-specific keratinocytes (TSK) are not appropriate for studying breast tissue and should be reserved for skin & melanoma studies. 2) Furthermore, spatial cluster 29 in the BAS1 Her2+ breast cancer (Fig 3g-k) appears in regions annotated as ‘carcinoma in situ’ and many of the genes described are markers of myoepithelial cells (*KRT5/14*). It is possible this region represents contaminating normal breast epithelium? Can the authors should provide pathology annotations and a high-resolution insert of this region to confirm this?

Response: We thank the reviewer for the comment. For each issue, we have done the following analyses, specifically,

1) We have conducted a comprehensive search for the term of keratinocytes in breast cancer using Google Scholar, and found that this term is uncommon. Consequently, we have removed the relevant description as suggested in the manuscript.

2) We have performed a detailed analysis of spatial cluster 29, and observed that (i) cluster 29 represents the connective tissue adjacent to ductal carcinoma in site; (ii) the cells in the cluster exhibit over-expression of markers for basal and myoepithelial cells (e.g., *KRT23*, *KRT4/15*, *KRT6B*, and *ACTA2*), cancer-related fibroblasts (*COL1A1* and *COL1A2*), endothelial cells (*ENG* and *VWF*), as well

as interleukins and chemokines (*CXCL8*, *CX3CL1*, *CCL17*, and *CXCL3*)²¹; and (iii) the gene-modules active in the cluster are involved in various functions, including angiogenesis, chemokine and interleukin-mediated signaling pathways, collagen degradation, mesenchyme migration, innate immune response, cell-cell adhesion, neutrophil and monocyte chemotaxis. In summary, these results suggested that cluster 29 represents a heterogeneous tissue composed of basal tumor cells, myoepithelial cells and a fibrovascular niche, indicating a potential route for cancer cell invasion through migration, which is consistent with previous findings²². The results for the comment are shown in Fig.3g-k (ref to Comment #9) and supplementary Fig.8h, and the related text has been included on pages 8, 9, 14, 34, and 35.

Supplementary Figure 8. Method comparisons on the BAS1 (i.e., Her2⁺ breast cancer) sample. a Spatial clustering by Squidpy, STAGATE, and stMVC. **b** UMAP visualization of the latent features extracted by Squidpy, STAGATE, stMVC, and stKeep, respectively. Each spatial cluster is indicated by a color. **c** The identified gene-module for spatial cluster 28 by stKeep. Regulator genes and target genes are highlighted in red and blue, respectively. **d** Functional annotation of two gene modules for spatial cluster 28 by stKeep. **e** Spatial expression of genes over-expressed in spatial cluster 28. **f** Spatial

interaction strength between *FNI* and its two receptors. **g** The identified gene-module 2 for spatial cluster 29 by stKeep. **h** Spatial expression of basal genes: *KRT6B*, *KRT15*, *KRT16*, *KRT17*, and *KRT81*. Source data are provided as a Source Data file.

- 21 Kumar, T. et al. A spatially resolved single-cell genomic atlas of the adult human breast. *Nature* 620, 181-191, doi:10.1038/s41586-023-06252-9 (2023).
- 22 Camp, J. T. et al. Interactions with fibroblasts are distinct in Basal-like and luminal breast cancers. *Molecular Cancer Research* 9, 3-13 (2011).

Comment #11: Breast myoepithelial cells make up the outer layer of the breast epithelium and are thought of the physical barrier which break down as breast cancers transition from in-situ to invasive stages. 1) In Fig 5d, it is not expected to observe no enrichment of myoepithelial-1 or -2 signals in normal ductal or carcinoma in-situ regions, where they are known to be by histology. Their enrichment only in invasive carcinoma regions seems like an artefact and are likely due to their similar gene expression profiles to basal-like cancer cells. 2) Interactions such as *COMP* -> *ITGAV*, *CD47* etc. seem to be mostly enriched in invasive cancer regions (bottom left tissue; Fig 5i) and are likely all interactions of basal-like cancer cells in the TNBC sample used. The authors should revise the manuscript to reflect this or show that their CCC findings are indeed restricted to myoepithelial cells rather than cancer cells.

Response: We thank the reviewer for the comment, and have done the following analyses for each issue. Specifically,

1) In our analysis result, myoepithelial 1 cells are enriched in normal and invasive carcinoma regions, while myoepithelial 2 cells exhibit enrichment in carcinoma in-situ and invasive carcinoma regions. In light of the comment, we have revised Fig.5d and related text to make it more clear on pages 38, and 39.

(2) We have further checked the gene expression data distribution of *COMP* on scRNA-seq data, and found that *COMP* is exclusively expressed on myoepithelial 2 cells, as shown in Fig.5h. Hence, we believed that the interactions (e.g., *COMP* -> *ITGAV*, *CD47*) should be restricted to myoepithelial 2 cells. In light of the comment, we have included the gene expression distribution plot into Fig.5h and the related text on pages 11, 38, and 39.

Fig.5. stKeep identifies key TFs, ligands, and receptors in tumor-associated myoepithelial cells using paired scRNA-seq data from the TNBC sample. **a** H&E plot of the tissue with manual annotation of 16 regions based on histological features: invasive carcinoma (red), carcinoma in situ (orange), lymphocyte (golden), stromal and adipose (blue), and normal ductal (purple). **b** Spatial clustering of cells by stKeep, indicated by different colors. **c** UMAP plot showing 10 cell subtypes of 1,627 epithelial cells (left panel), with a dot plot displaying marker gene expression for each subtype (right panel). Note the marker genes for each subtype are as follows: luminal progenitor 1 (*S100A1*, *PPP1R1B*, and *RGCC*), luminal progenitor 2 (*MMP7*, *KRT23*, and *SLC34A2*), myoepithelial 1 (*C2orf40*, *MYH11*, *CNN1*, and *OXTR*), myoepithelial 2 (*CDH13*, *COMP*, *THY1*, *FST*, *BGN*, *IGFBP6*, and *S100A2*), mature luminal (*TFF3*, *AGR3*, *TFF1*, *AFF3*, and *FBP1*), cancer cycling (*RRM2*, *CCNA2*, *UBE2C*, *CDK1*, *CDCA3*, *NUF2*, *MND1*, and *CDCA8*), cancer basal 1 (*TMSB10*, *GDI2*, and *VIM*), cancer basal 2 (*KIF1A*, *SLCO1A2*, and *DCLK1*), cancer basal 3 (*SOX11*, *RNF144A*, *CTXN1*, *TTL7*, and *GPC2*), cancer basal 4 (*NIN2*, *SLCO5A1*, and *GGACT*). Dot size and color indicate the percentage

and mean expression level of each gene. **d** Heatmap showing the proportion of 10 epithelial cell subtypes in different spatial clusters. **e** Spatial distribution of luminal progenitor 2 in IDC and BAS1 samples. **f** Gene set enrichment analysis (GSEA) for differentially expressed genes (DEGs) in myoepithelial 2 compared to myoepithelial 1. **g** The gene-module specific to myoepithelial 2 cells by stKeep. Presented here the top 200 over-expressed genes characterized by myoepithelial 2 in comparison to other clusters. A regulator and its target genes are colored in red and blue, respectively. **h** Scatter plot displaying the specificity level (evaluated by the Gini index) and fold change level of the up-regulated ligand genes in myoepithelial 2 compared to myoepithelial 1 (left panel). UMAP plot of *COMP* expression levels (right panel). **i** Spatial expression of *COMP* and its interaction strength with five receptors. **j** Overall survival rate of TNBC patients based on *COMP* expression using TCGA by GEPIA2. Source data are provided as a Source Data file.

13 Tang, Z., Kang, B., Li, C., Chen, T. & Zhang, Z. GEPIA2: an enhanced web server for large-scale expression profiling and interactive analysis. *Nucleic Acids Res* **47**, W556-W560 (2019).

Comment #12: Can the authors clarify why all significance values in the paper result in 2.22e-16?

Response: Yes, the software R automatically use 2.22e-16 to denote very small numbers close to zero in the default significant digits. To clarify this, we have added the text on page 6.

REVIEWER COMMENTS

Reviewer #1 (Remarks to the Author):

The authors have properly addressed all my comments. I believe the manuscript has been substantially improved and suitable for publication.

Reviewer #2 (Remarks to the Author):

We thank the authors for the effort they've put in carefully addressing the comments. All comments have been well-addressed, particularly the scalability of the method and contribution of each graph to the quality of the downstream results. The relevant information has been added/modified in the manuscript and supplementary data. We have no further comments.

Reviewer #2 (Remarks on code availability):

Code and data are presented clearly and there are sufficient instructions to run the software and example analysis.

Reviewer #3 (Remarks to the Author):

Reviewer #4 (Remarks to the Author):

Thank you to the authors for their detailed rebuttal responses. The inclusion of new metrics (e.g. silhouette scores), re-analysis of additional high-res spatial transcriptomics datasets, and additional clarifications improves the interpretation of findings from the stKEEP method. I am satisfied with most of the revisions made to the manuscript, and I believe this method will be a useful addition to the community for analyzing spatial transcriptomics data.

However, the findings related to tumor-associated myoepithelial cells and their cell-cell interactions in breast cancer (Figure 5) are underwhelming. My main concern is that this finding is concluded from spatial transcriptomics data on the Visium platform which lacks cellular resolution. Myoepithelial cells share many transcriptional similarities with other cell types including cancer-associated fibroblasts (CAFs) and basal-like breast cancer cells. Thus, inferring their spatial location and cell-cell communication should certainly factor in other cell types of the tumor microenvironment such as fibroblasts. This can help avoid signals being confounded by non-epithelial cell types that express similar ligands and receptors. For example, activated fibroblasts also express markers of myoepithelial cells, such as Acta2, Thy1, and the ligand COMP, and these cells are highly abundant in invasive breast cancer regions, suggesting that the mapping of myoepithelial cells (Fig 5d) and the expression of COMP (Fig 5i) are likely confounded by fibroblasts. In Figure 5d, myoepithelial populations-1 and -2 also exhibit low distributions following spatial mapping with GraphST in invasive carcinoma domains compared to other epithelial cell types, suggesting that this signal are likely confounded by other cell types, such as fibroblasts, not factored in to the analysis. Furthermore, this low mapping is consistent with the known histology of myoepithelial cells around ductal structures, which are not really present in invasive carcinoma regions. The authors should reconsider making such strong conclusions on myoepithelial cells based on this analysis alone, or make major revisions to this specific analysis to strengthen these results significantly.

Reviewer #4 (Remarks on code availability):

Code repository is really well documented with clear installation instructions, and appears to be ready for use by the community.

Reviewer #1

Comment #1: The authors have properly addressed all my comments. I believe the manuscript has been substantially improved and suitable for publication.

Response: We thank the reviewer for the comment and recognition.

Reviewer #2

Comment #1: We thank the authors for the effort they've put in carefully addressing the comments. All comments have been well-addressed, particularly the scalability of the method and contribution of each graph to the quality of the downstream results. The relevant information has been added/modified in the manuscript and supplementary data. We have no further comments.

Response: Yes, we appreciate the reviewer for the comment and recognition.

Comment #2: Code and data are presented clearly and there are sufficient instructions to run the software and example analysis.

Response: We thank the reviewer for the comment.

Reviewer #3

Comment #1: I co-reviewed this manuscript with one of the reviewers who provided the listed reports. This is part of the Nature Communications initiative to facilitate training in peer review and to provide appropriate recognition for Early Career Researchers who co-review manuscripts.

Response: We thank the reviewer for the comment.

Reviewer #4

Comment #1: Thank you to the authors for their detailed rebuttal responses. The inclusion of new metrics (e.g., silhouette scores), re-analysis of additional high-res spatial transcriptomics

datasets, and additional clarifications improves the interpretation of findings from the stKEEP method. I am satisfied with most of the revisions made to the manuscript, and I believe this method will be a useful addition to the community for analyzing spatial transcriptomics data.

Response: Yes, we appreciate the reviewer for the comment and approval.

Comment #2: However, the findings related to tumor-associated myoepithelial cells and their cell-cell interactions in breast cancer (Figure 5) are underwhelming. My main concern is that this finding is concluded from spatial transcriptomics data on the Visium platform which lacks cellular resolution. Myoepithelial cells share many transcriptional similarities with other cell types including cancer-associated fibroblasts (CAFs) and basal-like breast cancer cells. Thus, inferring their spatial location and cell-cell communication should certainly factor in other cell types of the tumor microenvironment such as fibroblasts. This can help avoid signals being confounded by non-epithelial cell types that express similar ligands and receptors. For example, activated fibroblasts also express markers of myoepithelial cells, such as *Acta2*, *Thy1*, and the ligand *COMP*, and these cells are highly abundant in invasive breast cancer regions, suggesting that the mapping of myoepithelial cells (Fig 5d) and the expression of *COMP* (Fig 5i) are likely confounded by fibroblasts. In Figure 5d, myoepithelial populations-1 and -2 also exhibit low distributions following spatial mapping with GraphST in invasive carcinoma domains compared to other epithelial cell types, suggesting that this signal is likely confounded by other cell types, such as fibroblasts, not factored into the analysis. Furthermore, this low mapping is consistent with the known histology of myoepithelial cells around ductal structures, which are not really present in invasive carcinoma regions. The authors should reconsider making such strong conclusions on myoepithelial cells based on this analysis alone, or make major revisions to this specific analysis to strengthen these results significantly.

Response: We agree with the reviewer for the comment. In light of the comment, we have done the following. Specifically, we have (i) redefined the marker genes for myoepithelial 2 by considering their specificity relative to all other cell types, including *FST*, *S100A2*, and *CALML3*; (ii) analyzed the gene expression distribution of myoepithelial 2 markers (*FST*, *S100A2*, and *CALML3*), commonly used myoepithelial markers in breast pathology ¹ (*CNN1*, *NGFR*, *CDH3*, *SERPINB5*, *KRT14*, *KRT17*, and *TP63*), and markers for cancer-associated fibroblasts (CAFs) (*COL1A2*, *COL3A1*, *DCN*, *MMP2*, and *COL1A1*), as well as *COMP*, using single-cell RNA-seq and spatial transcriptomics data; and (iii) re-calculated the specificity of ligands secreted by myoepithelial 2 cells versus all other cell types using the Gini index ² (a

statistical measure to estimate a degree of inequality in the distribution of genes among different cell populations). Upon analysis, we observed that

- (i) myoepithelial cells are enriched in both normal ductal and invasive carcinoma regions, especially myoepithelial 2 cells, which are primarily situated at the margins of the carcinoma nest and in contact with the stroma, aligning with prior findings in triple-negative breast cancer (TNBC) ^{3,4};
- (ii) *COMP* is exclusively secreted by myoepithelial 2 cells compared to all other cell types using the Gini index;
- (iii) in the invasive carcinoma region, the signature gene scores of myoepithelial 2 cells and CAFs are significantly correlated with *COMP* expression, with coefficients of 0.38 and 0.49, respectively. Notably, myoepithelial 2 cells express *COMP* much more than CAFs in the scRNA-seq data; and
- (iv) strong interactions between *COMP* and its receptors (*ITGAV*, *SDC1*, *ITGB1*, and *ITGA5*) are inferred by stKeep in the invasive cancer region. These inferred CCCs are significantly associated with the gene signature scores of myoepithelial 2 cells and CAFs, respectively. Furthermore, these receptors are involved in cancer progression, metastasis, and invasiveness ⁵⁻⁸.

Collectively, both myoepithelial 2 cells and CAFs can promote cancer invasiveness through the expression of *COMP*. Additionally, our analysis revealed that *COMP* is regulated by *TP63*, suggesting that myoepithelial 2 cells potentially upregulate *TP63* and its target *COMP* to enhance cancer invasiveness. Please refer to the revised Fig.5c, g-l for details. In light of the comment, we have revised the text on pages 11, 12, 39, and 40.

Fig.5. stKeep identifies key TFs, ligands, and receptors in tumor-associated myoepithelial cells using paired scRNA-seq data from the TNBC sample. **a** H&E plot of the tissue with manual annotation of 16 regions based on histological features: invasive carcinoma (red), carcinoma in situ (orange), lymphocyte (golden), stromal and adipose (blue), and normal ductal (purple). **b** Spatial clustering by stKeep, indicated by different colors. **c** UMAP plot showing 10 cell subtypes of 1,627 epithelial cells (left panel), with a dot plot displaying marker gene expression for each subtype (right panel). Note that the marker genes for each subtype are as

follows: luminal progenitor 1 (*SI00A1*, *PPP1R1B*, and *RGCC*), luminal progenitor 2 (*MMP7*, *KRT23*, and *SLC34A2*), myoepithelial 1 (*C2orf40*, *MYH11*, *CNN1*, and *OXTR*), myoepithelial 2 (*FST*, *SI00A2*, and *CALML3*), mature luminal (*TFF3*, *AGR3*, *TFF1*, *AFF3*, and *FBP1*), cancer cycling (*RRM2*, *CCNA2*, *UBE2C*, *CDK1*, *CDCA3*, *NUF2*, *MND1*, and *CDCA8*), cancer basal 1 (*TMSB10*, *GDI2*, and *VIM*), cancer basal 2 (*KIF1A*, *SLCO1A2*, and *DCLK1*), cancer basal 3 (*SOX11*, *RNF144A*, *CTXN1*, *TLL7*, and *GPC2*), cancer basal 4 (*NINJ2*, *SLCO5A1*, and *GGACT*). Dot size and color indicate the percentage and mean expression level of each gene. **d** Heatmap showing the proportion of 10 epithelial cell subtypes in different spatial clusters. **e** Spatial distribution of luminal progenitor 2 in IDC and BAS1 samples. **f** Gene set enrichment analysis (GSEA) for DEGs in myoepithelial 2 compared to myoepithelial 1. **g** The gene-module specific to myoepithelial 2 cells by stKeep. Presented here are the top 200 over-expressed genes characterized by myoepithelial 2 in comparison to other clusters. A regulator and its target genes are colored in red and blue, respectively. **h** Scatter plot displaying the specificity level (evaluated by the Gini index) and fold change level of the up-regulated ligand genes in myoepithelial 2 compared to myoepithelial 1. **i** Dot plot displaying expression levels of representative genes for myoepithelial 2 cell, myoepithelial cell, and CAF across various cell types. **j** Spatial feature plots of gene signature score for myoepithelial 2 cell (*CALML3*, *FST*, and *SI00A2*), myoepithelial cell (*CNN1*, *NGFR*, *CDH3*, *SERPINB5*, *KRT14*, *KRT17*, and *TP63*), and CAF (*COL1A2*, *COL3A1*, *DCN*, *MMP2*, and *COL1A1*). **k** Spatial expression of *COMP* and its interaction strength with four receptors. **l** Spearman correlation between inferred CCCs associated with *COMP* and gene signature scores for CAFs and myoepithelial 2 cells. **m** Overall survival rate of TNBC patients based on *COMP* expression using TCGA by GEPIA2. Source data are provided as a Source Data file.

- 1 Dewar, R., Fadare, O., Gilmore, H. & Gown, A. M. Best practices in diagnostic immunohistochemistry: myoepithelial markers in breast pathology. *Archives of pathology & laboratory medicine* **135**, 422-429 (2011).
- 2 Zuo, C. *et al.* Elucidating tumor heterogeneity from spatially resolved transcriptomics data by multi-view graph collaborative learning. *Nat Commun* **13**, 5962 (2022).
- 3 Hayashi, Y., Aoki, Y., Eto, R. & Tokuoka, S. Findings Of Myoepithelial Cells In Human Breast Cancer Ultrastructural And Immunohistochemical Study By Means Of Anti-Myosin Antibody. *Pathology International* **34**, 537-552 (1984).
- 4 Cima, L. *et al.* Triple-negative breast carcinomas of low malignant potential: review on diagnostic criteria and differential diagnoses. *Virchows Archiv*, 1-18 (2022).

- 5 Cheuk, I. W.-Y. *et al.* ITGAV targeting as a therapeutic approach for treatment of metastatic breast cancer. *American journal of cancer research* **10**, 211 (2020).
- 6 Pantano, F. *et al.* Integrin alpha5 in human breast cancer is a mediator of bone metastasis and a therapeutic target for the treatment of osteolytic lesions. *Oncogene* **40**, 1284-1299 (2021).
- 7 Chute, C. *et al.* Syndecan-1 induction in lung microenvironment supports the establishment of breast tumor metastases. *Breast cancer research* **20**, 1-12 (2018).
- 8 Barnawi, R. *et al.* β 1 Integrin is essential for fascin-mediated breast cancer stem cell function and disease progression. *International Journal of Cancer* **145**, 830-841 (2019).

Comment #3: Code repository is really well documented with clear installation instructions, and appears to be ready for use by the community.

Response: We thank the reviewer for the comment.

REVIEWERS' COMMENTS

Reviewer #4 (Remarks to the Author):

Thank you to the authors for the additional analysis of cancer-associated fibroblasts in their findings of myoepithelial cells. Showing that the expression of COMP can be derived from other types, such as CAFs, is an important acknowledgement of the limitation in studying single-cell gene expression across low resolution spatial transcriptomics datasets such as Visium.

On this note, it is also important for the authors to clarify in the text that these myoepithelial-2 cells could also be neoplastic/cancer in origin and have undergone malignant transformation. The cited study #57 also describes neoplastic myoepithelial cells in invasive carcinoma. Signal from myoepithelial-2 cells is clearly in an invasive breast carcinoma region (indicated by pathology) and there is not enough evidence to simply call them 'tumor-associated' unless the authors present data to show that they are otherwise non-malignant.

Reviewer #4

Comment #1: Thank you to the authors for the additional analysis of cancer-associated fibroblasts in their findings of myoepithelial cells. Showing that the expression of COMP can be derived from other types, such as CAFs, is an important acknowledgement of the limitation in studying single-cell gene expression across low resolution spatial transcriptomics datasets such as Visium.

Response: Yes, we thank the reviewer for the comment and approval.

Comment #2: On this note, it is also important for the authors to clarify in the text that these myoepithelial-2 cells could also be neoplastic/cancer in origin and have undergone malignant transformation. The cited study #57 also describes neoplastic myoepithelial cells in invasive carcinoma. Signal from myoepithelial-2 cells is clearly in an invasive breast carcinoma region (indicated by pathology) and there is not enough evidence to simply call them 'tumor-associated' unless the authors present data to show that they are otherwise non-malignant.

Response: We agree with the reviewer for the comment. As suggested, we have made the following changes. Specifically,

- (1) We have replaced “tumor-associated myoepithelial cells” with “neoplastic myoepithelial cells” throughout the whole manuscript; and
- (2) We have included the following statement: “these cells may originate from neoplastic/cancer and undergo malignant transformation” on page 11.